# Alterations in the amplitude and burst rate of beta oscillations impair reward-dependent motor learning in anxiety

**Sebastian Sporn[1,2], Thomas Hein[2], Maria Herrojo Ruiz[2,3]\***

[1]School of Psychology, University of Birmingham, Birmingham, United Kingdom; [2]Department of Psychology, Goldsmiths University of London, London, United Kingdom; [3]Center for Cognition and Decision Making, Institute for Cognitive Neuroscience, National Research University Higher School of Economics, Moscow, Russian Federation

**Abstract** Anxiety results in sub-optimal motor learning, but the precise mechanisms through which this effect occurs remain unknown. Using a motor sequence learning paradigm with separate phases for initial exploration and reward-based learning, we show that anxiety states in humans impair learning by attenuating the update of reward estimates. Further, when such estimates are perceived as unstable over time (volatility), anxiety constrains adaptive behavioral changes. Neurally, anxiety during initial exploration increased the amplitude and the rate of long bursts of sensorimotor and prefrontal beta oscillations (13–30 Hz). These changes extended to the subsequent learning phase, where phasic increases in beta power and burst rate following reward feedback were linked to smaller updates in reward estimates, with a higher anxiety-related increase explaining the attenuated belief updating. These data suggest that state anxiety alters the dynamics of beta oscillations during reward processing, thereby impairing proper updating of motor predictions when learning in unstable environments.

**\*For correspondence:**
M.Herrojo-Ruiz@gold.ac.uk

**Competing interests:** The authors declare that no competing interests exist.

## Introduction

Anxiety involves anticipatory changes in physiological and psychological responses to an uncertain future threat (*Grupe and Nitschke, 2013*; *Bishop, 2007*). Previous studies have established that trait anxiety interferes with prefrontal control of attention in perceptual tasks, whereas state anxiety modulates the amygdala during detection of threat-related stimuli (*Bishop, 2007*; *Bishop, 2009*). An emerging literature additionally identifies the dorsomedial and dorsolateral prefrontal cortex (dmPPC and dlPFC) and the dorsal anterior cingulate cortex (dACC) as central brain regions modulating sustained anxiety, both in subclinical and clinical populations (*Robinson et al., 2019*).

Computational modeling work has started to examine the mechanisms through which anxiety might impair learning, revealing that individuals with high trait anxiety do not correctly estimate the likelihood of outcomes during aversive or reward learning in uncertain environments (*Browning et al., 2015*; *Huang et al., 2017*; *Pulcu and Browning, 2019*). In the area of motor control, research has shown that stress and anxiety have detrimental effects on performance (*Baumeister, 1984*; *Beilock and Carr, 2001*). These results have been interpreted as anxiety interferring with information-processing resources, and as a shift towards an inward focus of attention and an increase in conscious processing of movement (*Eysenck and Calvo, 1992*; *Pijpers et al., 2005*). The effects of anxiety on motor learning are, however, often inconsistent, and a mechanistic understanding of these effects is still lacking. Delineating mechanisms through which anxiety influences motor learning is important to ameliorate the impact of anxiety in different settings, including in motor rehabilitation programs.

**eLife digest** Feeling anxious can hinder how well someone performs a task, a phenomenon that is sometimes called "choking under pressure". Anxiety may also impair a person's ability to learn a new manual task, like juggling or playing the piano; however, it remains unclear exactly how this happens.

People learn manual tasks more quickly if they can practice first, and the more someone varies their movements during these trial runs, the faster they learn afterwards. Yet, anxiety can affect movement; for example, anxious people often make repetitive motions like hand-wringing or fidgeting. There is also evidence that very anxious people may learn less from the outcomes of their actions.

To understand how anxiety may affect the learning of manual tasks, Sporn et al designed experiments where people learned to play a short sequence of notes on a piano. The main experiment involved 60 participants and was split over two phases. In the first 'exploration' phase, participants had to play the piano sequence using any timing they liked and were encouraged to explore different rhythms. In the second 'learning' phase, participants were rewarded with a higher score the closer they got to playing the notes with a certain rhythm, without being told that this was their specific goal.

To see how anxiety affected performance, the participants were split into three groups. One group were told in the initial exploration phase that they would give a public talk after they completed the piano task, which reliably made them more anxious. A second group were told about the anxiety-inducing public speaking only during the learning phase; while a third group – the controls – were not aware of any public speaking task.

People in the second group could learn the rhythm as well as the controls. Participants who were made anxious during the exploration phase, however, scored fewer points and were less likely to learn the piano sequence in the second phase. They also varied their movements less in the first phase.

As a follow-up, Sporn et al. repeated the experiment with 26 people but without the initial exploration phase. This time the anxious participants were less able to learn the piano sequence and scored fewer points. This suggests that the initial exploration in the previous experiment had enabled later anxious participants to succeed in the learning phase despite being anxious.

Finally, Sporn et al. also used a technique called electroencephalography (or EEG for short) to record brain activity and observed differences in participants with and without anxiety, particularly when they received their scores. The EEG signals showed that anxiety altered rhythmic patterns of brain activity called "sensorimotor beta oscillations", which are known to be involved in both movement and learning.

Motor variability could be one component of motor learning that is affected by anxiety; it is defined as the variation of performance across repetitions (*van Beers et al., 2004*), and is affected by various factors including sensory and neuromuscular noise (*He et al., 2016*). As a form of action exploration, movement variability is increasingly recognized to benefit motor learning (*Todorov and Jordan, 2002*; *Wu et al., 2014*; *Pekny et al., 2015*), particularly during reward-based learning, with discrepant effects in motor adaptation paradigms (*He et al., 2016*; *Singh et al., 2016*). These findings are consistent with the vast amount of research on reinforcement learning that demonstrates increased learning following initial exploration (*Sutton and Barto, 1998*; *Olveczky et al., 2005*).

Yet contextual factors can reduce variability. For instance, an induced anxiety state leads to ritualistic behavior, characterized by movement redundancy, repetition, and rigidity (*Lang et al., 2015*). This finding resembles the reduction in behavioral variability and exploration that manifests across animal species during phasic fear in reaction to certain imminent threats (*Morgan and Tromborg, 2007*). On the basis of these results, we set out to test the hypothesis that state anxiety modulates motor learning through a reduction in motor variability.

A second component that could be influenced by anxiety is the flexibility to adapt to changes in the task structure during learning. Individuals who are affected by anxiety disorders exhibit an intolerance of uncertainty, which contributes to excessive worry and emotional dysregulation

(*Ouellet et al., 2019*). Turning to non-clinical populations, computational studies have established that highly anxious individuals exhibit difficulties in estimating environmental uncertainty both in aversive and reward-based tasks (*Browning et al., 2015*; *Huang et al., 2017*; *Pulcu and Browning, 2019*). Failure to adapt to volatile or unstable environments thus impairs learning of action-outcome contingencies in these settings. Accordingly, in the context of motor learning, and more specifically, in reward-based motor learning, we proposed that an increase in anxiety would affect individuals' estimation of uncertainty about the stability of the task structure, such as the rewarded movement.

On the neural level, we posited that changes in motor variability are driven by activity in premotor and motor areas. Support for our hypothesis comes from animal studies demonstrating that variability in the primate premotor cortex tracks behavioral variability during motor planning (*Churchland et al., 2006*). Further evidence supports the hypothesis that changes in variability in single-neuron activity in motor cortex drive motor exploration during initial learning, and reduce it following intensive training (*Mandelblat-Cerf et al., 2009*; *Santos et al., 2015*). In addition, the basal ganglia are crucial for modulating variability during learning and production, as shown in songbirds and, indirectly, in patients with Parkinson's disease (*Kao et al., 2005*; *Olveczky et al., 2005*; *Pekny et al., 2015*).

In the present study, we analyzed sensorimotor beta oscillations (13–30 Hz) as a candidate brain rhythm associated with the modulation of motor exploration and variability. Beta oscillations are modulated with different aspects of performance and motor learning (*Herrojo Ruiz et al., 2014*; *Bartolo and Merchant, 2015*; *Tan et al., 2014*), as well as in reward-based learning (*HajiHosseini et al., 2012*). Increases in sensorimotor beta power following movement have been proposed to signal greater reliance on prior information about the optimal movement (*Tan et al., 2016*), which would reduce the impact of new evidence on the update of motor commands. We therefore tested the additional hypothesis that changes in sensorimotor beta oscillations mediate the effect of anxiety on belief updates and the estimation of uncertainty driving reward-based motor learning. Crucially, in addition to assessing sensorimotor brain regions, we were interested in prefrontal areas because of prior work in clinical and subclinical anxiety linking the prefrontal cortex (dmPFC and dlPFC) and the dACC to the maintenance of anxiety states, including worry and threat appraisal (*Grupe and Nitschke, 2013*; *Robinson et al., 2019*). Thus, beta oscillations across sensorimotor and prefrontal electrode regions were evaluated.

Traditionally, the primary focus of research on oscillations was on power changes, although there is a renewed interest in assessing dynamic properties of oscillatory activity, such as the presence of brief bursts (*Poil et al., 2008*). Brief oscillation bursts are considered to be a central feature of physiological beta waves in motor-premotor cortex and the basal ganglia (*Feingold et al., 2015*; *Tinkhauser et al., 2017*; *Little et al., 2018*). Accordingly, we assessed both the power and burst distribution of beta oscillations to capture dynamic changes in neural activity that were induced by anxiety and their link to behavioral effects. To test our hypotheses, we recorded electroencephalography (EEG) in three groups of participants while they completed a reward-based motor sequence learning paradigm, with separate phases for motor exploration (without reinforcement) and reward-based learning (using reinforcement). We manipulated anxiety by informing participants about an upcoming public speaking task (*Lang et al., 2015*). Using a between-subject design, the anxiety manipulation targeted either the motor exploration or the reward-based learning phase. Analysis of the EEG signals aimed to assess anxiety-related changes in the power and burst distribution in sensorimotor and prefrontal beta oscillations in relation to changes in behavioral variability and reward-based learning.

## Results

Sixty participants completed our reward-based motor sequence learning task, consisting of three blocks of 100 trials each over two phases (*Figure 1*): an initial motor exploration (block1, termed exploration hereafter) and a reward-based learning phase (block2 and block3: termed learning hereafter). The rationale for including a motor exploration phase in which participants did not receive trial-based feedback or reinforcement was based on findings indicating that initial motor variability (in the absence of reinforcement) can influence the rate at which participants learn in a subsequent motor task (*Wu et al., 2014*). If state anxiety reduces the expression of motor variability during the exploration phase, subsequent motor learning would be affected.

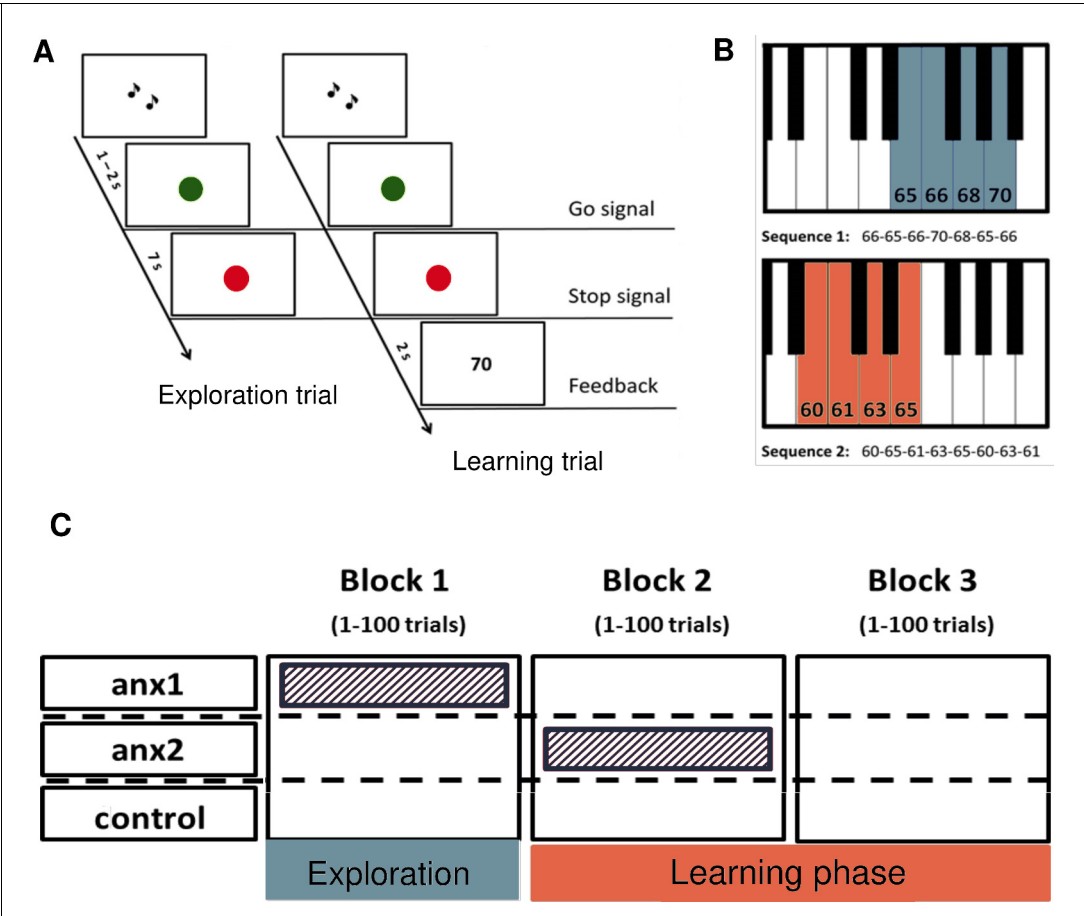

**Figure 1.** A novel paradigm for testing reward-based motor sequence learning. (**A**) Schematic of the task. Participants performed sequence1 during 100 initial exploration trials, followed by 200 trials over two blocks of reward-based learning performing sequence2. During the learning blocks, participants received a performance-related score between 0–100 that would lead to monetary reward. (**B**) The pitch content of the sequences used in the exploration (sequence1) and reward-based learning blocks (sequence2), respectively. (**C**) Schematic of the anxiety manipulation. The shaded area denotes the phase in which anxiety was induced in each group, using the threat of an upcoming public speaking task, which took place immediately after that block was completed.

Prior to the experimental task, we recorded 3 min of EEG at rest with eyes open in each participant. Next, on a digital piano, participants played two different sequences of seven and eight notes during the exploration and learning phases, respectively (*Figure 1B*). The sequence patterns were designed so that the key presses would span a range of four neighboring keys on the piano. Participants were explicitly taught the tone sequences prior to the start of the experiment, yet precise instructions about the timing or loudness (keystroke velocity, Kvel) were not provided. The rationale for selecting two different sequences for the exploration and learning phases was to avoid carry-over effects of learning or a preferred performance pattern from the exploration period into the reward-based learning phase (following *Wu et al., 2014*).

During the initial exploration phase, participants were informed that they could freely change the pattern of temporal intervals between key presses (inter-keystroke intervals, IKIs) and/or the loudness of the performance in every trial, and that no reward or feedback would be provided. During learning, performance-based feedback in the form of a 0–100 score was provided at the end of each trial. Participants were informed that the overall average score would be translated into monetary reward. They were directly instructed to explore the temporal or loudness dimension (or both) and to use feedback scores to discover the unknown performance objective (which, unbeknownst to them, was related to the pattern of IKIs). The task-related dimension was therefore timing, whereas keystroke velocity was the non-task related dimension.

The performance measure that was rewarded during learning was the vector norm of the pattern of temporal differences between adjacent IKIs (see 'Materials and experimental design'). Different combinations of IKIs could lead to the same rewarded norm of IKI-difference values, and therefore to the same score. Participants were unaware of the existence of these multiple solutions. The multiplicity in the mapping between performance and score could lead participants to perceive an increased level of volatility in the environment (changes in the rewarded performance over time). This motivated us to assess their estimation of volatility during reward-based learning and its modulation by anxiety. In addition, we investigated whether higher initial variability would lead to higher scores during subsequent reward-based learning, independently of changes in variability during this latter phase. If initial exploration improves learning of the mapping between the actions and their sensory consequences (even without external feedback), then participants could learn better from performance-related feedback during the learning phase regardless of their use of variability in this phase. Alternatively, it could be that participants who also use more variability during learning discover the hidden goal by chance.

Participants were pseudo-randomly allocated to either a control group or to one of two experimental groups (*Figure 1C*): anxiety during exploration (anx1); and anxiety during the first block of learning (anx2). We measured changes in heart-rate variability (HRV) and heart-rate (HR) four times throughout the experimental session: resting state (3 min, prior to performance blocks); block1; block2; and block3. In addition, the state subscale from the State-Trait Anxiety Inventory (STAI, state scale X1, 20 items; *Spielberger, 1970*) was assessed four times: prior to the resting state recording and also immediately before the beginning of each block, and thus after the induction of anxiety in the experimental groups. The HRV index and STAI state anxiety subscale were able to dissociate in each experimental group between the phase targeted by the anxiety manipulation and the initial resting phase (within-group effects, see statistical results in *Figure 2*). In addition, significant between-group differences in HRV (not in STAI) further confirmed the specificity of the HRV changes in the targeted blocks (statistical details in *Figure 2*). These results confirmed that the experimental manipulation succeeded in inducing physiological and psychological responses within each experimental group that were consistent with an anxious state during the targeted phase, as reported previously (*Feldman et al., 2004*).

Statistical analysis of behavioral and neural measures focused on the separate comparison between each experimental group and the control group (contrasts: anx1 – controls, anx2 – controls). See 'Materials and methods'.

## Behavioral results

### Lower initial task-related variability is associated with poorer reward-based learning

All groups of participants demonstrated significant improvement in the achieved scores during reward-based learning, confirming that they effectively used feedback to approach the hidden target performance (changes in average score from block2 to block3 — anx1: p=0.008, non-parametric effect size estimator for dependent samples, $\Delta_{dep}$ = 0.93, 95% confidence interval, termed simply CI hereafter, CI = [0.86, 0.99]; anx2: p=0.004, $\Delta_{dep}$ = 0.83, CI = [0.61, 0.95]; controls: p=0.001, $\Delta_{dep}$ = 0.92, CI = [0.72, 0.98]).

Assessment of motor variability was performed separately in the task-related temporal dimension and in the non-task-related keystroke velocity dimension. Temporal variability—and similarly for Kvel variability—was estimated using the across-trials coefficient of variation of IKI (termed cvIKI hereafter; *Figure 3A–B*). This index was computed in bins of 25 trials, which therefore provided four values per experimental block. We hypothesized that in the total population, a higher degree of task-related variability during the exploration phase (that is, playing different temporal patterns in each trial), and therefore higher cvIKI, would improve subsequent reward-based learning, as this latter phase rewarded the temporal dimension. A non-parametric rank correlation analysis across the 60 participants revealed that participants who achieved higher scores in the learning phase exhibited a larger across-trials cvIKI during the exploration period (Spearman $\rho = 0.45, P = 0.003$; *Figure 3C*).

A similar result was obtained when excluding anx1 participants from the correlation analysis, supporting the hypothesis that in the subsample of 40 participants who did not undergo the anxiety manipulation during exploration there was a significant association between the level of task-related

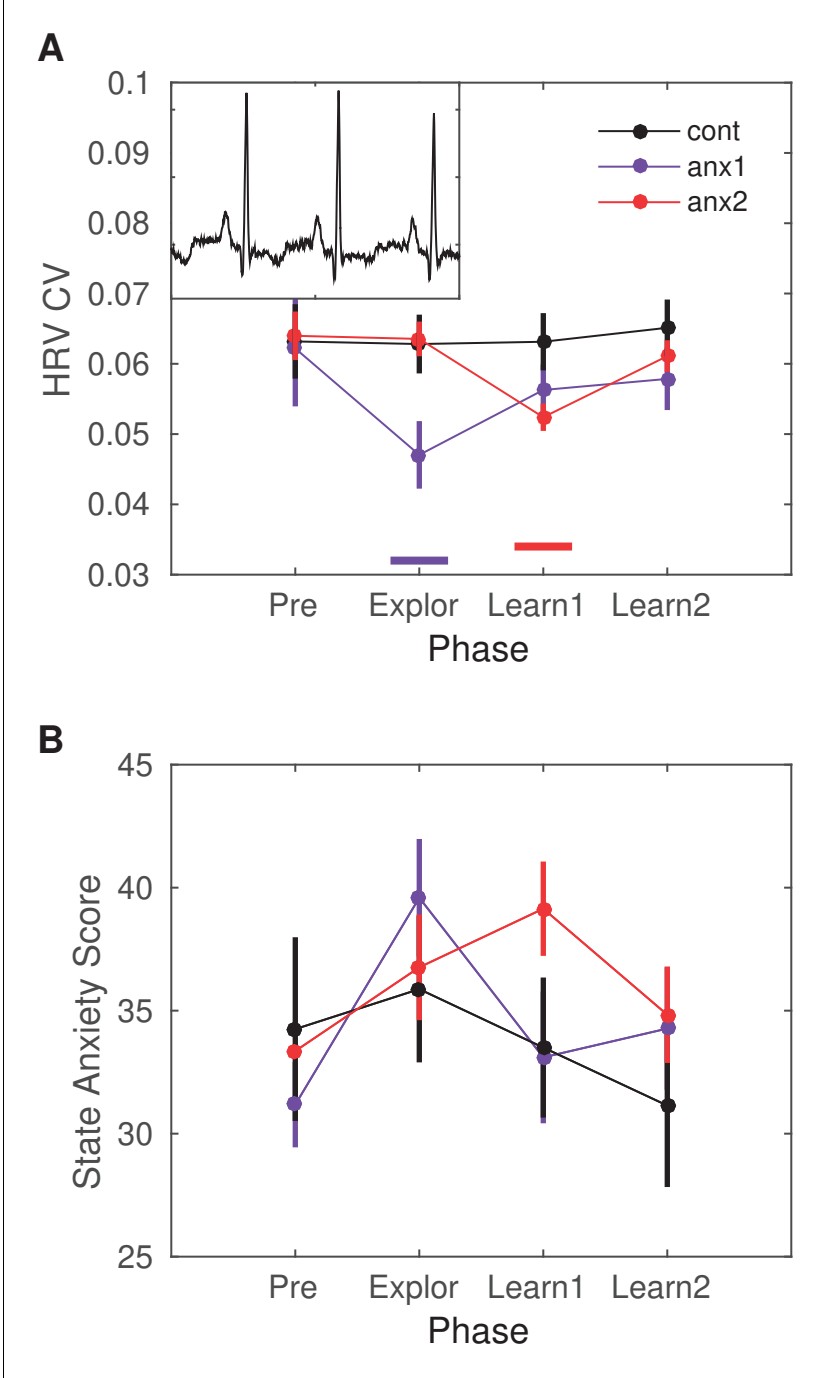

**Figure 2.** Heart-rate variability (HRV) modulation by the anxiety manipulation. (**A**) The average HRV measured as the coefficient of variation (CV) of the inter-beat-interval is displayed across the experimental blocks: initial resting state recording (Pre), initial exploration (Explor), first block of learning (Learn1) and, last block of learning (Learn2). Relative to Pre, there was a significant drop in HRV in anx1 participants during initial exploration (within-subject statistics with paired permutation tests, $P<0.05$ after controlling the false discovery rate [FDR] at level q = 0.05 due to multiple comparisons, termed $P_{FDR} : P_{FDR}<0.05, \Delta_{dep} = 0.81, CI = [0.75, 0.87]$). In anx2 participants, the drop in HRV was found during the first learning block, which was targeted by the anxiety manipulation ($P_{FDR}<0.05, \Delta_{dep} = 0.78, CI = [0.71, 0.85]$). Between-group comparisons revealed that anx1, relative to the control group, exhibited a significantly lower HRV during the exploration phase ($P_{FDR}<0.05, \Delta = 0.75, CI = [0.65, 0.85]$, purple bar at the bottom). The anx2 group manifested a significant drop in HRV relative to controls during the first learning block ($P_{FDR}<0.05, \Delta = 0.71, CI = [0.62, 0.80]$, red bar at the bottom). These results demonstrate a group-

*Figure 2 continued on next page*

*Figure 2 continued*

specific modulation of anxiety relative to controls during the targeted blocks. The mean HR did not change within or between groups (*P*>0.05). (**B**) STAI state anxiety score in each group across the different experimental phases. Participants completed the STAI state anxiety subscale first at the start of the experiment before the resting state recording (Pre) and subsequently again immediately before each experimental block (and right after the anxiety induction: Explor, Learn1, Learn2). There was a within-group significant increase in the score for each experimental group during the phase targeted by the anxiety manipulation (anx1: Explor relative to Pre, average score 40 [2] and 31 [2], respectively; $P_{FDR}$<0.05, $\Delta_{dep} = 0.74, CI = [0.68, 0.80]$; anx2: Learn1 relative to Pre, average score 39 [2] and 34 [2], respectively; $P_{FDR}$<0.05, $\Delta_{dep} = 0.78, CI = [0.68, 0.86]$). Between-group differences were non-significant.

variability and the subsequent score ($\rho = 0.41, P = 0.04$). No significant rank correlation was found between the scores and cvKvel (*P*>0.05).

We also assessed whether the degree of cvIKI during learning was associated with the average score and found an inverted pattern: there was a significant negative non-parametric rank correlation between the cvIKI index and the mean score ($\rho = -0.44, P = 0.002$; ***Figure 3D***). No significant effect was found for the cvKvel parameter (*P*>0.05).

Notably, the amount of variability in timing and keystroke velocity used by participants was not correlated (cvIKI and cvKvel during initial exploration: $\rho = 0.021, P = 0.788$, and during learning: $\rho = -0.030, P = 0.844$). This indicates that in our task, participants could vary the temporal and velocity dimensions separately. On the other hand, however, the generally lower cvKvel values in all blocks and groups further indicate that participants may not have been able to substantially vary this dimension. Finally, the degree of cvIKI during the learning and exploration phases were not correlated ($\rho = 0.029, P = 0.848$). These findings suggest that achieving higher scores during reward-based learning in our paradigm cannot be accounted for by a general tendency towards more exploration throughout all experimental blocks. In fact, larger sustained task-related variability during learning was detrimental to maintaining the performance close to the inferred target (***Figure 3D***).

## Anxiety during initial exploration reduces task-related variability and impairs subsequent reward-based learning

Next, we assessed pair-wise differences in the behavioral measures between the control group and each experimental group (anx1 and anx2) separately. Participants who were affected by state anxiety during initial exploration (anx1) achieved significantly lower scores in the subsequent reward-based learning phase relative to control participants (***Figure 4A***: *P*<0.05 after controlling the false discovery rate [FDR] at level $q = 0.05$ due to multiple comparisons, termed $P_{FDR}$ thereafter; $\Delta = 0.78, CI = [0.54, 0.92]$). By contrast, in the anx2 group scores did not statistically differ from the scores in the control group ($P_{FDR}$>0.05). A planned comparison between both experimental groups demonstrated significantly higher scores in anx2 than in anx1 ($P_{FDR}$<0.05, $\Delta = 0.67, CI = [0.51, 0.80]$).

During the initial exploration block, anx1 used a lower degree of cvIKI than the control group (***Figure 4B***; $P_{FDR}$<0.05; $\Delta = 0.67, CI = [0.52, 0.85]$). There was no between-groups (anx1, controls) difference in cvKvel (***Figure 4C***; $P_{FDR}$>0.05). Performance in anx2 in this phase did not significantly differ from performance in the control group, either for cvIKI or for cvKvel ($P_{FDR}$>0.05).

Subsequently, during the learning blocks, there were no significant between-group differences in cvIKI or cvKvel ($P_{FDR}$>0.05). In each group, there was a significant drop in the use of temporal variability from the first to the second learning block, corresponding to a transition from exploration to the exploitation of the rewarded options (significant drop in cvIKI from block2 to block3 in control, anx1, and anx2 participants; $P_{FDR}$<0.05; effect size — $\Delta_{dep} = 0.77, CI = [0.53, 0.87]$ in controls; $\Delta_{dep} = 0.55, CI = [0.50, 0.61]$ in anx1; $\Delta_{dep} = 0.83, CI = [0.62, 0.94]$ in anx2). This outcome further indicated that all groups successfully completed the reward-based learning task, although anx1 participants achieved lower scores than the reference control group.

Detailed analyses of the trial-by-trial changes in scores and performance using a Bayesian learning model and their modulation by anxiety are reported below. General performance parameters, such as the average performance tempo or the mean keystroke velocity did not differ between groups, either during initial exploration or learning (*P*>0.05). Participants completed sequence1 in 3.0 (0.1) seconds on average, between 0.68 (0.05) and 3.68 (0.10) s after the GO signal (non-significant differences between groups, *P*>0.05). During learning, they played sequence2 with an average duration of

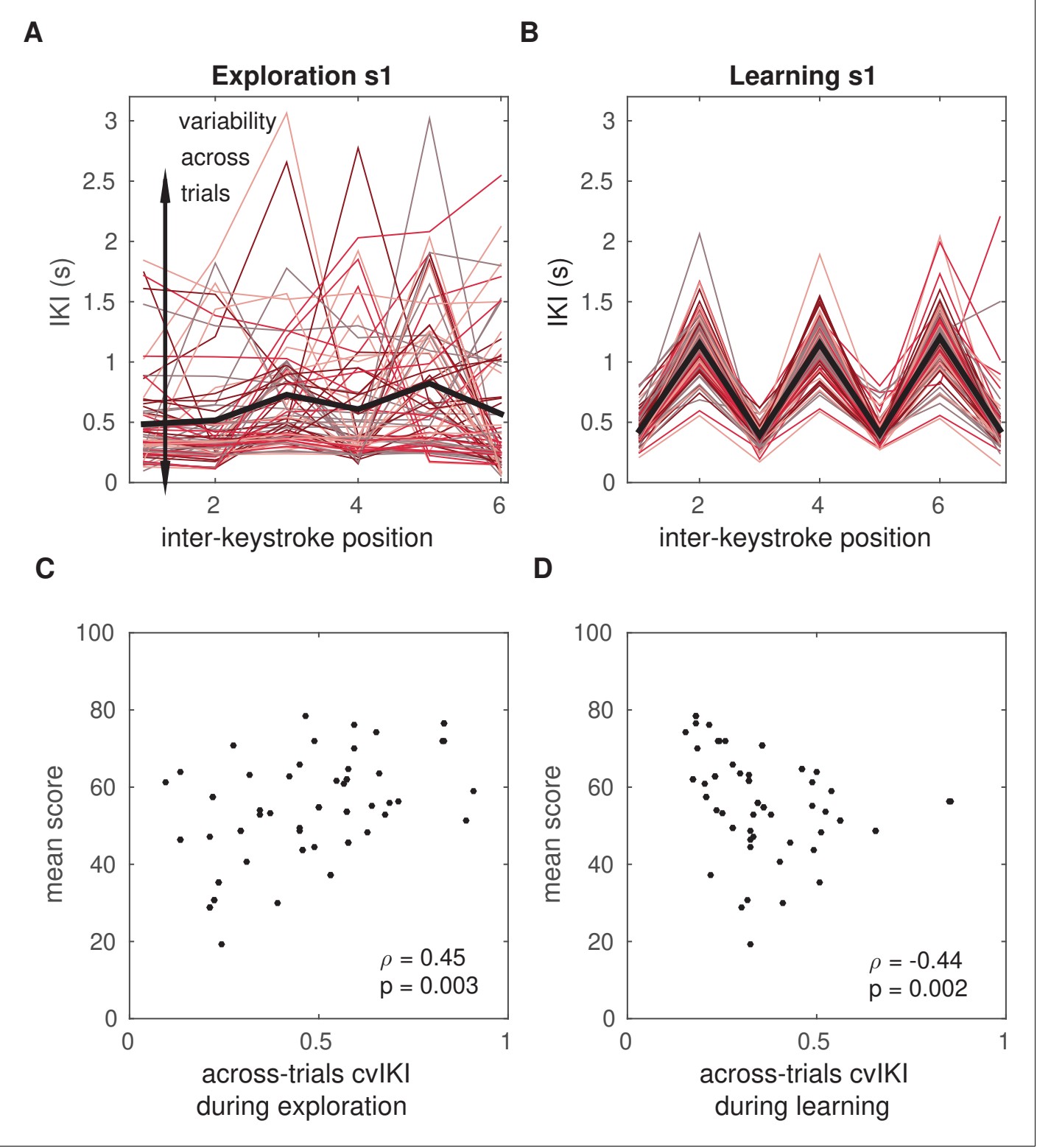

**Figure 3.** Temporal variability during initial exploration and during reward-based learning. (**A, B**) Illustration of timing performance during initial exploration (**A**) and learning (**B**) blocks for one representative participant, s1. The x-axis represents the position of the inter-keystroke interval (sequence1: seven notes, corresponding to six inter-keystroke temporal intervals; sequence2: eight notes, corresponding to seven inter-keystroke intervals). The y-axis shows the inter-keystroke interval (IKI) in ms. Black lines represent the mean IKI pattern. Red-colored traces represent the individual timing performance in each of the 100 (**A**) and 200 (**B**) trials during exploration and learning blocks, respectively. Task-related temporal variability was

*Figure 3 continued on next page*

Figure 3 continued

measured using the across-trials coefficient of variation of IKI, cvIKI. This measure was computed in successive bins of 25 trials, which allowed us to track changes in cvIKI across time. (C) Non-parametric rank correlation in the total population (N = 60) between the across-trials cvIKI during exploration (averaged across the four 25-trial bins) and the average score achieved subsequently during learning (Spearman $\rho = 0.45, P = 0.003$). (D) Same as panel (C) but using the individual value of the across-trials cvIKI from the learning phase (cvIKI was averaged here across all eight 25-trial bins; Spearman $\rho = -0.44, P = 0.002$).

4.7 (0.1) s, between 0.72 (0.03) and 5.35 (0.10) s (non-significant differences between groups, $P>0.05$). The mean learned solution was not significantly different between groups, either during the first or second learning block ($P>0.05$; *Figure 4—figure supplement 1*; but see trial-by-trial changes below).

These outcomes demonstrate that in our paradigm, state anxiety reduced task-related motor variability when induced during the exploration phase and this effect was associated with lower scores during subsequent reward-based learning. State anxiety, however, did not modulate task-related motor variability or the scores achieved when induced during reward-based learning. Finally, the different experimental manipulations did not affect the mean learned solution in each group.

## State anxiety during reward-based learning reduces learning rates if there is no prior exploration phase

Because anx2 participants performed at a level that was not significantly different from that found in control participants during learning, we asked whether the unconstrained motor exploration during the initial phase might have counteracted the effect of anxiety during learning blocks. Alternatively, it could be that the anxiety manipulation was not salient enough in the context of reward-based learning. To assess these alternative scenarios, we performed a control behavioral experiment with new experimental (anx3) and control groups (N = 13 each, see sample size estimation in 'Materials and methods'). Participants in each group performed the two learning blocks 2 and 3 (*Figure 1C*), but without completing a preceding exploration block. In anx3, state anxiety was induced exclusively during the first learning block, as in the original experiment. We found that the HRV index was significantly reduced in anx3 relative to controls during the manipulation phase ($P_{FDR}<0.05, \Delta = 0.72, CI = [0.62, 0.83]$), but not during the final learning phase (block3, $P_{FDR}>0.05$). STAI state subscale scores rose during the anxiety manipulation in anx3 (but not in controls) relative to the initial scores (within-group effect, $P_{FDR}<0.05, \Delta = 0.68, CI = [0.59, 0.78]$).

Overall, the anx3 group achieved a lower average score (and final monetary reward) than control participants ($P = 0.0256; \Delta = 0.64, CI = [0.50, 0.71]$). In addition, anx3 participants achieved significantly lower scores than control participants during the first learning block ($P_{FDR}<0.05, \Delta = 0.68, CI = [0.54, 0.79]$, *Figure 4D*), but not during the second learning block ($P_{FDR}>0.05$). Notably, however, the degree of cvIKI or cvKvel did not differ between groups ($P_{FDR}<0.05$, *Figure 4E–F*). The mean performance tempo, loudness and the mean learned solution during learning did not differ significantly between groups, as in the main experiment ($P>0.05$). Thus, removal of the initial exploration phase led to the impairment of reward-based learning by the anxiety manipulation, and this effect was not associated with a change in the use of task-related variability or in general average performance parameters.

## Bayesian learning modeling reveals the effects of state anxiety on reward-based motor learning

To assess our hypotheses regarding the mechanisms underlying participants' performance during reward-based learning, we used several versions of a Bayesian learning model, which were based on the two-level hierarchical Gaussian filter for continuous input data (HGF; *Mathys et al., 2011*; *Mathys et al., 2014*). The HGF was introduced by *Mathys et al., 2011* to model how an agent infers a hidden state in the environment (a random variable), $x_1$, as well as its rate of change over time ($x_2$, environmental volatility). This corresponds to a perceptual model, which is further coupled with a response model to generate responses based on those inferred states. In the two-level HGF, beliefs about those hierarchically related hidden states ($x_1, x_2$) are continuous variables evolving as Gaussian random walks coupled through their variance. Their value ($x_i, i = 1, 2$) at trial $k$ will be normally

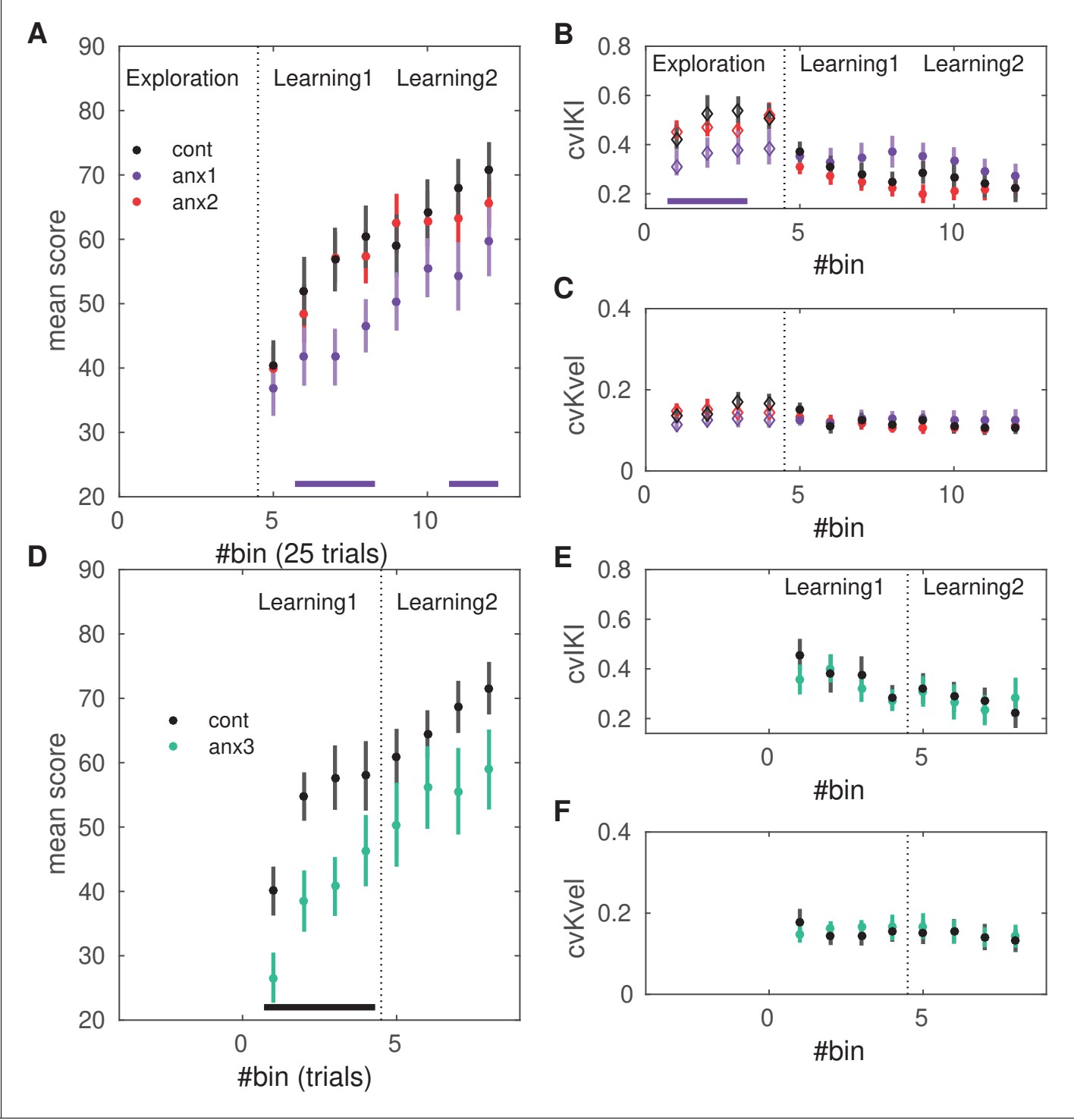

**Figure 4.** Effects of anxiety on behavioral variability and reward-based learning. The score was computed as a 0–100 normalized measure of proximity between the norm of the pattern of differences in inter-keystroke intervals performed in each trial and the target norm. All of the behavioral measures shown in this figure are averaged within bins of 25 trials. (**A**) Scores achieved by participants in the anx1 (N = 20), anx2 (N = 20), and control (N = 20) groups across bins 5:12 (trial range 101–300), corresponding to blocks 2 and 3 and the learning phase. Participants in anx1 achieved significantly lower scores than control participants ($P_{FDR}<0.05$, denoted by the bottom purple line). (**B**) Changes in across-trials cvIKI, revealing a significant drop in task-related exploration during the initial phase in anx1 relative to control participants ($P_{FDR}<0.05$). Anx2 participants did not differ from control participants. (**C**) Same as panel (**B**) but for the across-trials cvKvel. (**D–F**) Control experiment: effect of anxiety on variability and learning after removal of the initial exploration phase. Panels (**D–F**) are displayed in the same way as panels (**A–C**) for experimental (N = 13) and control (N = 13) groups. Significant

*Figure 4 continued on next page*

*Figure 4 continued*

between-group differences are denoted by the black bar at the bottom ($P_{FDR}<0.05, \Delta = 0.71, CI = [0.64, 0.78]$). (F) In anx3 participants (green), there was a significant drop in the mean scores during the first learning block relative to control participants ($P_{FDR}<0.05, \Delta = 0.77, CI = [0.68, 0.86]$). Bars around the mean show ± SEM.

The online version of this article includes the following figure supplement(s) for figure 4:

**Figure supplement 1.** Mean learned solution in each group.

distributed around their previous value at trial $k - 1$. Thus, the posterior distribution of beliefs about these states is fully determined by the sufficient statistics $\mu_i$ (mean) and $\sigma_i$ (variance, representing estimation uncertainty). Beliefs are updated given new sensory input via prediction errors (PEs). In some implementations of the HGF, the series of sensory inputs are replaced by a sequence of outcomes, such as reward value in a binary lottery (*Mathys et al., 2014*; *Diaconescu et al., 2017*) or electric shock delivery in a one-armed bandit task (*de Berker et al., 2016*). In these cases, similarly to the case of sensory input, an agent can learn the causes of the observed outcomes and thus the likelihood that a particular event will occur. In our study, the trial-by-trial input observed by the participants was the series of feedback scores (hereafter input refers to feedback scores). Crucial to the HGF is the weighting of the PEs by the ratio between the estimation uncertainty of the current level and the lower level, or the inverse ratio when using precision (inverse variance or uncertainty of a distribution). Further details are provided in the 'Materials and methods'.

Different implementations of the HGF have recently been used in combination with neuroimaging data to investigate how the brain processes different types of hierarchically-related prediction errors (PEs) within the framework of predictive coding (*Diaconescu et al., 2017*; *Weber et al., 2019*). The HGF can be fit to the behavioral data from each individual participant, thus providing dynamic trial-wise estimates of belief updates that depend on hierarchical PEs weighted by precision (precision-weighted PE or pwPE). In predictive coding models, precision is viewed as crucial for representing uncertainty and updating the posterior expectations about the hidden states (*Sedley et al., 2016*). In the HGF, time-varying pwPEs reflect how participants learn stimulus-outcome or response-outcome associations and their changes over time (*Mathys et al., 2014*; *Diaconescu et al., 2017*).

Here, we adapted the HGF to model participants' estimation of quantity $x_1$, which represented their beliefs about the expected reward (input score, normalized 0–1) for the current trial. Beliefs about $x_1$ on trial $k$ were thus determined by the expectation of reward $\mu_1^k$ (mean of the posterior distribution of $x_1$) and the uncertainty about this estimate (variance, $\sigma_1^k$). The model also estimated participants' beliefs about environmental volatility $x_2$, related to changes in the reward tendency and determined by $(\mu_2^k, \sigma_2^k)$ on trial $k$. The belief trajectories about the external states $x_1$ and $x_2$ generated by the model were further used to estimate the most likely response corresponding with those beliefs. A schematic illustrating the model structure and the belief trajectories is shown in *Figure 5*.

Assessment of the HGF for simulated responses revealed that the expectation of volatility (change in reward tendency) was higher in agents that modulated their performance to a greater extent across trials and thereby observed a broader range of feedback scores (see different examples for simulated performances in *Figure 5—figure supplement 1*).

We implemented eight versions of the HGF with different response models. The response model defines the mapping from the trajectories of perceptual beliefs onto the observed responses of each participant. We were interested in how HGF quantities on the previous trial explained changes in performance on the subsequent trial. To assess that relationship, we considered two scenarios characterized by the choice of a different performance measure in the response model. The performance measures used were: (1) the trialwise coefficient of variation of consecutive IKI values (cv across sequence positions; termed $\mathrm{cvIKI_{trial}}$ to dissociate it from the measure of across-trials variability, cvIKI); (2) the trialwise performance tempo (mean of IKI within the trial across sequence positions, termed $\mathrm{mIKI_{trial}}$; here we used the logarithm of this measure in milliseconds, $\log(\mathrm{mIKI_{trial}})$, as in *Marshall et al. (2016)*. Accordingly, we constructed two families of models describing the link between a participant's inferred perceptual quantities on the previous trial $k - 1$ and their *changes* from trial $k - 1$ to $k$ in one of those performance measures:

$$\Delta\mathrm{cvIKI_{trial}^k} = \mathrm{cvIKI_{trial}^k} - \mathrm{cvIKI_{trial}^{k-1}}$$

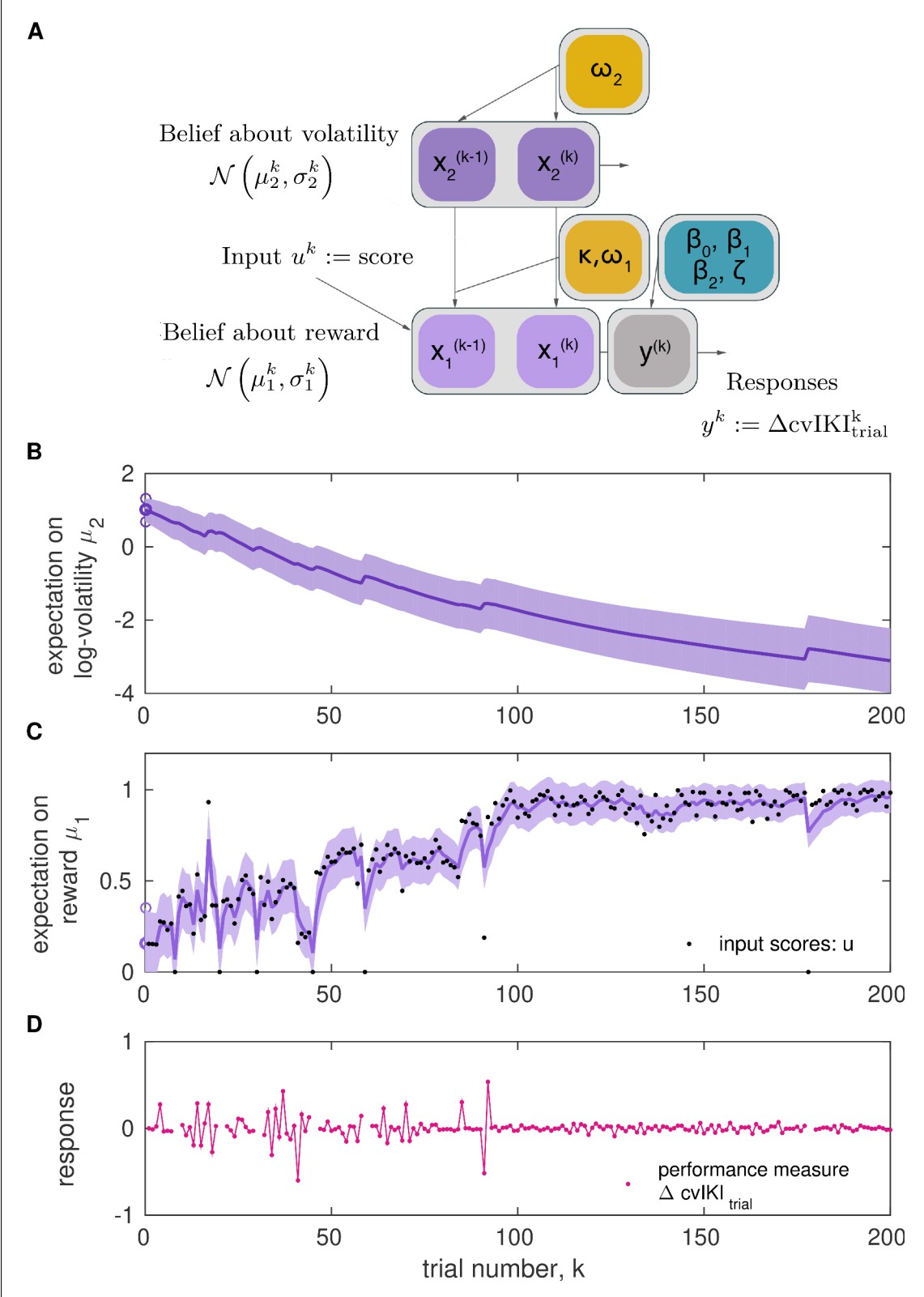

**Figure 5.** Two-level Hierarchical Gaussian Filter for continuous inputs. (A) Schematic of the two-level HGF, which models how an agent infers a hidden state in the environment (a random variable), $x_1$, as well as its rate of change over time ($x_2$, environmental volatility). Beliefs about those two hierarchically related hidden states ($x_1$, $x_2$) at trial $k$ are updated by the sensory input ($u^k$, observed feedback scores in our study) for that trial via prediction errors (PEs). The states $x_1$ and $x_2$ are continuous variables evolving as coupled Gaussian random walks, where the step size (variance) of the
*Figure 5 continued on next page*

*Figure 5 continued*

random walk depends on a set of parameters (shown in yellow boxes). The lowest level is coupled to the level above through the variance of the random walk: $x_1^k \sim \mathcal{N}(x_1^{k-1}, \exp(\kappa x_2^{k-1} + \omega_1))$. The posterior distribution of beliefs about these states is fully determined by the sufficient statistics $\mu_i$ (mean) and $\sigma_i$ (variance) for levels $i = 1, 2$. The equations describing how expectations ($\mu_i$) change from trial $k - 1$ to $k$ are *Equation 6* and *Equation 10*. The response model generates the most probable response, $y^k$, according to the current beliefs, and is modulated by the response model parameters $\beta_0, \beta_1, \beta_2, \zeta$. In the winning model, the response parameter was the change between trial $k - 1$ and $k$ in the degree of temporal variability across keystrokes: $y^k = \Delta\mathrm{cvIKI}_{\mathrm{trial}}^k$, normalized to range 0–1. (B, C) Example of belief trajectories (mean, variance) associated with the two levels of the HGF for continuous inputs. Panel (C) displays the expectation on the first level, $\mu_1^k$, which represents an individual's expectation (posterior mean) of the true reward values for the trial, $x_1^k$. Black dots represent the trial-wise input (feedback scores, $u^k$). Panel (B) shows the trial-by-trial beliefs about log-volatility $x_2^k$, determined by the expectation $\mu_2^k$ and associated variance. Shaded areas denote the variance or estimation uncertainty on that level. (D) Illustration of the performance measure used as response in the winning model, $y^k = \Delta\mathrm{cvIKI}_{\mathrm{trial}}^k$.

The online version of this article includes the following figure supplement(s) for figure 5:

**Figure supplement 1.** Trial-by-trial belief trajectories for simulated performances.
**Figure supplement 2.** Simulated trial-by-trial belief trajectories in an ideal learner.
**Figure supplement 3.** $\beta$ coefficients of the winning response model.
**Figure supplement 4.** Example in one control participant of the association between pwPEs and performance.
**Figure supplement 5.** Example in one anx1 participant of the association between pwPEs and performance.
**Figure supplement 6.** Grand-average trialwise residuals.

$$\Delta\log(\mathrm{mIKI}_{\mathrm{trial}})^k = \log(\mathrm{mIKI}_{\mathrm{trial}}^k) - \log(\mathrm{mIKI}_{\mathrm{trial}}^{k-1})$$

Variable $\mathrm{cvIKI}_{\mathrm{trial}}$ was chosen because it is tightly linked to the variable associated with reward: higher differences in IKI values between neighboring positions lead not only to a higher vector norm of IKI patterns but also to a higher coefficient of variation of IKI values in that trial (and indeed $\mathrm{cvIKI}_{\mathrm{trial}}$ was positively correlated with the feedback score across participants, nonparametric Spearman $\rho = 0.69, P < 10^{-5}$). Alternatively, we considered the scenario in which participants would speed or slow down their performance without altering the relationship between successive intervals. Therefore, we used a performance measure related to the mean tempo, mIKI. We did not choose a performance measure associated with keystroke velocity because our results in the previous sections demonstrate that participants did not consistently modulate cvKvel across trials—either because they realized that this parameter was non-task-related or because they were not able to substantially vary the loudness of the key press. Similarly to *Marshall et al. (2016)*, in each family of models we defined four types of response models to explain the performance measure as a linear function of relevant HGF perceptual parameters on the previous trial, such as the expectation of reward ($\mu_1$) or volatility ($\mu_2$) and the pwPEs on these estimates (labeled $\epsilon_1$ and $\epsilon_2$, respectively; see *Equation 14* and *Equation 15*). One example is illustrated here:

$$\Delta\mathrm{cvIKI}_{\mathrm{trial}}^k = \beta_0 + \beta_1\mu_1^{k-1} + \beta_2\epsilon_1^{k-1} + \zeta \tag{1}$$

where $\beta_0$ represents a constant value (intercept) and $\zeta$ is a Gaussian noise variable. Details on the alternative models are provided in the 'Materials and methods' section.

In each model, the feedback scores and the performance measure at each trial $k$ were used to update model parameters, and the log model-evidence was used to optimize the model fit (*Diaconescu et al., 2017*; *Soch and Allefeld, 2018*). More details on the modeling approach can be found in the 'Materials and methods' section and in *Figure 5*.

Between-group comparison focused on four variables, the mean trajectories of perceptual beliefs ($\mu_1$ and $\mu_2$, means of the posterior distributions for $x_1$ and $x_2$; *Figure 5*), and the uncertainty about those beliefs (variance of the posterior distributions, $\sigma_1$ and $\sigma_2$; note that the inverse variance is the precision, termed $\pi_1$ and $\pi_2$, corresponding with the confidence placed on those beliefs). As indicated above, volatility estimates are related to the rate of change in reward estimates, and accordingly we predicted a higher expectation of volatility $\mu_2$ for participants exhibiting more variation in $\mu_1$ values. In addition, the perceptual model parameters $\omega_1$ and $\omega_2$, which characterize the learning style of each participant (see *Figure 5—figure supplement 2*), and the parameters $\beta_0, \beta_1, \beta_2, \zeta$, characterizing the response model, were contrasted between groups.

Random Effects Bayesian Model Selection (BMS) was used to assess at the group level (N = 60) the different models of learning (*Stephan et al., 2009*); code freely available from the MACS toolbox, (*Soch and Allefeld, 2018*). First, the models were grouped into two families corresponding with each performance measure ($\Delta\text{cvIKI}_{\text{trial}}$ and $\Delta\log(\text{mIKI}_{\text{trial}})$). The log-family evidence (LFE) was calculated from the log-model evidence (LME). BMS then determined which family of models provided more evidence. In the winner family, additional BMS determined the final optimal model. BMS provided stronger evidence for the family of models defined for $\Delta\text{cvIKI}_{\text{trial}}$, with an exceedance probability of 1, and an expected frequency of 0.9353 (similar values in experimental and control groups). Next, among all four models in that family, the winning model (exceedance probability 1, model frequency 0.8614) explained the performance measure $\Delta\text{cvIKI}_{\text{trial}}$ as a linear function of the pwPE relating to reward, $\epsilon_1$, and volatility, $\epsilon_2$, on the previous trial:

$$\Delta\text{cvIKI}_{\text{trial}}^{\text{k}} = \beta_0 + \beta_1 \epsilon_1^{k-1} + \beta_2 \epsilon_2^{k-1} + \zeta \qquad (2)$$

The $\beta_0$ and $\beta_1$ coefficients were significantly different than zero in each experimental and control group ($P_{FDR}$<0.05, controlled for multiple comparisons arising from three group tests; *Figure 5—figure supplement 3*). On average, $\beta_0$ was positive, and $\beta_1$ was negative. By contrast, $\beta_2$ was positive in the control group yet negative in the anx1 and anx2 groups ($P_{FDR}$<0.05). Because pwPEs directly modulate the update in the expectation of beliefs, these findings imply that smaller pwPEs relating to reward on the previous trial (smaller update in the expectation of reward at $k-1$) were associated in all groups with increases in $\text{cvIKI}_{\text{trial}}^{\text{k}}$ for the next trial. On the other hand, a negative $\beta_1$ also indicates that larger pwPE for reward on the previous trial decreased changes in the performance variable on the following trial. In addition, exclusively in control participants, there was a positive association between larger pwPE relating to volatility at $k-1$ (greater update in the expectation on volatility on the last trial) and a follow-up increment in $\text{cvIKI}_{\text{trial}}^{\text{k}}$. In anx1 and anx2 participants, however, trials of larger pwPE driving updates on volatility were followed by reduced changes in trial-wise temporal variability. The results imply that a larger increase in the expectation of volatility on the previous trial promoted larger subsequent changes in the relevant performance variable in control participants (*Figure 5—figure supplement 4*), whereas in anx1 and anx2, it led to reductions in task-related behavioral changes (*Figure 5—figure supplement 5*).

The HGF and the winning response model provided a good fit to the behavioral data from each group, as shown in the examination of the residuals (*Figure 5—figure supplement 6*). Further, there were no systematic differences in the model fits across groups (trial-averaged residuals were compared between each experimental and control group with permutation tests; $P$>0.05 in both comparisons; $P = 0.1598$ for anx1 and control groups; $P = 0.5646$ for anx2 and control groups). The low mean residual values further indicate that the model captured the fluctuations in data well (trial-averaged residuals and SEM: $0.0004[0.00095]$ in controls; $-0.001[0.0013]$ in anx1; and $0.0003[0.0003]$ in anx2).

Using the winning model, we next evaluated between-group differences in the mean trajectories of perceptual beliefs and their uncertainty throughout learning (*Figure 6A–C*). Participants in the anx1 relative to the control group had a lower estimate of the mean tendency for $x_1$ ($P_{FDR}$<0.05, $\Delta = 0.75, CI = [0.59, 0.89]$). This indicates a lower expectation of reward in the current trial. Note that this outcome could be anticipated from the behavioral results shown in *Figure 4A*. The expectation on log-volatility was significantly smaller in anx1 than in control participants ($P_{FDR}$<0.05, $\Delta = 0.71, CI = [0.60, 0.81]$). This quantity was also partly reduced in the anx2 group relative to the control group ($P_{FDR}$<0.05, $\Delta = 0.69, CI = [0.53, 0.75]$). In addition, the uncertainty about environmental volatility, $\sigma_2$, was larger in the anx1 and anx2 participants when compared to control participants (control relative to anx1, $P_{FDR}$<0.05, $\Delta = 0.71, CI = [0.65, 0.89]$; control relative to anx2, $P_{FDR}$<0.05, $\Delta = 0.65, CI = [0.52, 0.86]$). Because larger estimation uncertainty on the current HGF level contributes toward larger steps in the update equations for that level (due to larger precision weights on the PEs, *Equation 5*), this last outcome suggests that anx1 and anx2 participants updated their estimates of environmental volatility with larger steps (albeit in a negative direction as indicated by the negative slope of the underlying trends in *Figure 6C*, reducing $\mu_2$). No differences between anx2 and control participants in the $\mu_1$ estimates were found. Neither did we obtain between-group differences in $\sigma_1$.

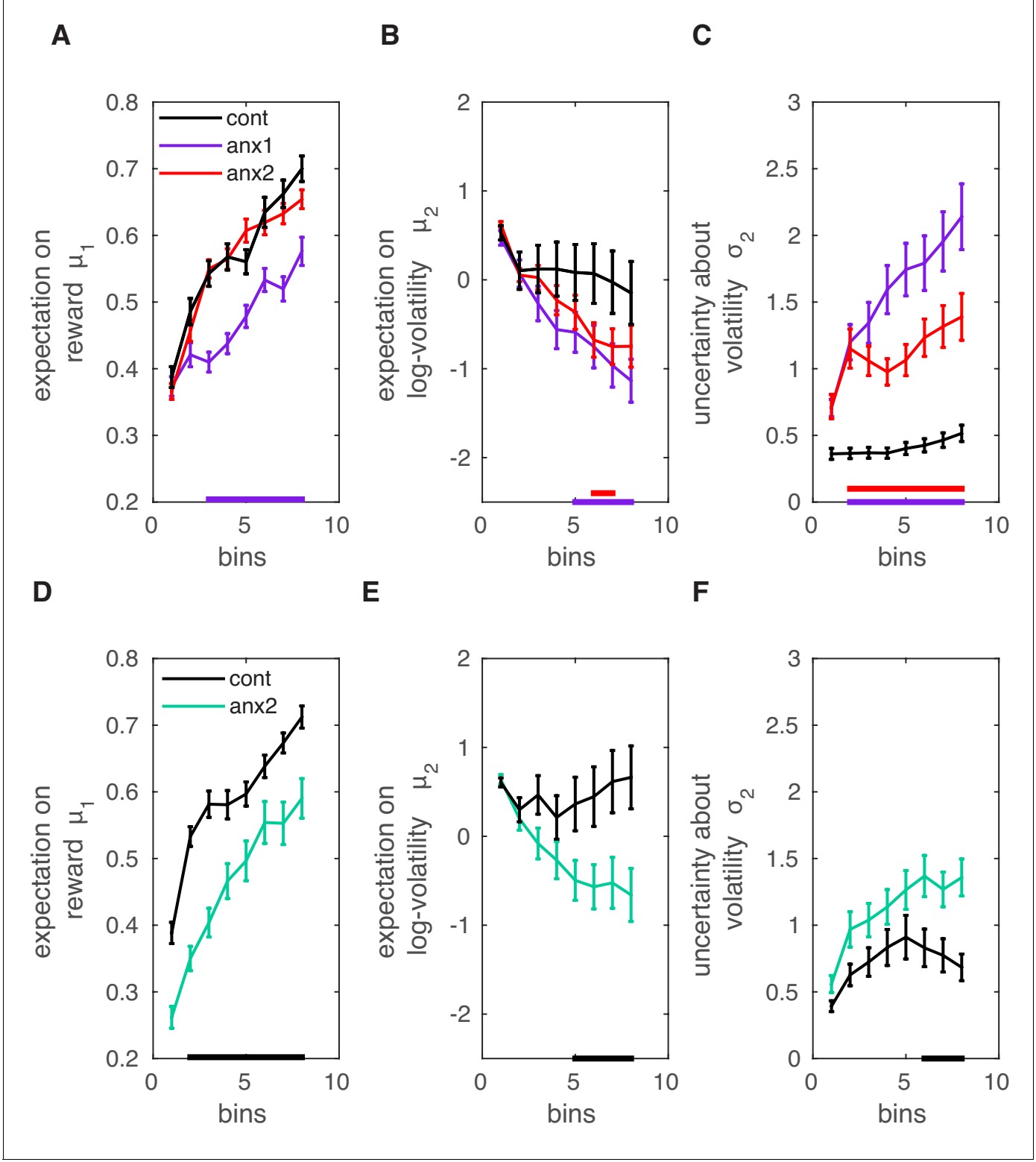

**Figure 6.** Computational modeling analysis. Data shown as mean and ± SEM. (**A**) In the main experiment, anx1 participants underestimated the tendency for $x_1$ (meaning their expectation on reward in the current trial was lower; $P_{FDR}$<0.05, $\Delta = 0.75$, $CI = [0.59, 0.89]$, purple bar at the bottom). (**B**) In addition, the expectation on environmental (phasic) log-volatility $\mu_2$ was significantly smaller in anx1 participants than in control participants ($P_{FDR}$<0.05, $\Delta = 0.71$, $CI = [0.60, 0.81]$). Similar results were obtained in the anx2 group as compared to the control group ($P_{FDR}$<0.05, $\Delta = 0.69$, $CI = [0.53, 0.75]$). (**C**) The uncertainty about environmental volatility was higher in anx1 and anx2 relative to control participants (anx1: $P_{FDR}$<0.05, $\Delta = 0.71$, $CI = [0.65, 0.89]$; anx2: $P_{FDR}$<0.05, $\Delta = 0.65$, $CI = [0.52, 0.86]$). Larger $\sigma_2$ in the anx1 and anx2 groups contributed to the larger

*Figure 6 continued on next page*

Figure 6 continued

update steps of the estimate $\mu_2$, shown in panel (B). (D–F) Same as panels (A–C) but in the separate control experiment. (D) The expectation on the reward tendency, $\mu_1$, was lower for anx3 participants relative to control participants ($P_{FDR}$<0.05, $\Delta = 0.80$, $CI = [0.68, 0.95]$, denoted by the black bar at the bottom). (E) Same as panel (B): anx3 participants had a reduced expectation of environmental volatility ($P_{FDR}$<0.05, $\Delta = 0.67$, $CI = [0.55, 0.76]$). (F) Anx3 participants were also more uncertain about their phasic volatility estimates relative to control participants ($P_{FDR}$<0.05, $\Delta = 0.65$, $CI = [0.51, 0.77]$). Thus, the anxiety manipulation in the control experiment biased participants to make larger updates of their expectation of phasic volatility.

The online version of this article includes the following figure supplement(s) for figure 6:

**Figure supplement 1.** Correlation between HGF volatility estimates and the variance of the distribution of feedback scores.

To understand why anx2 did not substantially differ from the control group in their expectation of reward yet had significantly lower volatility estimates (resembling those of the anx1 group), we looked more closely at *Figure 5—figure supplement 1*. This figure shows the HGF trajectories for perceptual beliefs and related quantities for a series of simulated responses. The results indicate that lower expectation of volatility can result from a smaller variance in the distribution of observed feedback scores, but also from a behavior characterized by smaller changes from trial to trial in the performance variable ($\Delta\text{cvIKI}_{\text{trial}}$). Accordingly, as a post-hoc analysis, we tested whether anx2 participants had smaller variance in the distribution of feedback scores when compared to control participants. This was the case (means [SEM]) were 0.064 [0.004] in control participants and 0.052 [0.003] in anx2, $P_{FDR}$<0.05. Anx1 participants also contributed to a similar effect (means [SEM] were 0.051 [0.002], $P_{FDR}$<0.05, smaller in anx1 than in the control group). Furthermore, anx2 participants had, on average, smaller $\Delta\text{cvIKI}_{\text{trial}}$ values than the control group (means [SEM] were 0.005 [0.0011] in controls and 0.0032 [0.0007] in anx2, $P_{FDR}$<0.05). The same results were obtained for the anx1 group (0.0013 [0.0009], $P_{FDR}$<0.05). Thus, anx2 participants achieved high scores, as did control participants, yet they observed a reduced set of scores. In addition, their task-related behavioral changes from trial to trial were more constrained. These smaller trial-to-trial behavioral changes in anx2 indicated a tendency to exploit their inferred optimal performance, leading to consistently high scores. This different strategy of successful performance ultimately accounted for the reduced estimation of environmental volatility in this group, and contrasted with the higher $\mu_2$ values obtained in control participants.

As an additional post-hoc analysis, and based on the insights obtained from *Figure 5—figure supplement 1*, we assessed in the total population whether volatility estimates were associated with the change in performance variable $\Delta\text{cvIKI}_{\text{trial}}$ or with the variance of the distribution of feedback scores. There was only a small yet significant non-parametric correlation between the HGF log-volatility estimates $\mu_2$ and the variance of the distribution of feedback scores across the 200 trials (Spearman $\rho = 0.3029$, $P$<0.0190, *Figure 6—figure supplement 1*). This outcome suggests that participants who encountered more variable feedback scores in association with their performance also had a higher expectation of volatility.

Along with the above-mentioned group effects on relevant expectation and uncertainty trajectories, we found significant differences between anx1 and control participants in the perceptual parameter $\omega_2$ (mean and SEM values: $-5.2$ [0.50] in controls, $-3.6$ [0.49] in anx1; $P_{FDR}$<0.05), but not in $\omega_1$ ($-4.8$ [0.72] in controls, $-4.8$ [0.52] in anx1; $P$>0.05). Parameter $\omega_2$ modulates the rate at which volatility changes, with higher values—as obtained in anx1 participants—leading to sharper and more pronounced steps of update in volatility (*Figure 5—figure supplement 2C*). This can also be described as a different learning style (*Weber et al., 2019*). Participants in the anx2 group did not differ from control participants in $\omega_1$ ($-4.1$ [0.47], $P$>0.05) or $\omega_2$ ($-4.0$ [0.74], $P$>0.05).

In the second experiment, in which anx3 participants demonstrated a pronounced drop in scores relative to those of control participants during the anxiety manipulation, we found that on the group level, the winning family of models was also the one associated with the performance parameter $\Delta\text{cvIKI}_{\text{trial}}$ (model frequency 0.8747 and exceedance probability of 1). Further, the best individual model within that family was the one that explained $\Delta\text{cvIKI}_{\text{trial}}^{k}$ as a function of $\epsilon_1^{k-1}$ and $\epsilon_2^{k-1}$ (exceedance probability of 1, and model frequency of 0.9051). Between-group comparisons in relevant model parameters demonstrated that, like anx1 participants in the main study, anx3 participants in this control experiment had a lower estimate of the mean tendency for $x_1$ ($P_{FDR}$<0.05, $\Delta = 0.80$, $CI = [0.68, 0.95]$; *Figure 6D–F*), and also had a reduced expectation on

environmental volatility ($P_{FDR}$<0.05, $\Delta = 0.67, CI = [0.55, 0.76]$). In addition, the anxiety manipulation led participants to have higher uncertainty about their phasic volatility estimates relative to control participants ($P_{FDR}$<0.05, $\Delta = 0.65, CI = [0.51, 0.77]$). No differences in the uncertainty about estimates for $x_1$ were found. The perceptual parameters $\omega_1$ and $\omega_2$ did not differ between groups ($P$>0.05; average values of $\omega_1$ and $\omega_2$ were $-4.9$ [SEM 0.32] and $-3.4$ [0.41] in the control group, and $-5.6$ [0.39] and $-4.4$ [0.44] in the anx3 group). Last, among all response parameters, $\beta_0, \beta_1, \beta_2, \zeta$, we found that exclusively $\beta_2$ (modulating the impact of $\epsilon_2^{k-1}$ on $\Delta \mathrm{cvIKI}_{\mathrm{trial}}^{k}$) was significantly different between groups (larger in control participants; $P = 0.041, \Delta = 0.68, CI = [0.55, 0.76]$). Converging with the main experiment, parameters $\beta_0$ and $\beta_1$ were on average positive and negative, respectively, in each group.

## Electrophysiological analysis

The analysis of the EEG signals focused on sensorimotor and prefrontal (anterior) beta oscillations and aimed to assess separately (i) tonic and (ii) phasic (or event-related) changes in spectral power and burst rate. Tonic changes in average beta activity would be an indication that the anxiety manipulation had an effect on the general modulation of underlying beta oscillatory properties. Complementing this analysis, assessment of the phasic changes in the measures of beta activity during trial performance and following feedback presentation allowed us to investigate the neural processes that drive reward-based motor learning and their alteration by anxiety. These analyses focused either on all channels (tonic changes) or on a subset of channels across contralateral sensorimotor cortices and anterior regions (phasic changes; see statistical analysis details in 'Materials and methods').

### State anxiety prolongs beta bursts and enhances beta power during exploration

We first looked at the general averaged properties of beta activity in this phase and their modulation by anxiety. The first measure we used was the standard averaged normalized power spectral density (PSD) of beta oscillations. Normalization of the raw PSD into decibels (dB) was carried out using the average PSD from the initial rest recordings (3 min) as reference. This analysis revealed a significantly higher beta-band power in a small contralateral sensorimotor region in anx1 participants relative to that in control participants during initial exploration ($P$<0.025, two-sided cluster-based permutation test, FWE-corrected; *Figure 7—figure supplement 1*). In anx2 participants, the beta power in this phase was not significantly different than that in controls (*Figure 7—figure supplement 1*, $P$>0.05). No significant between-group changes in PSD were found in lower (< 13Hz) or higher (> 30Hz) frequency ranges ($P$>0.05).

Next, we analyzed the between-group differences in the distribution of beta bursts extracted from the amplitude envelope of beta oscillations during initial exploration (*Figure 7A*). This analysis was motivated by evidence from recent studies suggesting that differences in the duration, rate, and onset of beta bursts could account for the association between beta power and movement in humans (*Little et al., 2018*; *Torrecillos et al., 2018*). To identify burst events and to assess the distribution of their duration, we applied an above-threshold detection method, which was adapted from previously described procedures (*Poil et al., 2008*; *Tinkhauser et al., 2017*; *Figure 7B*). In this analysis, we selected epochs locked to the GO signal at 0 s and extending up to 11 s. This interval included the STOP signal at 7 s and—in reward-based learning trials only—the feedback score at 9 s. Bursts extending for at least one cycle were selected. Using a double-logarithmic representation of the probability distribution of burst durations, we obtained a power law and extracted the (absolute) slope, $\tau$, also termed the 'life-time' exponent (*Poil et al., 2008*). Modeling work has revealed that a power law in the burst-duration distribution, reflecting the fact that the oscillation bursts have no characteristic scale, indicates that the underlying neural dynamics operate in a state close to criticality, and thus are beneficial for information processing (*Poil et al., 2008*; *Chialvo, 2010*).

Crucially, because the burst duration, rate, and slope provide complementary information, we focused our statistical analysis of the tonic beta burst properties on the slope or life-time exponent, $\tau$. A smaller slope corresponds to a burst distribution that is biased towards more frequent long bursts.

In all of our participants, the double-logarithmic representation of the distribution of burst duration followed a decaying power-law with slope values $\tau$ in the range 1.4–1.9. The life-time exponents

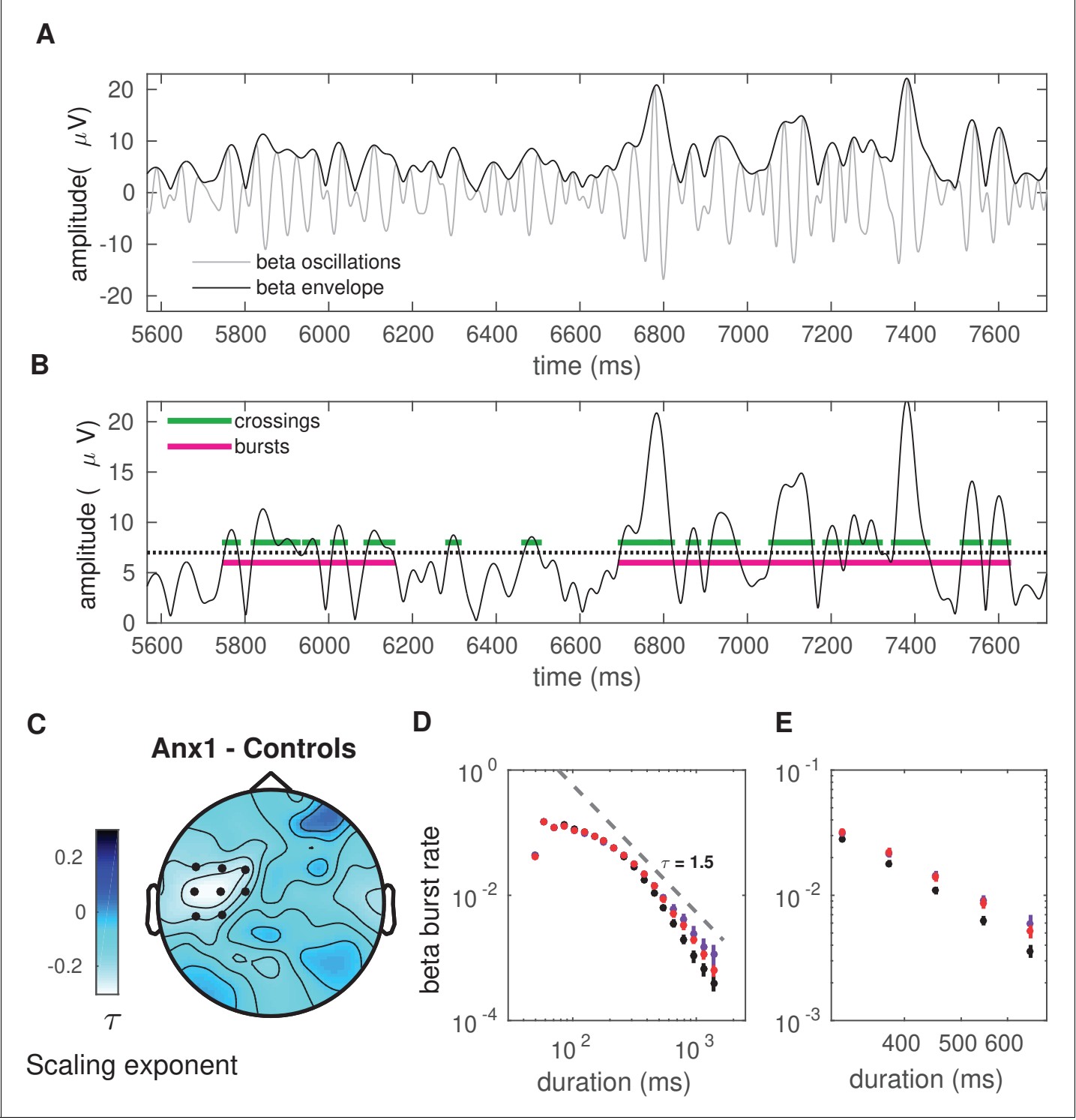

**Figure 7.** Anxiety during initial exploration prolongs the life-time of sensorimotor beta-band oscillation bursts. (**A**) Illustration of the amplitude of beta oscillations (gray line) and the amplitude envelope (black line) for one representative subject and channel. (**B**) Schematic overview of the threshold-crossing procedure used to detect beta oscillation bursts. A threshold of 75% of the beta-band amplitude envelope was selected and beta bursts extending for at least one cycle were accepted. Windows of above-threshold amplitude crossings detected in the beta-band amplitude envelope (black line) are denoted by the green lines, whereas the windows of the associated bursts are marked by the magenta lines. (**C**) Scalp topography for between-group changes in the scaling exponent $\tau$ during initial exploration. A significant negative cluster was found in an extended region of left sensorimotor electrodes, resulting from a smaller life-time exponent in anx1 than in control participants. (Black dots indicate significant electrodes, two-tailed cluster-based permutation test, $P_{FWE}<0.025$.) (**D**) Probability distribution of beta-band oscillation-burst life-times within the 50–2000 ms range for

*Figure 7 continued on next page*

*Figure 7 continued*

each group during initial exploration. The double-logarithmic representation reveals a power law within the fitted range (first duration bin excluded from the fit, as in *Poil et al., 2008*). For each power law, we extracted the slope, $\tau$, also termed the life-time exponent. The dashed line illustrates a power law with $\tau$ = 1.5. The smaller scaling exponent found in anx1 participants, as compared to control participants, was associated with long-tailed distributions of burst duration, reflecting the presence of frequent long bursts. Anx2 participants did not differ from control participants in the scaling exponent. Data are shown as mean and ± SEM in the electrodes pertaining to the significant cluster in panel (C). (E) Enlarged display of panel (D) showing that bursts of duration 500 ms or longer were more frequent in anx1 than in control participants.

The online version of this article includes the following figure supplement(s) for figure 7:

**Figure supplement 1.** Sensorimotor beta power is modulated by anxiety during initial exploration.

were smaller in the anx1 group than in the control group at left sensorimotor electrodes, corresponding with a long-tailed distribution (1.43 [0.30]; 1.70 [0.15]; $P_{FDR}<0.05, \Delta = 0.81, CI = [0.75, 0.87]$). No differences in slope values $\tau$ were found between anx2 and control participants. The smaller life-time exponents in anx1 in sensorimotor electrodes were also reflected in a longer mean burst duration: 182 (10) ms in the anx1 group, 153 (2) ms in control participants (166 [6] ms in anx2 participants). The differences in slope in the distribution of burst duration in anx1 reflected the more frequent presence of long bursts ( >500 ms) and the less frequent brief bursts in this group relative to control participants (*Figure 7D–E*).

We next turned to our main goal and asked whether there were between-group differences in the beta oscillatory properties at specific periods throughout the initial exploration trials, above and beyond the general block-averaged changes reported above. The results in *Figure 4* establish that state anxiety during the initial exploration phase reduced task-related motor variability, but also subsequently led to impaired reward-based learning. We therefore sought to assess whether the anxiety-related reduction in motor variability during exploration was associated with altered dynamics in beta-band oscillatory activity at specific time intervals during trial performance.

In anx1 participants, the mean beta power increased after completion of the sequence performance and further following the STOP signal, and these changes were significantly more pronounced than in control participants ($P_{FDR}<0.05, \Delta = 0.72, CI = [0.63, 0.80]$; *Figure 8A*). This significant effect was localized to contralateral sensorimotor and right prefrontal channels. As a post-hoc analysis, the time course of the burst rate was assessed separately in beta bursts of shorter (<300 ms) and longer (>500 ms) duration, following the results from *Figure 7* showing a pronounced dissociation between longer and brief bursts in the experimental and control groups. In addition, this split was motivated by previous studies linking longer beta bursts to detrimental performance (e.g. beta bursts longer than 500 ms in the basal ganglia of Parkinson's disease patients are associated with worse motor symptoms; *Tinkhauser et al., 2017*).

The rate of long oscillation bursts displayed a similar time course and topography to those of the power analysis, with an increased burst rate after movement termination and after the STOP signal in anx1 participants relative to control participants ($P_{FDR}<0.05, \Delta = 0.69, CI = [0.61, 0.78]$; *Figure 8B*). By contrast, brief burst events were less frequent in anx1 than in control participants, albeit exclusively during performance ($P_{FDR}<0.05, \Delta = 0.74, CI = [0.65, 0.82]$; *Figure 8C*). No significant effects were found when comparing any of these measures between anx2 and control participants.

Additional post-hoc control analyses were carried out to dissociate the separate effects of anxiety and motor performance on the time course of the beta-band oscillation properties during initial exploration. These analyses demonstrated that, when controlling for changes in motor variability, anxiety alone could explain the findings of larger post-movement beta-band PSD and rate of longer bursts, while also explaining the reduced rate of brief bursts during performance (*Figure 8—figure supplement 1*). Similar outcomes were found when controlling for changes in the mean total duration of the sequence (*Figure 8—figure supplement 2*), the variability of the sequence length (the coefficient of variation of sequence duration; *Figure 8—figure supplement 3*), and mean keystroke velocity (*Figure 8—figure supplement 4*).

Motor variability did also partially modulate the beta power and burst measures, after excluding anxious participants. This effect, however, had a small effect size and was limited to contralateral sensorimotor electrodes (*Figure 8—figure supplement 5*). In a last post-hoc analysis, we found that the average beta power following the STOP signal in those same significant sensorimotor electrodes was negatively correlated with the across-trials temporal variability, such that participants with a

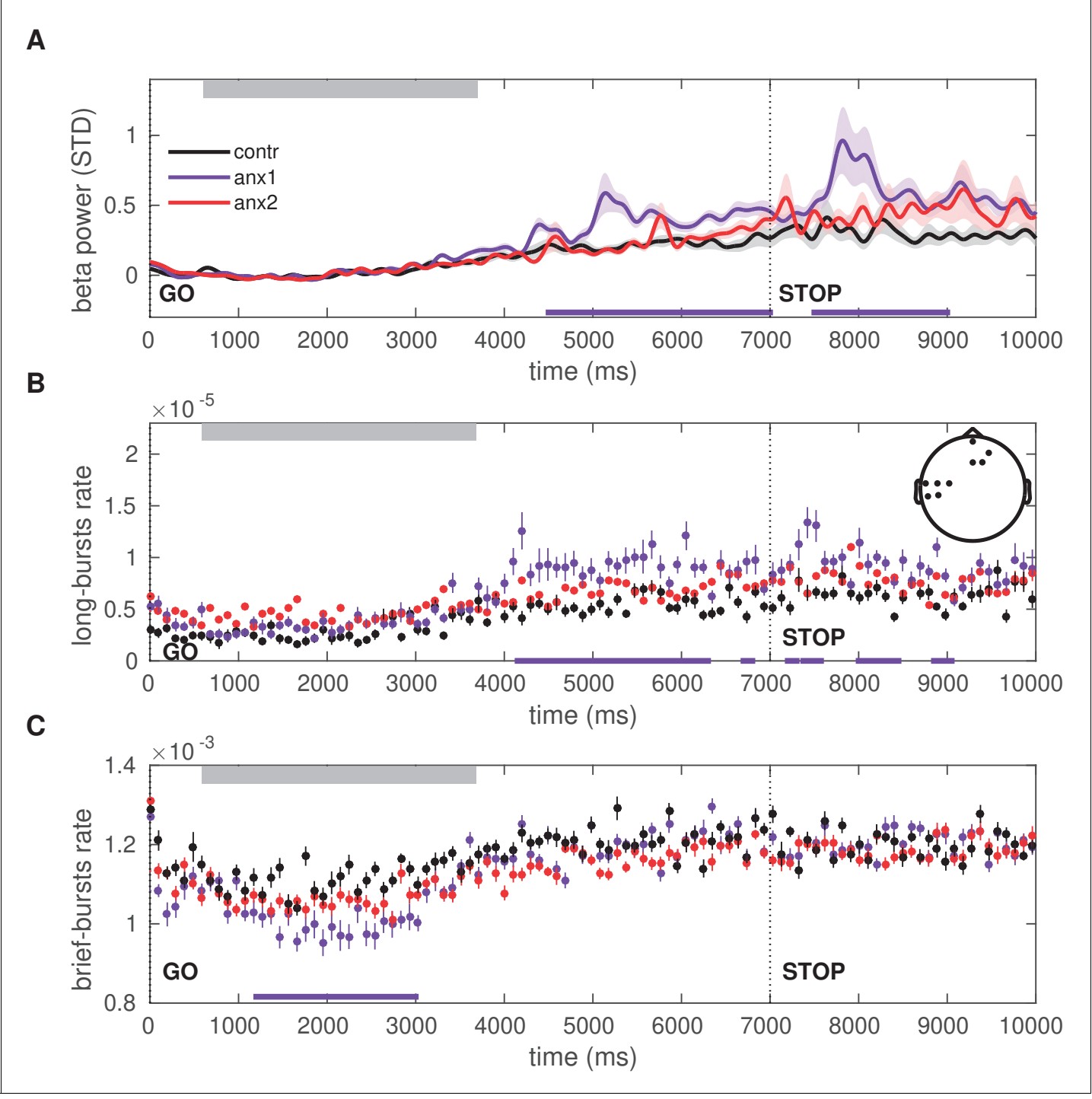

**Figure 8.** Time course of the beta power and burst rate during trials in the exploration block. (A) The time representation of the beta power throughout trial performance shows two distinct time windows of increased power in participants affected by the anxiety manipulation: following sequence performance and after the STOP signal ($P_{FDR}<0.05, \Delta = 0.72, CI = [0.63, 0.80]$; black bars at the bottom indicate the windows of significant differences). Shaded areas indicate the SEM around the mean. Performance of sequence1 was completed on average between 680 (50) and 3680 (100) ms, denoted by the gray rectangle at the top. The STOP signal was displayed at 7000 ms after the GO signal, and the trial ended at 9000 ms. (B) The rate of oscillation bursts of longer duration (>500 ms) exhibited a similar temporal pattern, with increased burst rate in anx1 participants following movement and the STOP signal ($P_{FDR}<0.05, \Delta = 0.69, CI = [0.61, 0.78]$). (C) In contrast to the rate of long bursts, the rate of brief oscillation bursts was reduced in anx1 relative to control participants, albeit during performance ($P_{FDR}<0.05, \Delta = 0.74, CI = [0.65, 0.82]$). All averaged values in panels (A–C) are estimated across the significant sensorimotor and prefrontal electrodes shown in the inset in panel (B).

*Figure 8 continued on next page*

*Figure 8 continued*

The online version of this article includes the following figure supplement(s) for figure 8:

**Figure supplement 1.** Post-movement increases in the beta-band amplitude and burst rate can be explained by state anxiety after matching participants on temporal variability.

**Figure supplement 2.** Post-movement increases in the beta-band amplitude and burst rate can be explained by state anxiety after matching participants on the sequence duration.

**Figure supplement 3.** Post-movement increases in the beta-band amplitude and burst rate can be explained by state anxiety after matching participants on the variability of the total sequence duration.

**Figure supplement 4.** Post-movement increases in the beta-band amplitude and burst rate can be explained by state anxiety after matching participants on the mean keystroke velocity.

**Figure supplement 5.** Changes in motor variability without concurrent changes in state anxiety only partially account for the observed alterations in post-movement beta amplitude and burst rate.

**Figure supplement 6.** Correlation between average beta power and the degree of task-related behavioral variability across trials during the exploration phase.

---

smaller increase in sensorimotor beta power after the STOP signal had a larger expression of task-related variability in this initial block (Spearman $\rho = -0.4397, P = 0.0001$; *Figure 8—figure supplement 6*).

## Reduced beta power and reduced presence of long beta bursts during feedback processing promotes the update of beliefs about reward

During learning, the general average level of PSD did not differ between groups ($P_{FDR}$<0.05; *Figure 9—figure supplement 1A–C*), neither was there a significant between-group difference in the scaling exponent of the distribution of beta-band oscillation bursts ($P_{FDR}$>0.05, *Figure 9—figure supplement 1D–E*; mean $\tau$ across contralateral and prefrontal electrodes: 1.78 [0.06] in control, 1.61 [0.10] in anx1, 1.70 [0.06] in anx2 group). The lack of significant between-group differences in these measures indicated that during reward-based motor learning, there were no pronounced tonic changes in average beta activity induced by the previous (anx1) or concurrent (anx2) anxiety manipulation.

*Figure 4* had established that motor variability (or other motor output variables) did not differ in learning blocks between experimental and control groups, and therefore could not explain the significant and pronounced drop in scores in anx1 participants. Accordingly, we next aimed to assess whether alterations in the beta-band measures over time during trial performance or in feedback processing could account for that effect. In the anx1 group, the mean beta power increased towards the end of the sequence performance more prominently than in control participants, and this effect was significant in sensorimotor and prefrontal channels ($P_{FDR}$<0.05, $\Delta = 0.67, CI = [0.56, 0.78]$; *Figure 9A*). A significant increase with similar topography and latency was observed in the anx2 group relative to control participants ($P_{FDR}$<0.05, $\Delta = 0.61, CI = [0.56, 0.67]$). An additional and particularly pronounced enhancement in beta power appeared in anx1 and anx2 participants within 400—1600 ms following presentation of the feedback score. This post-feedback beta increase was significantly larger in anx1 than in the control group ($P_{FDR}$<0.05, $\Delta = 0.65, CI = [0.55, 0.75]$; no significant effect in anx2, $P$>0.05).

Further, we found that the time course of the beta burst rate exhibited a significant increase in anx1 participants relative to that in control participants within 400–1600 ms following feedback presentation, similar to the power results (*Figure 9B*; $P_{FDR}$<0.05, $\Delta = 0.82, CI = [0.70, 0.91]$). The rate of brief oscillation bursts was, by contrast, smaller in anx1 than in control participants, albeit exclusively during performance and not during feedback processing (*Figure 9C*; $P_{FDR}$<0.05, $\Delta = 0.70, CI = [0.56, 0.84]$). The significant effects in anx1 participants were observed in left sensorimotor and right prefrontal electrodes. There were no significant differences between anx2 and control groups in the rate of brief or long bursts throughout the trial ($P$>0.05).

To rule out the possibility that the feedback-related changes in beta activity were accounted for by concurrent movement-related artifacts (e.g. larger artifacts in anx1 than in control participants), we performed a control analysis of higher gamma band activity, which has been consistently associated with muscle artifacts in previous studies (*Muthukumaraswamy, 2013*). This control

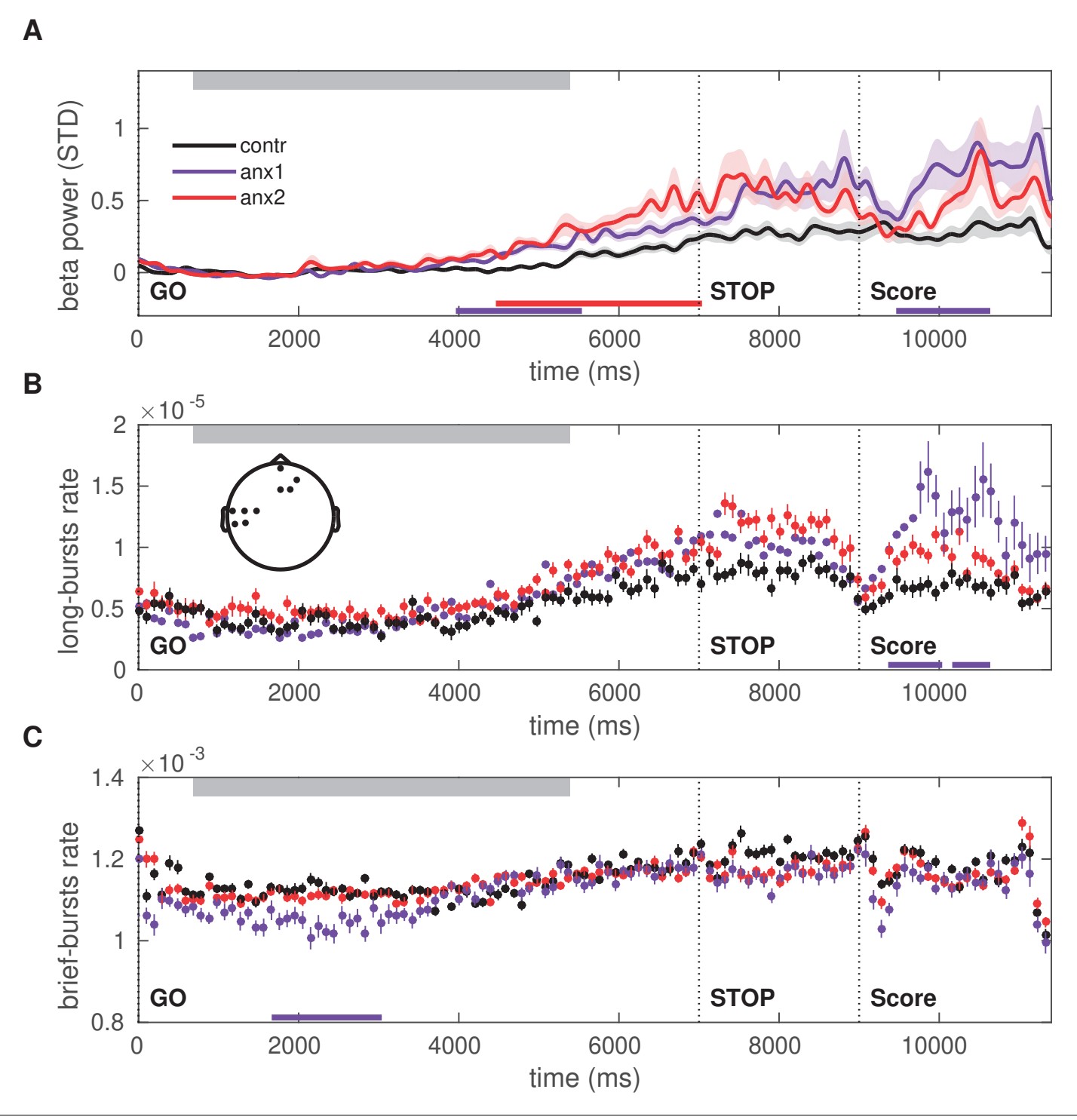

**Figure 9.** Time course of the beta power and burst rate throughout trial performance and following reward feedback. (A) Time course of the feedback-locked beta power during sequence performance in the learning blocks, shown separately for anx1, anx2 and control groups. Average across sensorimotor and prefrontal electrode regions as in panel (B). Shaded areas indicate the SEM around the mean. Participants completed sequence2 on average between 720 (30) and 5350 (100) ms, denoted by the top gray box. The STOP signal was displayed 7000 ms after the GO signal, and was followed at 9000 ms by the feedback score. This representation shows two distinct time windows of significant differences in beta activity between the anx1 and control groups: at the end of the sequence performance and subsequently following feedback presentation ($P_{FDR} < 0.05$, $\Delta = 0.65$, $CI = [0.55, 0.75]$, respectively, denoted by the purple bar at the bottom). Anx2 participants also exhibited an enhanced beta power towards the end of the sequence performance ($P_{FDR} < 0.05$, $\Delta = 0.61$, $CI = [0.56, 0.67]$). (B) Time course of the rate of longer (>500 ms) oscillation bursts

*Figure 9 continued on next page*

Figure 9 continued

during sequence performance in the learning blocks. Anx1 participants exhibited a prominent rise in the burst rate 400–1600 ms following the feedback score, which was significantly larger than the rate in control participants ($P_{FDR}$<0.05, $\Delta = 0.82$, $CI = [0.70, 0.91]$). Data display the mean and ± SEM. The topographic map indicates the electrodes of significant effects for panels (**A–C**) ($P_{FDR}$<0.05). (**C**) Same as panel (**B**) but showing the rate of shorter beta bursts (<300 ms) during sequence performance in the learning blocks. Between-group comparisons demonstrated a significant drop in the rate of brief oscillation bursts in anx1 participants relative to control participants at the beginning of the performance ($P_{FDR}$<0.05, $\Delta = 0.70$, $CI = [0.56, 0.84]$), but not after the presentation of the feedback score. In all panels, the traces of the mean power and burst rates were displayed after averaging across the significant sensorimotor and prefrontal electrodes shown in the inset in panel (**B**).

The online version of this article includes the following figure supplement(s) for figure 9:

**Figure supplement 1.** Beta power spectral density and burst rate during reward-based learning.

**Figure supplement 2.** Higher gamma band activity analysis rules out an explanation in which muscle artifacts influence feedback-related changes in power.

analysis found no evidence for movement artifacts affecting differently anx1 or control groups (*Figure 9—figure supplement 2*).

Having established that, relative to control participants, anx1 participants exhibited a phasic increase in beta activity and an increase in the rate of long bursts 400–1600 ms following feedback presentation, we next investigated whether these post-feedback beta changes could account for the altered reward and volatility estimates in the anx1 group (*Figure 6*). In the proposed predictive coding framework, superficial pyramidal cells encode PEs weighted by precision (precision-weighed PEs or pwPEs), and these are also the signals that are thought to dominate the EEG (*Friston and Kiebel, 2009*). A dissociation between high (gamma >30 Hz) and low (beta) frequency of oscillations has been proposed to correspond with the encoding of bottom-up PEs and top-down predictions, respectively (*Arnal and Giraud, 2012*). Operationally, however, beta oscillations have been associated with the change in predictions or expectations ($\Delta \mu_i$) rather than with predictions themselves (*Sedley et al., 2016*). In the HGF, the update equations for $\mu_1$ and $\mu_2$ are determined exclusively by the pwPE term in that level, such that the change in predictions, $\Delta \mu_i$, is equal to pwPE (see *Equation 14* and *Equation 15*). Accordingly, we assessed whether the trialwise feedback-locked beta power or burst rate represented the magnitude of pwPEs in that trial that serve to update expectations on reward ($\mu_1$) and environmental volatility ($\mu_2$).

For each participant, we assessed simultaneously the effect of $\epsilon_1$ and $\epsilon_2$ on the trial-by-trial feedback-locked beta activity by running a multiple linear regression. These two regressors were not linearly correlated with each other (Pearson r coefficient in the total population was 0.1 on average [median = 0.1], and individual correlation p-values were $P$>0.05 in 80% of all participants). For the multiple linear regression analysis, trial-wise estimates of beta power (or burst rate) were averaged within 400–1600 ms following feedback presentation and across the sensorimotor and prefrontal electrodes where the post-feedback group effects were found (*Figure 9*). The results indicate that $\epsilon_1$ had a significant negative effect on the measure of beta power (*Figure 10*; similarly for the rate of long bursts, see *Figure 10—figure supplement 1*), as $\beta_1$ was significantly smaller than zero in each group ($P_{FDR}$<0.05). In addition, the $\beta_1$ coefficient was decreased in anx1 relative to the control group ($P_{FDR}$<0.05, $\Delta = 0.72$, $CI = [0.57, 0.81]$; there were no differences between anx2 and control group). Thus, a reduction in $\epsilon_1$ contributed to an increase in post-feedback beta power and the rate of long beta bursts. The intercept also significantly differed between anx1 and control groups, with a larger coefficient representing a larger level of post-feedback beta power as found in anx1 ($P_{FDR}$<0.05, $\Delta = 0.69$, $CI = [0.55, 0.75]$; no differences were obtained in anx2 relative to control participants). The $\beta_2$ coefficient modulating the contribution of $\epsilon_2$ to beta activity was not different than 0 in any group ($P$>0.05). Accordingly, these results provide evidence for a pattern of neural oscillatory modulation that is associated with the updating of beliefs about reward. Furthermore, they link enhanced post-feedback beta activity—as found in anx1—to reduced pwPE about reward.

## Discussion

The results revealed several interrelated mechanisms through which state anxiety impairs reward-based motor learning. First, state anxiety reduced motor variability during an initial exploration phase. This was associated with limited improvement in scores during subsequent learning.

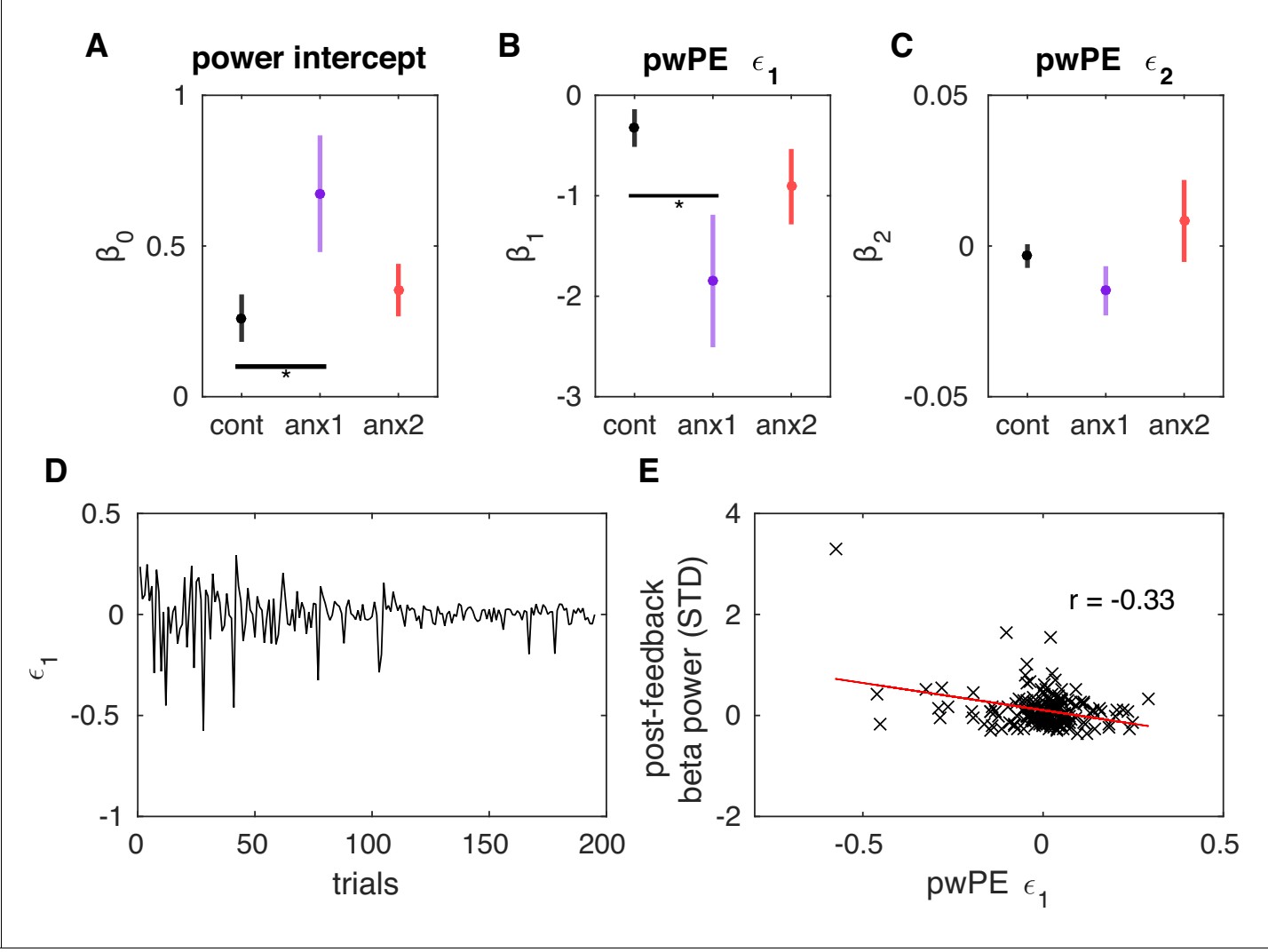

**Figure 10.** Post-feedback increases in beta power represent attenuated precision-weighted prediction errors about reward estimates. (A–C) Mean (and SEM) values of the $\beta$ coefficients that explain the post-feedback beta power as a linear function of a constant value (beta power) (A), the precision-weighted prediction errors driving updates in the expectation of reward (pwPE, $\epsilon_1$) (B), and pwPE driving updates in the expectation of volatility ($\epsilon_2$) (C). The measure of beta power used here was the average within 400–1600 ms following feedback presentation and across sensorimotor and prefrontal electrodes ,as shown in *Figure 9*. The $\beta$ values are plotted separately for each control and experimental group. The $\beta_0$ and $\beta_1$ regression coefficients were significantly different from 0 in all groups ($P_{FDR}<0.05$). In addition, $\beta_0$ was larger in the anx1 group relative to the control group ($P_{FDR}<0.05$, denoted by the horizontal black line and the asterisk). In anx1 relative to control participants, we found that $\beta_1$ was negative and significantly smaller in anx1 participants ($P_{FDR}<0.05$). Thus, a reduction in $\epsilon_1$ contributed to an increase in post-feedback beta power. The multiple regression analysis did not support a significant contribution of the second regressor, pwPE relating to volatility, to explaining the changes in beta power (see main text, also $\beta_2$ on average did not differ from 0 in any group of participants, $P>0.05$). (D) Illustration of the trajectories of pwPE $\epsilon_1$ in one representative anx1 subject. (E) The linear regression between the trial-wise beta power and pwPE $\epsilon_1$ for the same representative subject.

The online version of this article includes the following figure supplement(s) for figure 10:

**Figure supplement 1.** The rate of long beta bursts following feedback is modulated by the magnitude of precision-weighted prediction errors relating to reward.

**Figure supplement 2.** Topographic map illustrating the EEG channels used for the feedback-locked oscillatory analysis.

Second, the smaller change in the expectation of reward throughout time led to a decrease in the expectation of volatility. Along with those results, we observed an overestimation of uncertainty about volatility due to state anxiety, which promoted the drop in the volatility estimate. Additional computational results demonstrated that larger precision-weighted prediction errors relating to reward and volatility had the effect of constraining the trial-to-trial behavioral adaptations in state

anxiety. This contrasted with the findings for volatility in control participants, where larger pwPE relating to this quantity promoted behavioral exploration.

On the neural level, anxiety during initial exploration was associated with elevated sensorimotor beta power and a distribution of bursts of sensorimotor beta oscillations with a longer tail (smaller scaling exponent). The latter result indicated a more frequent presence of longer bursts, resembling recent findings of abnormal burst duration in movement disorders (*Tinkhauser et al., 2017*). The anxiety-induced higher rate of long burst events and higher beta power during initial exploration also manifested in prefrontal electrodes and extended to the following learning phase, where phasic trial-by-trial feedback-locked increases in these measures accounted for the attenuated updating of expectation on reward. These results provide the first evidence that state anxiety induces changes in the distribution of sensorimotor and prefrontal beta bursts, as well as in beta power, which may account for the observed deficits in the update of predictions during reward-based motor learning.

Evidence from our main experiment suggested that the finding of anxiety-related reduced motor variability during exploration was associated with the outcome of subsequently impaired learning from reward. These results validate previous accounts on the relationship between motor variability and Bayesian inference (*Wu et al., 2014*). In addition, the association between larger initial task-related variability and higher scores during the following learning phase extends results on the facili-atory effect of exploration on motor learning, at least in tasks that require learning from reinforce-ment (*Wu et al., 2014*; *Pekny et al., 2015*; *Dhawale et al., 2017*; see also critical view in *He et al., 2016*).

Crucially, state anxiety constrained the total amount of task-related variability only when induced during the initial exploration phase. The lack of between-group differences in cvIKI during learning in both experiments suggests that this measure could not account for the anxiety-related deficits in reward-based learning. Our Bayesian learning model provided additional insight on this aspect. The modelling results suggested that state anxiety can impair learning from reward not only by influencing the posterior distributions of beliefs (expectations and uncertainty) but also by altering how pwPE relating to those beliefs affect behavioral variability. The response model consistently demonstrated in experimental and control groups that smaller pwPEs driving reward updates on the previous trial (leading to decreased expectation of reward) were followed by an increase in task-related motor variability (higher exploration). On the other hand, trials of larger pwPE relating to reward were followed by reduced task-related behavioral changes. By contrast, the effect of pwPE on volatility differed substantially in control and anxiety groups. Although large pwPEs on volatility promoted subsequent larger task-related behavioral changes in control participants, they constrained behavioral exploration in the anx1 and anx2 groups.

Accordingly, state anxiety facilitated the use of task-related variability during reward-based learn-ing only in trials following smaller pwPE reducing volatility estimates. This led participants who were affected by the prior or concurrent state anxiety manipulation to underestimate environ-mental volatility. Thus, they had the expectation that reward estimates are more stable throughout time. Anx1 and anx2 participants also had larger uncertainty about volatility. This implies that they were less confident about their volatility estimate, and allowed for a greater influence of new infor-mation in updating this quantity. This finding is additionally reinforced in anx1 by the result of a larger $\omega_2$, reflecting a different learning style that is characterized by sharper and more pronounced steps of update in $\mu_2$. The results align well with recent computational work in decision-making tasks, showing that high trait anxiety leads to alterations in uncertainty estimates and adaptation to the changing statistical properties of the environment (*Browning et al., 2015*; *Huang et al., 2017*; *Pulcu and Browning, 2019*).

Notwithstanding the similarities in the anx1 and anx2 groups concerning the expectation of vola-tility and associated uncertainty, the fact that anx2 participants achieved high scores in the task and were not impaired in learning requires further clarification. Our post-hoc analyses revealed that the drop in $\mu_2$ in anx2 could be accounted for by the narrower distribution of scores encountered by this group. In addition, these participants introduced smaller trial-to-trial changes in temporal variability when compared to control participants. Thus, anx2 participants had a tendency to exploit the cur-rent motor program more than control participants, suggesting a more conservative approach to success. Anx1 participants also introduced smaller trial-to-trial changes in trial-wise temporal variability ($\mathrm{cvIKI}_{\mathrm{trial}}$), yet their behavioral changes had a slower benefit on reward. In both groups, however, the more pronounced tendency to exploit the current motor program was associated with

alterations in how pwPE relating to volatility influenced behavioral changes. Overall, our findings provide the first evidence that computational mechanisms similar to those described for trait anxiety and decision-making underlie the effect of temporary anxious states on motor learning. This might be particularly the case in the context of learning from rewards, such as feedback about success or failure, which is considered one of the fundamental processes through which motor learning is accomplished (*Wolpert et al., 2011*).

Previous studies manipulating psychological stress and anxiety to assess motor learning showed both a deleterious and a faciliatory effect (*Hordacre et al., 2016*; *Vine et al., 2013*; *Bellomo et al., 2018*). Differences in experimental tasks, which often assess motor learning during or after high-stress situations but not during anxiety induction in anticipation of a stressor, could account for the previous mixed results. Here, we adhered to the neurobiological definition of anxiety as a psychological and physiological response to an upcoming diffuse and unpredictable threat (*Grupe and Nitschke, 2013*; *Bishop, 2007*). Accordingly, anxiety was induced using the threat of an upcoming public speaking task (*Feldman et al., 2004*; *Lang et al., 2015*), and was associated with a drop in the HRV and an increase in state anxiety scores during the targeted blocks. Although the average state anxiety scores were not particularly high, they were significantly higher during the targeted phases than during the initial resting state phase. Future studies should use more impactful stressors to study the effect of the full spectrum of state (and trait) anxiety on motor learning (*Bellomo et al., 2018*).

What is the relationship between the expression of motor variability and state anxiety? As hypothesized, state anxiety during initial exploration reduced the use of variability across trials. This converges with recent evidence demonstrating that anxiety leads to ritualistic behavior (repetition, redundancy, and rigidity of movements) that allow the subject to regain a sense of control (*Lang et al., 2015*). The outcome also aligns well with animal studies in which evidence shows a reduction in motor exploration when the stakes are high (high-reward situations, social context; *Kao et al., 2008*; *Dhawale et al., 2017*; *Woolley et al., 2014*). These interpretations, however, seem to stand in contrast with our findings in anx2 participants, who were affected by the anxiety manipulation during learning but with no significant effect on the total degree of motor variability expressed during this phase. Similar results were obtained in the second experiment, as anx3 and control participants did not differ in the amount of across-trials variability expressed during learning. Bayesian computational modelling clarified these findings demonstrating that anx2 participants used increased exploitation of their current motor program. Also, their trial-to-trial changes in temporal variability were smaller than those in the control group, particularly following large pwPEs that increased the expectation on volatility. This outcome was also found in both anx1 and anx3 participants in the second experiment. Thus, anxiety consistently constrained dynamic trial-to-trial changes in temporal variability—with these changes negatively influenced by pwPEs on volatility. Notably, however, the strategy in anx2 participants of more extensively exploiting the inferred rewarded solution (relative to control participants) was successful, and therefore differs from the learning impairment exhibited by anx1 participants. In the second experiment, removing the initial exploration phase led to impaired reward-based learning in anx3 participants. This group also tended to explore less than controls at the trial level as a function of changes in volatility pwPEs. Thus, the combined evidence suggests that normal use of initial variability in anx2 participants protected their performance from the subsequent impact of the anxiety manipulation. Initial use of variability in anx2 might promote faster learning of the mapping between actions and their asociated outcome, contributing to successful goal-directed exploitation. We interpret these results to indicate that initial unconstrained exploration is important for later *subsequent* successful motor learning.

Some considerations should be taken into account. Task-related motor variability might be pivotal for learning from reinforcement or reward signals (*Sutton and Barto, 1998*; *Dhawale et al., 2017*; *Wu et al., 2014*), whereas in other contexts, such as during motor adaptation, the evidence is conflicting (*He et al., 2016*; *Singh et al., 2016*). An additional consideration is that greater levels of motor variability could reflect both an intentional pursuit of an explorative regime and an unintentional higher level of motor noise, in the latter case similar to that observed in previous work (*Wu et al., 2014*; *Pekny et al., 2015*). A recent study established that motor learning is improved by the use of intended exploration, not motor noise (*Chen et al., 2017*). Our paradigm cannot dissociate intended and unintended exploration. This limitation will be addressed in future work by

using a separate initial phase with regular performance to assess motor noise as a measure of unintended exploration.

Another consideration is that our use of an initial exploration phase that did not provide reinforcement or feedback signals was motivated by the work of *Wu et al. (2014)*, which demonstrated a correlation between initial variability (no feedback) and learning curve steepness in a subsequent reward-based learning phase—a relationship previously observed in the zebra finch (*Kao et al., 2005*; *Olveczky et al., 2005*; *Ölveczky et al., 2011*). This suggests that higher levels of motor variability do not solely amount to increased noise in the system. Instead, this variability represents a broader action space that can be capitalized upon during subsequent reinforcement learning by searching through previously explored actions (*Herzfeld and Shadmehr, 2014*). Accordingly, an implication of our results is that state anxiety could impair the potential benefits of an initial exploratory phase for subsequent learning.

Last, we used a reward-based motor learning paradigm in which different performances could provide the same feedback score. The rationale for using this task was to explore the effect of state anxiety on volatility estimates, as recent work demonstrates that anxiety primarily affects learning in volatile conditions (*Browning et al., 2015*; *Huang et al., 2017*). This scenario, however, implied that a high expression of task-related motor variability during learning would be associated with a more volatile perception of the task, which is indeed supported by our correlation results. This could be a confounding factor when explaining the group effects. Importantly, however, further analyses revealed that the total degree of motor variability during learning and the mean learned performance did not differ between groups, suggesting that these are not confounding factors that could explain the reward-based-learning group results. Instead, our findings underscore that computational mechanisms related to how pwPE on reward and volatility influence behavioral changes are the main factors driving the effects of concurrent or prior state anxiety on reward-based motor learning.

At the neural level, an important finding was that anxiety during initial exploration increased the power of beta oscillations and the rate of long beta bursts (long-tailed distribution). The increases in power and the rate of long-lived bursts manifested after completion of the sequence, reflecting an anxiety-related enhancement of the post-movement beta rebound (*Kilavik et al., 2012*; *Kilavik et al., 2013*). This effect was observed in a region of contralateral sensorimotor and right prefrontal channels, and could be explained by anxiety alone, despite a small effect of motor variability on the modulation of these neural changes across sensorimotor electrodes. Further, larger sensorimotor beta power at the termination of the sequence performance was associated with a more constrained use of task-related variability. Our analyses did not provide a detailed anatomical localization of the effect, but the findings in sensorimotor regions that partially contribute to changes in motor variability are consistent with the involvement of premotor and motor cortex in driving motor variability and learning, as previously reported in animal studies (*Churchland et al., 2006*; *Mandelblat-Cerf et al., 2009*; *Santos et al., 2015*). The results also converge with the representation in the premotor cortex of temporal and sequential aspects of rhythmic performance (*Crowe et al., 2014*; *Kornysheva and Diedrichsen, 2014*).

During learning, an unexpected result was that, in anx2 participants, there was an increase in beta power at the end of the sequence performance but not during feedback processing—and despite the anxiety manipulation successfully affecting the HRV. This outcome, as well as the lack of beta burst effects in this group, seems to be in agreement with the lack of learning impairments when compared with control participants. An additional unexpected result during learning blocks was the presence in anx1 participants of higher rates of long bursts and greater beta power at the end of the trial and during feedback processing, across both sensorimotor and prefrontal electrodes. These phasic changes in beta activity in anx1 participants extended from the previous phase, and the outcome aligns with the finding of prefrontal involvement in the emergence and maintenance of anxiety states (*Davidson, 2002*; *Grupe and Nitschke, 2013*; *Bishop, 2007*). Thus, our results revealed that, in the context of motor learning, anxious states induce changes in sensorimotor and prefrontal beta power and burst distribution. These changes are maintained after physiological measures of anxiety return to baseline, and thus continue to affect relevant behavioral parameters. Anxiety has been shown to modulate different oscillatory bands depending on the context, such as gamma activity in visual areas and amygdala when processing fearful faces (*Schneider et al., 2018*), alpha activity in response to processing emotional faces (*Knyazev et al., 2008*) or theta activity

during rumination (*Andersen et al., 2009*). Beta-band oscillations could be particularly relevant to flesh out the effects of anxiety on performance during motor tasks.

Mechanistically, phasic trial-by-trial feedback-locked changes in the sensorimotor beta power and burst distribution were related to the computational alterations in updating expectations on reward found in anx1 participants, and thus explained their poorer performance during reward-based learning. Specifically, a higher rate of long beta bursts and increased power following feedback were associated with a reduced update in the expectation of reward.

The computational quantity that determines the update of expectations in our Bayesian model is the precision-weighted PEs. Here, pwPE relating to reward were inversely related to the rate of long beta bursts and beta power, and were therefore attenuated in anx1 participants because of their enhanced feedback-related beta activity. We found no significant contribution of pwPE relating to volatility to explaining changes in beta activity, suggesting that additional frequency ranges should be considered when linking hierarchical pwPEs to neural oscillations during learning. In the context of the predictive coding hypothesis, PEs (or pwPEs) are hypothesized to be mediated by gamma oscillations, whereas the neuronal signaling of predictions is mediated by lower frequencies (e.g., alpha 8–12 Hz, *Friston et al., 2015*). Further studies point to beta oscillations as the cortical oscillatory rhythm associated with encoding predictions, although the evidence to date is scarce (*Arnal and Giraud, 2012*). More recently, beta oscillations have been associated with the change to predictions rather than with predictions themselves (*Sedley et al., 2016*), which is consistent with our findings as pwPEs were the quantities determining the change to predictions. In line with these results, a post-performance increase in beta power during motor adaptation is considered to index confidence in priors, and thus a reduced tendency to change the ongoing motor command (*Tan et al., 2016*). More generally, beta oscillations along cortico-basal ganglia networks have been proposed to gate incoming information to modulate behavior (*Leventhal et al., 2012*) and to maintain the current motor state (*Engel and Fries, 2010*). Consequently, the phasic increase in beta power and the rate of beta bursts following feedback presentation could represent neural states that impair the encoding of pwPEs and the update of predictions about lower level quantities—reward here—induced by anxiety states. Notably, the modulation of feedback-locked beta activity was not explained by changes in pwPE relating to volatility. We speculate that the effect of reduced reward estimates on the expectation of volatility in the HGF suggests that abnormal increases in beta activity following feedback presentation indirectly influenced volatility estimates, while it had a direct effect on reward expectation.

Our findings show that the assessment of neural activity in sensorimotor regions is crucial to understanding the effects of anxiety on motor learning and to determining mechanisms, above and beyond the role of prefrontal control of attention, in mediating the effects of anxiety on cognitive and perceptual tasks (*Bishop, 2007*; *Bishop, 2009*; *Eysenck and Calvo, 1992*). Our data imply that the combination of Bayesian learning models and analysis of oscillation properties can help us to better understand the mechanisms through which anxiety modulates motor learning. Future studies should investigate how the brain circuits that are involved in anxiety interact with motor regions to affect motor learning. In addition, assessing burst properties across both beta and gamma frequency ranges would further allow us to delineate and dissociate the neural mechanisms responsible for anxiety biasing decision-making and motor learning.

## Materials and methods

### Participants and sample-size estimation

Sixty right-handed healthy volunteers (37 females) aged 18 to 44 (mean 27 years, SEM, 1 year) participated in the main study. In a second, control experiment, 26 right-handed healthy participants (16 females, mean age 25.8, SEM 1, range 19–40) took part in the study. Participants gave written informed consent prior to the start of the experiment, which had been approved by the local Ethics Committee at Goldsmiths University. Participants received a base rate of either course credits or money (15 GBP; equally distributed across groups) and were able to earn an additional sum up to 20 GBP during the task depending on their performance.

We used pilot data from a behavioral study using the same motor task to estimate the minimum sample sizes for a statistical power of 0.95, with an $\alpha$ of 0.05, using the MATLAB (The MathWorks,

Inc, MA, USA) function sampsizepwr. In the pilot study, we had one control and one experimental group of 20 participants each. In the experimental group, we manipulated the reward structure during the first reward-based learning block (in this block, feedback scores did not count towards the final average monetary reward). For each behavioral measure (motor variability and mean score), we extracted the standard deviation (sd) of the joint distribution from both groups and the mean value of each separate distribution (e.g., m1, control; m2, experimental), which provided the following minimum sample sizes:

Between-group comparison of behavioral parameters (using a two-tailed t-test): MinSamplSizeA = sampsizepwr('t',[m1 sd],m2,0.95) = 18–20 participants.

Accordingly, we recruited 20 participants for each group in the main experiment. Next, using the behavioral data from the anxiety and control groups in the current main experiment, we estimated the minimum sample size for the second, behavioral control experiment:

Between-group comparison of behavioral parameters (using a two-tailed t-test): MinSamplSizeA = sampsizepwr('t',[m1 sd],m2,0.95) = 13 participants.

Therefore, for the second control experiment, we recruited 13 participants for each group.

## Apparatus

Participants were seated at a digital piano (Yamaha Digital Piano P-255, London, UK) and in front of a PC monitor in a light-dimmed room. They sat comfortably in an arm-chair with their forearms resting on the armrests of the chair. The screen displayed the instructions, feedback and visual cues for the start and end of a trial (*Figure 1A*). Participants were asked to place four fingers of their right hand (excluding the thumb) comfortably on four pre-defined keys on the keyboard. Performance information was transmitted and saved as Musical Instrument Digital Interface (MIDI) data, which provided time onsets of keystrokes relative to the previous one (inter-keystroke-interval—IKI in ms), MIDI velocities (related to the loudness, in arbitrary units, a.u.), and MIDI note numbers that corresponded to the pitch. The experiment was run using Visual Basic, an additional parallel port and MIDI libraries.

## Materials and experimental design

In all blocks, participants initiated the trial by pressing a pre-defined key with their left index finger. After a jittered interval of 1–2 s, a green ellipse appeared in the center of the screen representing the GO signal for task execution (*Figure 1A*). Participants had 7 s to perform the sequence, which was ample time to complete it before the green circle turned red indicating the end of the execution time. If participants failed to perform the sequence in the correct order or initiated the sequence before the GO signal, the screen turned yellow. In blocks 2 and 3 during learning, performance-based feedback in the form of a score between 0 and 100 was displayed on the screen 2 s after the red ellipse, that is, 9 s from the beginning of the trial. The scores provided participants with information regarding the target performance.

The performance measure that was rewarded during learning was the Euclidean norm of the vector corresponding to the pattern of temporal differences between adjacent IKIs for a trial-specific performance. Here, we denote the vector norm by $\|\Delta z\|$, with $\Delta \mathbf{z}$ being the vector of differences, $\Delta \mathbf{z} = (z_2 - z_1, z_3 - z_2, \ldots, z_n - z_{n-1})$, and $z_i$ representing the IKI at each keystroke ($i = 1, 2.., n$). Note that IKI values themselves represent the difference between the onset of consecutive keystrokes, and therefore $\Delta \mathbf{z}$ indicates a vector of differences of differences. Specifically, the target value of the performance measure was a vector norm of 1.9596 (e.g., one of the maximally rewarded performances leading to this vector norm of IKI-differences would consist of IKI values: [0.2, 1, 0.2, 1, 0,2, 1, 0.2] s; that is a combination of short and long intervals). The score was computed in each trial using a measure of proximity between the target vector norm $\|\Delta \mathbf{z}^t\|$ and the norm of the performed pattern of IKI differences $\|\Delta \mathbf{z}^p\|$, using the following expression:

$$score = 100\exp(-|\|\Delta \mathbf{z}^t\| - \|\Delta \mathbf{z}^p\||) \qquad (3)$$

In practice, different temporal patterns leading to the same vector norm $\|\Delta \mathbf{z}^p\|$ could achieve the same score. Participants were unaware of the existence of various solutions. Higher exploration across trials during learning could thus reveal that several IKI patterns were similarly rewarded. To account for this possibility, the perceived rate of change of the hidden goal (environmental volatility)

during learning was estimated and incorporated into our mathematical description of the relationship between performance and reward (see below).

## Anxiety manipulation

Anxiety was induced during block1 performance in group anx1, and during block2 performance in the anx2 group by informing participants about the need to give a 2 min speech to a panel of experts about an unknown art object at the end of that block (*Lang et al., 2015*). We specified that they would first see the object at the end of the block (it was a copy of Wassily Kandinsky' Reciprocal Accords [1942]) and would have 2 min to prepare for the presentation. Participants were told that the panel of experts would take notes during their speech and would be standing in front of the testing room (due to the EEG setup participants had to remain seated in front of the piano). Following the 2 min preparation period, participants were informed that due to the momentary absence of panel members, they instead had to present in front of the lab members. Participants in the control group had the task of describing the artistic object to themselves, and not in front of a panel of experts. They were informed about this secondary task before the beginning of block1.

## Assessment of state anxiety

To assess state anxiety, we acquired two types of data: (1) the short version of the Spielberger State-Trait Anxiety Inventory (STAI, state scale X1, 20 items; *Spielberger, 1970*) and (2) a continuous electrocardiogram (ECG, see EEG, ECG and MIDI recording session). The STAI X1 subscale was presented four times throughout the experiment. A baseline assessment at the start of the experiment before the resting state recording was followed by an assessment immediately before each experimental block to determine changes in anxiety levels. In addition, a continuous ECG recording was obtained during the resting state and three experimental blocks were used to assess changes in autonomic nervous system responses. The indexes of heart rate variability (HRV, coefficient of variation of the inter-beat-interval) and mean heart rate (HR) were evaluated, as their reduction has been linked to changes in anxiety state due to a stressor (*Feldman et al., 2004*).

## Computational model

Here, we provide details on the computational Bayesian model that we adopted to estimate participant-specific belief trajectories, determined by the mean (expectation) and variance (uncertainty) of the posterior distribution. The model was implemented using the HGF toolbox for MATLAB (http://www.translationalneuromodeling.org/tapas/). The model consists of a perceptual and a response model, representing an agent (a Bayesian observer) who generates behavioral responses on the basis of a sequence of sensory inputs that it receives. In many implementations of the HGF, the sensory input is replaced with a series of outcomes (e.g. feedback, reward) associated with participants' responses (*de Berker et al., 2016*; *Diaconescu et al., 2017*). As general notation, we let lowercase italics denote scalars (x), which can be further characterized by a trial superscript $x^k$ and a subscript i denoting the level in the hierarchy $x_i^k$ (i = 1, 2).

The HGF corresponds to the perceptual model, representing a hierarchical belief-updating process, that is a process that infers hierarchically related environmental states that give rise to sensory inputs (*Stefanics et al., 2018*; *Mathys et al., 2014*). In the version for continuous inputs (see *Mathys et al., 2014*; function $\mathrm{tapas\_hgf.m}$), we used the series of feedback scores as input: $u^k = \mathrm{score}$; normalized to range 0–1. From the series of inputs, the HGF then generates belief trajectories about external states, such as the reward value of an action or a stimulus. Learning occurs in two hierarchically coupled levels ($x_1$, $x_2$), one for 'perceptual' beliefs ($x_1$: the reward associated with the current performance), and the phasic volatility of those beliefs ($x_2$). These two levels evolve as coupled Gaussian random walks, with the lower level coupled to the higher level through its variance (inverse precision). The Gaussian random walk at each level $x_i$ is determined by its posterior mean ($\mu_i$) and its variance ($\sigma_i$). Further, the variance of the lower level, $x_1$, depends on $x_2$ through an exponential function:

$$f(x_2) = \exp(\kappa x_2 + \omega_1) \tag{4}$$

where $\kappa$ was fixed to 1 and $\omega_1$ is a model parameter that was estimated for each participant by fitting the HGF model to the experimental data (scores and responses) using Variational Bayes.

At the top level, the variance is typically fixed to a constant parameter, $\vartheta = exp(\omega_2)$, where $\omega_2$ is also a free parameter to be estimated in each individual. The specific coupling between levels indicated above has the advantage of allowing simple variational inversion of the model and the derivation of one-step update equations under a mean-field approximation. This is achieved by iteratively integrating out all previous states up to the current trial $k$ (see appendices in *Mathys et al., 2014*). Importantly, the update equations for the posterior mean at level $i$ and for trial $k$ depend on the prediction errors weighted by uncertainty $\sigma_i$ (or its inverse, precision $\pi_i = 1/\sigma_i$) according to the following expression:

$$\Delta\mu_i^k = \mu_i^k - \mu_i^{k-1} \propto \frac{\hat{\pi}_{i-1}^k}{\pi_i^k}\delta_{i-1}^k \tag{5}$$

The first term in the above expression is the change in the expectation for state $x_i$ on trial $k$, $\mu_i^k$, relative to the prediction on trial $k-1$, $\mu_i^{k-1}$. The prediction on trial $k-1$ is denoted by the 'hat' or diacritical mark ˆ, $\mu_i^{k-1} = \hat{\mu}_i^k$. The term prediction thus refers to the expectation of $x_i$ before seeing the feedback score from the current trial: it corresponds with the mean of the posterior distribution of $x_i$ up to trial $k-1$. By contrast, the term expectation refers to the mean of the posterior distribution of $x_i$ up to trial $k$. The difference term $\Delta\mu_i^k$ is proportional to the prediction error of the level below, $\delta_{i-1}^k$, representing the difference between the expectation $\mu_{i-1}^k$ and the prediction $\hat{\mu}_{i-1}^k$ of the level below . The prediction error is weighted by the ratio between the prediction of the precision of the level below, $\hat{\pi}_{i-1}^k$, and the precision of the current belief, $\pi_i^k$. Thus the product of the precision weights and the prediction error constitute the precision-weighed prediction error (pwPE), which therefore regulates the update of expectations on trial $k$: $\Delta\mu_i^k = \epsilon_i^k$. The pwPE expressions for level 1 and 2 are defined below in *Equation 14* and *Equation 15*. *Equation 5* illustrates that higher uncertainty in the current level ($\sigma_i^k$, lower $\pi_i^k$ in the denominator) leads to faster update of beliefs; moreover, smaller uncertainty (higher precision) of the prediction of the level below also increases the update of beliefs. For the two-level HGF model for continuous inputs, the generic equation *Equation 5* takes the explicit forms shown below (*Equation 6* and *Equation 10*; equations taken directly from the TAPAS toolbox; see also *Mathys et al., 2011*; *Mathys et al., 2014*).

Updates of expectations for level 1:

$$\mu_1^k = \hat{\mu}_1^k + \frac{\hat{\pi}_u^k}{\pi_1^k}\delta_u^k, \tag{6}$$

with $\hat{\pi}_u^k$ representing the prediction of the precision of the input (feedback scores; see *Table 1*) and $\delta_u^k$ the prediction error about the input:

$$\delta_u^k = u^k - \hat{\mu}_1^k, \tag{7}$$

Precision updates for level 1:

$$\pi_1^k = \hat{\pi}_1^k + \pi_u^k, \tag{8}$$

where $\hat{\pi}_1^k$ is defined as (using $\rho = 0, \kappa = 1, t^k = 1$):

$$\hat{\pi}_1^k = \frac{1}{\left(\frac{1}{\pi_1^{k-1}} + exp(\mu_2^{k-1} + \omega_1)\right)}, \tag{9}$$

Update of expectations for level 2:

$$\mu_2^k = \hat{\mu}_2^k + \frac{1}{2}\frac{1}{\pi_2^k}w_1^k\delta_1^k, \tag{10}$$

with

$$w_1^k = exp(\mu_2^{k-1} + \omega_1)\hat{\pi}_1^k \tag{11}$$

Precision updates for level 2:

$$\pi_2^k = \hat{\pi}_2^k + \frac{1}{2}\mathrm{w}_1^k(\mathrm{w}_1^k + (2\mathrm{w}_1^k - 1)\delta_1^k),$$ (12)

and

$$\hat{\pi}_2^k = \frac{1}{\frac{1}{\pi_2^{k-1}} + exp(\omega_2)}.$$ (13)

From *Equation 6* and *Equation 10*, it follows that the pwPEs for level 1 and 2, $\epsilon_1$ and $\epsilon_2$, respectively, are:

$$\epsilon_1^k = \mu_1^k - \hat{\mu}_1^k = \frac{\hat{\pi}_u^k}{\pi_1^k}\delta_u^k,$$ (14)

$$\epsilon_2^k = \mu_2^k - \hat{\mu}_2^k = \frac{1}{2}\frac{1}{\pi_2^k}\mathrm{w}_1^k\delta_1^k.$$ (15)

Next, we mapped the expectation on the inferred perceptual beliefs, reward $\mu_1$ and volatility $\mu_2$, and the corresponding pwPEs to the performance output that the participant generates during every trial using a separate response model. We adapted the family of response models used by *Marshall et al. (2016)* to our task. In that work, the authors explained participant's observed log (RT) responses on a trial-by-trial basis as a linear function of various HGF quantities using a multiple regression. We implemented similar models, but adapted them to our task (new scripts are available in the Open Science Framework Data Repository: https://osf.io/sg3u7/). The models we tested used two different performance parameters:

The coefficient of variation of inter-keystroke intervals, $\mathrm{cvIKI_{trial}}$, as a measure of the extent of timing variability within the trial.

The logarithm of the mean performance tempo in a trial, $\log(\mathrm{mIKI_{trial}})$, with IKI in milliseconds.

We were interested in how HGF quantities on the previous trial explained changes in the performance parameters in the subsequent trial and therefore used these dependent variables:

---

**Table 1.** Means and variances of the priors on perceptual parameters and initial values.
Priors on the parameters and initial values of the HGF perceptual model for continuous inputs. The continuous inputs here were the trial-by-trial scores that the participants received, normalized to the 0–1 range. Quantities estimated in the logarithmic space are denoted by log(). Prior mean and variance for $\mu_1^0$, as well as the prior mean for $\sigma_1^0$, $\omega_1$ and the precision of the input, $\pi_u^0$, were defined by the initial 20 input values. When providing prior values that depend on the first 20 input scores, we indicate the median across the total population of 60 participants. For the remaining quantities, the prior mean and variance were pre-defined according to the values indicated in the table.

| | Prior mean | Prior variance |
|---|---|---|
| $\log(\kappa)$ | log(1) | 0 |
| $\omega_1$ | log-variance of 1:20 input scores: −3.04 | 16 |
| $\omega_2$ | −4 | 16 |
| $\log(\pi_u^0)$ | negative log-variance of 1:20 input scores: 3.04 | 4 |
| $\mu_1^0$ | value of the first input score: 0.21 | variance of 1:20 input scores: 0.05 |
| $\log(\sigma_1^0)$ | log-variance of 1:20 input scores: −3.04 | 1 |
| $\mu_2^0$ | 1 | 0 |
| $\log(\sigma_2^0)$ | log(0.01) | 1 |
| $\beta_0$ | individual mean of behavioral parameter | 4 |
| $\beta_1$ | 0 | 4 |
| $\beta_2$ | 0 | 4 |

$$\Delta\text{cvIKI}_{\text{trial}}^{k} = \text{cvIKI}_{\text{trial}}^{k} - \text{cvIKI}_{\text{trial}}^{k-1}$$

$$\Delta\log(\text{mIKI}_{\text{trial}})^{k} = \log(\text{mIKI}_{\text{trial}}^{k}) - \log(\text{mIKI}_{\text{trial}}^{k-1})$$

For each of those two performance measures, the corresponding response model was a function of a constant component of the performance measure (intercept) and HGF quantities on the previous trial, such as: the expectation on reward ($\mu_1$), the expectation on volatility ($\mu_2$), the precision-weighted PE relating to reward ($\epsilon_1$), or the precision-weighted PE relating to volatility ($\epsilon_2$). In total, we assessed the following two families of four alternative response models HGF11-14 and HGF21-24.

Model HGF11:

$$\Delta\text{cvIKI}_{\text{trial}}^{k} = \beta_0 + \beta_1\mu_1^{k-1} + \beta_2\epsilon_1^{k-1} + \zeta$$

Model HGF12:

$$\Delta\text{cvIKI}_{\text{trial}}^{k} = \beta_0 + \beta_1\mu_1^{k-1} + \beta_2\mu_2^{k-1} + \zeta \tag{16}$$

Model HGF13:

$$\Delta\text{cvIKI}_{\text{trial}}^{k} = \beta_0 + \beta_1\mu_2^{k-1} + \beta_2\epsilon_2^{k-1} + \zeta \tag{17}$$

Model HGF14:

$$\Delta\text{cvIKI}_{\text{trial}}^{k} = \beta_0 + \beta_1\epsilon_1^{k-1} + \beta_2\epsilon_2^{k-1} + \zeta$$

Model HGF21:

$$\Delta\log(\text{mIKI}_{\text{trial}})^{k} = \beta_0 + \beta_1\mu_1^{k-1} + \beta_2\epsilon_1^{k-1} + \zeta \tag{18}$$

Model HGF22:

$$\Delta\log(\text{mIKI}_{\text{trial}})^{k} = \beta_0 + \beta_1\mu_1^{k-1} + \beta_2\mu_2^{k-1} + \zeta \tag{19}$$

Model HGF23:

$$\Delta\log(\text{mIKI}_{\text{trial}})^{k} = \beta_0 + \beta_1\mu_2^{k-1} + \beta_2\epsilon_2^{k-1} + \zeta \tag{20}$$

Model HGF24:

$$\Delta\log(\text{mIKI}_{\text{trial}})^{k} = \beta_0 + \beta_1\epsilon_1^{k-1} + \beta_2\epsilon_2^{k-1} + \zeta \tag{21}$$

The priors on the model parameters ($\omega_1, \omega_2$), the response model parameters ($\beta_0, \beta_1, \beta_2, \zeta$), the initial expected states ($\mu_1^0, \mu_2^0$) and the precision of the input ($\pi_u$) are provided in *Table 1*. All priors are Gaussian distributions in the space in which they are estimated and are therefore determined by their mean and variance. The variance is relatively broad to let the priors be modified by the series of inputs (feedback scores). Quantities that need to be positive (e.g., the variance or uncertainty of belief trajectories) are estimated in the log-space, whereas general unbounded quantities are estimated in their original space.

We used Random Effects Bayesian Model Selection (BMS) to assess the different models of learning at the group level (*Stephan et al., 2009*; code freely available from the MACS toolbox, *Soch and Allefeld, 2018*). First, the log-model evidence (LME) values for models HGF11-14 were combined to get the log-family evidence (LFE), and similarly for models HGF21-24. The LFE values were subsequently compared using BMS to assess which family of models provided more evidence. BMS generated (i) the estimated model-family frequencies, that is, how frequently each family of models is optimal in the sample of participants; and (ii) the exceedance probabilities, reflecting the posterior probability that one family is more frequent than the others (*Soch et al., 2016*). In the winner family, additional BMS determined the final optimal model.

## EEG, ECG and MIDI recording

EEG and ECG signals were recorded using a 64-channel (extended international 10–20 system) EEG system (ActiveTwo, BioSemi Inc) placed in an electromagnetically shielded room. During the recording, the data were high-pass filtered at 0.16 Hz. The vertical and horizontal eye-movements (EOG) were monitored by electrodes above and below the right eye and from the outer canthi of both eyes, respectively. Additional external electrodes were placed on both left and right earlobes as reference. The ECG was recorded using two external channels with a bipolar ECG lead II configuration. The sampling frequency was 512 Hz. Onsets of visual stimuli, key presses and metronome beats were automatically documented with markers in the EEG file. The performance was additionally recorded as MIDI files using the software Visual Basic and a standard MIDI sequencer program on a Windows Computer.

## EEG and ECG pre-processing

We used MATLAB and the FieldTrip toolbox (*Oostenveld et al., 2011*) for visualization, filtering and independent component analysis (ICA; runica). The EEG data were highpass-filtered at 0.5 Hz (Hamming windowed sinc finite impulse response [FIR] filter, 3380 points) and notch-filtered at 50 Hz (847 points). Artifact components in the EEG data related to eye blinks, eye movements and the cardiac-field artifact were identified using ICA. Following IC inspection, we used the EEGLAB toolbox (*Delorme and Makeig, 2004*) to interpolate missing or noisy channels using spherical interpolation. Finally, we transformed the data into common average reference.

Analysis of the ECG data with FieldTrip focused on detection of the QRS-complex to extract the R-peak latencies of each heartbeat and use them to evaluate the HRV and HR measures in each experimental block.

## Analysis of power spectral density

We first assessed the standard power spectral density (PSD, in $\mathrm{mV^2/Hz}$) of the continuous raw data in each performance block and separately for each group. The PSD was computed with the standard fast Fourier Transform (Welch method, Hanning window of 1 s with 50% overlap). The raw PSD estimation was normalized into decibels (dB) with the average PSD from the initial rest recordings (3 min). Specifically, the normalized PSD during the performance blocks was calculated as ten times the base-10 logarithm of the quotient between the performance-block PSD and the resting state power.

In addition, we assessed the time course of the spectral power over time during performance. Trials during sequence performance were extracted from −1 to 11 s locked to the GO signal. This interval included the STOP signal (red ellipse), which was displayed at 7 s, and—exclusively in learning blocks—the score feedback, which was presented at 9 s. Thus, epochs were effectively also locked to the STOP and Score signals. Artifact-free EEG epochs were decomposed into their time-frequency representations using a 7-cycle Morlet wavelet in successive overlapping windows of 100 ms within the total 12s-epoch. The frequency domain was sampled within the beta range from 13 to 30 Hz at 1 Hz intervals. For each trial, we thus obtained the complex wavelet transform, and computed its squared norm to extract the wavelet energy (*Ruiz et al., 2009*). The time-varying spectral power was then simply estimated by averaging the wavelet energy across trials. This measure of spectral power was further averaged within the beta-band frequency bins and normalized by subtracting the mean and dividing by the standard deviation of the power estimate in the pre-movement baseline period ([−1, 0] s prior to the GO signal).

## Extraction of beta-band oscillation bursts

We estimated the distribution, onset and duration of oscillation bursts in the time series of beta-band amplitude envelope. We followed a procedure adapted from previous work to identify oscillation bursts (*Poil et al., 2008*; *Tinkhauser et al., 2017*). In brief, we used as threshold the 75% percentile of the amplitude envelope of beta oscillations. Amplitude values above this threshold were considered to be part of an oscillation burst if they extended for at least one cycle (50 ms, as a compromise between the duration of one 13 Hz-cycle [76 ms] and 30 Hz-cycle [33 ms]). Threshold-crossings that were separated by less than 50 ms were considered to be part of the same oscillation burst. As an additional threshold, the median amplitude was used in a control analysis, which revealed qualitatively similar results, as expected from previous work (*Poil et al., 2008*). Importantly,

because threshold crossings are affected by the signal-to-noise ratio in the recording, which could vary between the different performance blocks, we selected a common threshold from the initial rest recordings separately for each participant (*Tinkhauser et al., 2017*). Distributions of the rate of oscillation bursts per duration were estimated using equidistant binning on a logarithmic axis with 20 bins between 50 ms and 2000 ms.

General burst properties were assessed during exploration and learning blocks separately, first as averaged values within the full block-related recording, and next as phasic changes over time during trial performance. Trial-based analysis focused on the interval 0–11000 ms following the GO signal, which included the time window following the STOP signal (at 7000 ms: exploration and learning blocks) and the score feedback (at 9000 ms: learning blocks).

### Statistical analysis

Statistical analysis of behavioral and neural measures focused on the separate comparison between each experimental group and the control group (contrasts: anx1 – controls, anx2 –controls). Differences between experimental groups, anx1 – anx2, were evaluated exclusively concerning the overall achieved monetary reward. We used non-parametric pair-wise permutation tests to assess differences between conditions or between groups in the statistical analysis of behavioral or computational measures. When multiple testing was performed, we implemented a control of the false discovery rate (FDR) at level q = 0.05 using an adaptive linear step-up procedure (*Benjamini et al., 2006*). This control provided an adapted threshold p-value (termed $P_{FDR}$). Further, to evaluate differences between sets of multi-channel EEG signals corresponding to two conditions or groups, we used two approaches:

1. Tonic changes in average beta PSD or the scaling exponent of the burst distribution were assessed using two-sided cluster-based permutation tests (*Maris and Oostenveld, 2007*) and an alpha level of 0.025. Here, we used all 64 channels and let the statistical method extract the significant clusters. Control of the family-wise error (FWE) rate was implemented in these tests to account for the problem of multiple comparison (*Maris and Oostenveld, 2007*).

2. Phasic or event-related changes in beta power or burst rate across time were assessed using pair-wise permutation tests at each time point and exclusively in a subset of channels across sensorimotor and anterior (prefrontal) electrode regions (*Figure 10—figure supplement 1*). The relevant subset was chosen to ameliorate the number of multiple comparisons arising from time and space—channels). When using these tests, we implemented a control of the FDR at level q = 0.05 to correct for multiple comparisons.

Non-parametric effect size estimators were used in association with our pair-wise nonparametric statistics, following *Grissom and Kim, 2012*. In the case of between-subject comparisons, the standard probability of superiority, $\Delta$, was used. $\Delta$ is defined as the proportion of greater values in sample B relative to A, when values in samples A and B are not paired: $\Delta = P(A>B)$ ranges from 0 to 1. The total number of comparisons is the product of the size of sample A and sample B ($Ntot = sizeA * sizeB$), and therefore, $\Delta = N(A>B)/Ntot$. In the case of ties, $\Delta$ is corrected by subtracting in the denominator the number of ties from the total number of comparisons ($Ntot - Nties$). For within-subject comparisons, we used the probability of superiority for dependent samples, $\Delta_{dep}$, which is the proportion of all within-subject (paired) comparisons in which the values for condition B are larger than for condition A. 95% confidence intervals (termed simply CI) for $\Delta$ and $\Delta_{dep}$ were estimated with bootstrap methods (*Ruscio and Mullen, 2012*). Last, associations between parameters were quantified using non-parametric rank correlations (Spearman $\rho$), which are robust against outliers. However, we used linear correlations in the case of multiple linear regressions for the HGF response model, following *Marshall et al. (2016)*.

## Acknowledgements

This research is supported by the British Academy through grant R134610 to MHR and by the Economic and Social Research Council through PhD grant ES/P00072X/1 to TPH. MHR was partially supported by the HSE Basic Research Program and the Russian Academic Excellence Project '5–100'. We thank Marta García Huesca and Silvia Aguirre for carrying out some of the EEG experiments.

## Additional information

### Funding

| Funder | Grant reference number | Author |
|---|---|---|
| British Academy | R134610 | Maria Herrojo Ruiz |
| Economic and Social Research Council | ES/P00072X/1 | Thomas Hein |
| National Research University Higher School of Economics | Basic Research Program | Maria Herrojo Ruiz |
| Ministry of Education and Science of the Russian Federation | Russian Academic Excellence Project 5–100 | Maria Herrojo Ruiz |

The funders had no role in study design, data collection and interpretation, or the decision to submit the work for publication.

### Author contributions

Sebastian Sporn, Conceptualization, Investigation, Writing - review and editing; Thomas Hein, Investigation, Writing - review and editing; Maria Herrojo Ruiz, Conceptualization, Software, Formal analysis, Supervision, Funding acquisition, Investigation, Visualization, Methodology, Writing - original draft, Project administration, Writing - review and editing

### Author ORCIDs

Maria Herrojo Ruiz https://orcid.org/0000-0001-8948-9444

### Ethics

Human subjects: Participants gave written informed consent prior to the start of the experiment, including written consent to potentially share de-identified data with other researchers. Experimental procedures were approved by the research ethics committee of Goldsmiths University of London.

### Decision letter and Author response

Decision letter https://doi.org/10.7554/eLife.50654.sa1
Author response https://doi.org/10.7554/eLife.50654.sa2

## Additional files

### Supplementary files

• Transparent reporting form

### Data availability

MIDI (performance) and EEG data, as well as new response model scripts, have been deposited in the Open Science Framework Data Repository under the accession code mfe2j.

The following dataset was generated:

| Author(s) | Year | Dataset title | Dataset URL | Database and Identifier |
|---|---|---|---|---|
| Ruiz MH | 2019 | Motor Learning and Anxiety - Data repository - behavioral, electrophysiological | https://osf.io/mfe2j/ | Open Science Framework, mfe2j |

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
