## [Decision Letter]

**Acceptance summary:**

In this article, the authors manipulate state anxiety and examine the relationship between anxiety and motor learning. Using electrophysiology and modeling approaches, they show that anxiety constrains flexible behavioral updating.

**Decision letter after peer review:**

Thank you for submitting your article "Alterations in the amplitude and burst rate of beta oscillations impair reward-dependent motor learning in anxiety" for consideration by *eLife*. Your article has been reviewed by three peer reviewers, including Nicole C Swann as the Reviewing Editor and Reviewer #3, and the evaluation has been overseen by a Reviewing Editor and Laura Colgin as the Senior Editor. The following individuals involved in review of your submission have agreed to reveal their identity: Preeya Khanna (Reviewer #1).

The reviewers have discussed the reviews with one another and the Reviewing Editor has drafted this decision to help you prepare a revised submission.

Summary:

This article addresses the relationship between anxiety and motor learning. Specifically, the authors show that anxiety during a baseline exploration phase caused subsequent impairments in motor learning. They go on to use a Bayesian modeling approach to show that this impairment was due to biased estimates of volatility and performance goal estimates. Finally, they couple their behavioral analyses to electrophysiology recordings with a particular focus on sensorimotor (and to a lesser extend prefrontal) beta. They show that post-movement beta rebound is elevated in the anxiety condition. The authors also utilized a novel "beta bursting approach", which in some ways recapitulated the beta power findings, but using a contemporary and exciting method which likely more accurately captures brain activity. Using this approach they show power difference may be driven by increases in burst duration in the anxiety condition – which parallels recent findings in Parkinson's disease populations.

Overall, the reviewers were impressed with many aspects of this manuscript. We appreciated the multi-modal approach (incorporating heart rate measures, clinical rating scales, modeling, and electrophysiology). We also found the behavioral results related to anxiety and motor learning particularly interesting given that they contribute to the existing literature on reward-based learning and volatility, but extends these findings to the motor domain. We also appreciated that the author's actually manipulated state anxiety (rather than relying on individual differences) since this approach allows stronger inferences to be made about causality. Finally, the reviewers noted that the extension of sensorimotor beta outside the motor domain is a novel contribution.

While we were overall enthusiastic, the reviewers did note difficulty in reading the paper. Although the writing was generally clear, the rationale and flow of the presentation of results, particularly for the modeling and EEG findings, were often difficult to understand. For instance, we felt that overall the presentation of the EEG results did not follow a logical flow, and it was sometimes difficult to understand why certain analytic methods were chosen. We also noted that the link between the different modalities could be made more clearly and that additional controls could be added to rule out a motoric contribution to the beta effects. Finally, additional information is needed for the model results. We elaborate on each of these points further below.

Essential revisions:

1) We suggest the authors carefully consider the nomenclature of the conditions and how each relates to motor learning.

For instance, referring to the "exploration" phase as "baseline" caused some confusion since "baseline" typically implies some "pre-manipulation" phase of task.

Related to above, further consideration of how the conditions map onto motor learning would be helpful. In this study, subjects were already instructed to explore task-related dimensions during the baseline period, but were not given feedback during this period. It is unclear how this maps to typical motor "exploration" in the reinforcement learning sense since there is no reinforcement during this period. Additionally, it isn't just a passive baseline measurement since subjects are actively doing something. Further interpretation of how this exploration/baseline phase maps onto other motor learning paradigms, either in the Introduction or Discussion section, would be helpful.

2) Similarly, the use of the terms "learning" and "training" for the second phase of the experiment caused us some confusion. A consistent terminology would have made the manuscript easier to follow.

3) Overall, a strength of the study is the use of many different modalities; however, at present, findings from these modalities are often not linked together. It would be helpful to tie the disparate methods together if some analyses were done to link the different measures. For instance, additional plots like those in Figure 3C-D could be included which correlate different measures to one another across participants. (For example, (a) correlating the model predictions (i.e. belief of environment volatility) and higher variability in cvIKI on a subject-to-subject basis to help link the more abstract model parameters to behavioral findings and (b) correlating post-feedback beta power with both volatility estimates and cvIKI variability.)

4) In general, the figures could benefit from more labeling and clarification. Some specific examples are mentioned below, but in general, it was not always clear which electrodes data were from, what time periods were shown, which groups, etc.

5) Please include model fits with the results (i.e. how well do they estimate subjects' behavior on a trial-by-trial basis and are there any systemic differences in the model fits across groups?).

6) Please provide a summary figure showing what data is included in the model and perhaps a schematic that illustrates what the model variables are and example trajectories that the model generates.

7) It would be helpful to provide examples to give some intuition about what types of behavior would drive a change in "volatility". For example, can more information be provided to help the reader understand if the results (presented in Figure 10 for instance) enable predictions about subjects' behavior? If beta is high on one trial during the feedback period, does that mean that the model makes a small change in the volatility estimate? How does this influence what the participants are likely to do on the next trial?

8) Generally, the EEG analysis opens up a massive search space (all electrodes, several seconds of data, block-wise analyses, trial-wise analyses, sample-wise analyses, power quantifications, burst-quantifications, long bursts, short bursts, etc.), and the presentation of the findings often jump around frequently between power quantification, burst-quantifications, block-wise, and trial-wise analysis etc. It would be much easier to follow if a few measurements were focused on that were a priori justified. These could be clearly laid out in the introduction with some explanation as to why they were investigated and what each measure might tell the reader. Then, if additional analyses were conducted, these should be explained as post hoc with appropriate justifications and statistical corrections.

9) The EEG results could be better connected to the other findings: for instance, by correlating beta results to model volatility estimates or cvIKI variability, as described above.

10) The reviewers felt that an important contribution of this paper was the potential non-motor findings related to sensorimotor beta. However, because there were also motoric differences between conditions, it seems very important to verify whether the beta differences were driven by motoric differences or anxiety-related manipulations. We appreciate the analyses in Figure 8—figure supplement 1 to try to rule out the motoric contribution to the sensorimotor beta differences, but note that this only controlled for certain kinds of movement variability. We would like to see controls for other possible differences in movement between the conditions, for instance differences in movement length, or movement length variability. Finally, is there a way to verify if the participants moved at all after they performed the task?

11) We would like to see what the between group differences for beta power and beta bursts look like during the rest period before the baseline? (For instance, if Figure 7 were generated for rest data?)

[Editors' note: further revisions were suggested prior to acceptance, as described below.]

Thank you for re-submitting your article "Alterations in the amplitude and burst rate of beta oscillations impair reward-dependent motor learning in anxiety" for consideration by *eLife*. Your article has been reviewed by three peer reviewers including Nicole C Swann as the Reviewing Editor and Reviewer #3, and the evaluation has been overseen by a Reviewing Editor and Laura Colgin as the Senior Editor. The following individuals involved in review of your submission have agreed to reveal their identity: Preeya Khanna (Reviewer #1); Jan R Wessel (Reviewer #2).

The reviewers have discussed the reviews with one another and the Reviewing Editor has drafted this decision to help you prepare a revised submission.

Summary:

In general, the authors did a good job addressing our comments. We were especially happy with clarified EEG analysis and, in particular, the care the authors took to avoid potential motoric drivers of beta differences. Given that the authors made significant changes to the manuscript in response to our previous comments, we send the paper out for re-review, and identified a few remaining items in need of clarification. The majority of these are related to the updated model, but we also had a few questions about the EEG analysis, code sharing, and minor points (typos, etc.) We elaborate on these below.

Essential revisions:

Related to the updated model: We have summarized some aspects of the modeling that we believe would benefit from additional explanation (to make the manuscript more broadly accessible). We apologize that we did not bring some of these up in the first submission, but these questions arose either due to the use of the new model or because of clarifications in the revision that provided new insight to us about the model.

1) Explanation/interpretation of the Bayesian modeling – Definitions:

Thank you for Figure 5 – this added clarity to the modeling work but we still are having trouble understanding the general structure of the model. It would be helpful to clearly define the following quantities that are used in the text (in the Materials and methods section before any equations are listed), and ideally also in a figure of example data.

"input" – does this mean the score for a specific trial k? We found this a little misleading since an "input" would usually mean some sort of sensory or perceptual input (as in Mathys 2014), but in this case it actually means feedback score (if we understood correctly). Also, please define u^k^ before Equation 3 and Equation 4, and ideally somewhere in Figure 5;

- Please clarify how the precision of the input is measured? (Used in Equation 3).

"predicted reward" – from the Mathys et al., 2014 paper we gathered that this is the mean of x_1_ obtained on the previous trial? Is this correct? If so, please clarify/emphasize. To increase broad accessibility of the manuscript, it would be helpful to summarize in words somewhere what the model is doing to make predictions. For example, in the paragraph after Equation 2, we weren't sure what the difference is between prediction of x_1_ and expectation of x_1_. Typically this terminology would correspond to: prediction = E(x_1_ on trial k | information up to trial k-1), and expectation = E(x_1_ on trial k | information up to trial k) but it wasn't clear to us.

"variance vs. precision vs. uncertainty" – These are well defined words but it would also help immensely to only use "variance" or "precision" or "uncertainty" in the explanation/equations. Mentally jumping back and forth gets confusing.

- "belief vs. expectation" – Are these the same? What is the mathematical definition (is it Equations 3 and 7)?

- "pwPE" – please list the equation for this somewhere in the methods, ideally before use of the epsilons in the response variable models.

2) Inputs/outputs of the model:

Inputs – we gather that the input to the model is the score that the participant receives. Then x_1_ gets updated according to Equation 3. So x_1_ is tracking the expected reward on this trial assuming that the reward on the previous trial must be updated by a prediction error from the current trial? Is this reasonable assumption for this task? What if the participants are exploring new strategies trial to trial? Why would they assume that the reward on the next trial is the same as the current trial (i.e. why is the predicted reward = uik-1)? Or is this the point (i.e. if trial to trial the subjects change their strategy a lot that this will end up being reflected as a higher "volatility")? It would be helpful to outline how the model reflects different regimes of behavior (i.e. what does more exploratory behavior look like vs. what is learning expected to look like).

Outputs – response models; please clarify why cvIKI_trial_ and log(mIKI) are the chosen responses since these are not variables that are directly responsible for the reward? We thought that the objective of this response modeling was to determine how a large prediction error on the previous trial would influence action on the next trial? Perhaps an output metric could be [similarity between trial k, trial k-1] = B0 + B1(uik-1) + B2(pwPE_k-1_)? So, depending on the reward and previous prediction error, you get a prediction of how similar the next trials' response is to the current trials' response? Right now, we don't understand what is learned from seeing that cvIKI_trial_ is higher with higher reward expectation (this is almost by necessity right? because the rewarded pattern needs high cvIKI) or higher prediction error.

3) Interpretation of the model:

Is the message from Figure 6A that the expected reward is lower for anx1 than anx2 and control? Since the model is trying to predict scores from actual score data, isn't this result expected given Figure 4A. Can the authors please clarify this?

We noticed that log(μ_2_) is lower for anx1 and anx2 than control. Should this correspond to a shallower slope (or a plateau in score that is reached more quickly) in Figure 4A over learning for anx1? If so, why don't we see that for anx2? If this is true, and given that cvIKI is no different for anx1, anx2, and control, wouldn't that mean that the reward rate is plateauing faster for anx1 and anx2 while they are still producing actions that are equally variable to control? So, are participants somehow producing actions that are variable yet getting the same reward – so they're getting "stuck" earlier on in the learning process? Can the authors provide some insight into what type of behavior trends to expect given the finding of Figure 6B-C? Right now all the reader gets as far as interpretation goes is that the anx1 group underestimates "environmental volatility" and that the mean behavior and cvIKI is the same across all groups.

Does underestimating volatility mean that subjects just keep repeating the same sequence over and over? If so, can that be shown? Or does it mean that they keep trying new sequences but fail to properly figure out what drives a higher reward? Since the model is fit on the behavior of the participants, it should be possible to explain more clearly what drives the different model fits.

Related EEG Analysis: We greatly appreciated the clarified EEG analysis. Re-reading this section, we were able to understand what was done much better, but had two queries related to the analysis.

1) We noted that the beta envelope in Figure 7A looks unusual. It looks almost like the absolute value of the beta – filtered signal rather than the envelope, which is typically smoother and does not follow peaks and troughs of the oscillation. Can the authors please clarify how this was calculated?

2) In subsection “Analysis of power spectral density”, the authors write: "The time-varying spectral power was computed as the squared norm of the complex wavelet transform, after averaging across trials within the beta range." This sounds like the authors may have calculated power after averaging across trials? Is this correct (i.e. was the signal averaged before the wavelet transform, such that trial to trial phase differences may cancel out power changes?), or do the authors mean that they averaged across trials after extracting beta power for each trial? If the former the author should emphasize that this is what they did, since it is unconventional.

3) To try to understand point 2 above, we checked if the authors had shared their code, and found that, although data was shared, code was not, as far as we could tell. *eLife* does require code sharing as part of their policies (https://reviewer.elifesciences.org/author-guide/journal-policies) so please include that.

---

## [Author Response]

Essential revisions:1) We suggest the authors carefully consider the nomenclature of the conditions and how each relates to motor learning.For instance, referring to the "exploration" phase as "baseline" caused some confusion since "baseline" typically implies some "pre-manipulation" phase of task.Related to above, further consideration of how the conditions map onto motor learning would be helpful. In this study, subjects were already instructed to explore task-related dimensions during the baseline period, but were not given feedback during this period. It is unclear how this maps to typical motor "exploration" in the reinforcement learning sense since there is no reinforcement during this period. Additionally, it isn't just a passive baseline measurement since subjects are actively doing something.

Agreed; the first experimental phase (termed baseline before) has been relabeled as “initial exploration” or, in some instances, “exploration” phase.

We prefer the term “initial exploration” as it should be understood as the first experimental phase (block 1). This does not imply that participants did not to use some degree of exploration in learning phase. The learning phase was indeed expected to require some degree of exploration during the first trials, followed by exploitation of the inferred performance goal (see below, and Figure 4—figure supplement 1). This transition from exploration to exploitation during the learning blocks directly relates to earlier investigations of reinforcement learning (see below).

In the revised manuscript, we have clarified why we used the initial motor exploration phase: “The rationale for including a motor exploration phase in which participants did not receive trial-based feedback or reinforcement was based on the findings that initial motor variability (in the absence of reinforcement) can influence the rate at which participants learn in a subsequent motor task (Wu et al., 2014).”

The findings of Wu et al., 2014 are significant in demonstrating that initial motor variability measured when participants perform ballistic arm movements in the absence of reinforcement or visual feedback can predict the rate of reward-based learning in a subsequent phase.

Similarly, in our study the initial motor exploration phase aimed to assess an individual's use of motor variability in the absence of feedback and when there was no hidden goal to infer. Motor variability here would be driven by internal motivation (and/or motor noise) and would not be guided by explicit external reward.

The fundamental question for us was to determine whether larger task-related variability during block 1 would improve subsequent reward-based learning, even if during the learning blocks a successful performance required participants to exploit the inferred goal. We have created Figure 4—figure supplement 1, which illustrates the result of progressive reduction in temporal variability in the learning blocks (increased exploitation) as participants approached and aimed to maintain the solution. This drop in temporal variability is one of the hallmarks of learning (Wolpert et al., 2010).

Based on our results, we suggest that initial exploration may facilitate learning of the mapping between the actions and their sensory consequences (even without external feedback)”, which had a positive influence on subsequent learning “from performance-related feedback”.

Further interpretation of how this exploration/baseline phase maps onto other motor learning paradigms, either in the Introduction or Discussion section, would be helpful.

Thanks. By assessing motor variability during an initial exploration period before a reward-based learning period, Wu et al., (2014) positively correlated initial variability with learning curve steepness during training – a relationship previously observed in the zebra finch (Kao et al. 2005, Olveczky et al., 2005, 2011). This suggests that higher levels of motor variability do not solely amount to increased noise in the system. Instead, this variability represents a broader action space that can be capitalised upon during subsequent reinforcement learning by searching through previously explored actions (Herzfeld and Shadmehr, 2014). However, two recent studies using visuomotor adaptation paradigms could not find a similar correlation between motor variability and the rate of motor adaptation (He et al., 2016, Singh et al., 2016). Aiming to align this discrepancy in results, Dhawale et al., (2017) identified that in contrast to Wu et al., (2014), the aforementioned studies gave task-relevant feedback during baseline, which in turn updates the internal model of the action, accentuating execution noise over planning noise. They hypothesise that variability driven by planning noise underlies learning-related motor exploration (Dhawale et al., 2017). In this study, we aimed to investigate the effect of state anxiety on initial variability prior to a reward-based learning period.

We had summarised those arguments in the previous Discussion. Now, we have also added:

Discussion section: “Another consideration is that our use of an initial exploration phase that did not provide reinforcement or feedback signals was motivated by the work of Wu and colleagues (2014), which demonstrated a correlation between initial variability (no feedback) and learning curve steepness in a subsequent reward-based learning phase– a relationship previously observed in the zebra finch (Kao et al., 2005; Olveczky et al., 2005, 2011). This suggests that higher levels of motor variability do not solely amount to increased noise in the system. Instead, this variability represents a broader action space that can be capitalised upon during subsequent reinforcement learning by searching through previously explored actions (Herzfeld and Shadmehr, 2014). Accordingly, an implication of our results is that state anxiety could impair the potential benefits of an initial exploratory phase for subsequent learning.”

2) Similarly, the use of the terms "learning" and "training" for the second phase of the experiment caused us some confusion. A consistent terminology would have made the manuscript easier to follow.

Agreed, we have settled for “learning”. The term “training” was used in analogy to Wu et al., (2014) – learning is more appropriate.

3) Overall, a strength of the study is the use of many different modalities; however, at present, findings from these modalities are often not linked together. It would be helpful to tie the disparate methods together if some analyses were done to link the different measures. For instance, additional plots like those in Figure 3C-D could be included which correlate different measures to one another across participants. (For example, (a) correlating the model predictions (i.e. belief of environment volatility) and higher variability in cvIKI on a subject-to-subject basis to help link the more abstract model parameters to behavioral findings and (b) correlating post-feedback beta power with both volatility estimates and cvIKI variability.)

Agreed.

a) The new family of response models used allowed us to obtain the best model that links trial-by-trial behavioural responses and HGF quantities. Details are provided below in our reply to Q7.

In brief, the winning response model explains the variability of temporal intervals within the trial (cvIKI_trial_) as a linear function of the reward estimates, μ_1_, and the precision-weighted PE about reward, ε_1_. This model outperformed other alternative response models that used μ_2_, ε_2_ and different combinations of μ_1_, μ_2_, ε_1_, ε_2_, as well as a different response measure (logarithm of the mean IKI).

Thus, an increase in the estimated reward μ_1_ and an enhanced pwPE ε_1_ that drives belief updating about reward would contribute to a larger degree of temporal variability (less isochronous performance) on the current trial. This result is intuitively meaningful as the score was directly related to the norm of the difference IKI values across successive keystrokes and the hidden goal actually required a relatively large difference between successive IKI values, which would also be associated with larger cvIKI_trial_ values. Thus, the winning response model captured how the inferred environmental states (μ_1_ and e_1_) mapped to the observed responses (cvIKI_trial_) on a trial-by-trial basis.

Note that the trial-wise measure cvIKI_trial_ is different from the standard measure of motor variability *across trials* we used in the manuscript, cvIKI.

New Figure 6—figure supplement 1: Across all our participants, the measure of changes in across-trial temporal variability (cvIKI: difference from learning block1 to block2) was positively associated with the changes in volatility estimates (μ_2_: difference between learning block2 and block1). This was revealed in a non-parametric Spearman correlation (rr_2_ = 0.398, p = 0.002), supporting that participants who performed more different timing patterns across trials in block2 relative to block1 also increased their volatility estimate in block2 as compared to block1. Conversely, participants who showed a tendency to exploit the rewarded performance decreased their estimate of volatility.

b) Correlations between post-feedback beta power and HGF estimates:

Because in the predictive coding framework the quantities that are thought to dominate the EEG signal are the pwPEs (Friston and Kiebel, 2009), we had assessed the relation between belief updates (regulated by pwPEs on level 1 and 2) and the post-feedback beta activity. The revised manuscript also follows this approach, but we have improved the analysis by assessing simultaneously the effect of e_1_ and e_2_ on the beta power activity running a multiple linear regression in all participants. The results indicate that both e_1_ and e_2_ have a significant negative effect on the beta activity (power and rate of long bursts) across participants. Furthermore, the analysis demonstrates that using e_2_ as second predictor in the multiple regression analysis adds significant predictive power to using simply e_1_ as a predictor.

We did not expect beta activity to facilitate the “encoding” of volatility estimates directly, but only precision-weighted PEs about volatility. Accordingly, our results linking post-feedback beta activity to pwPE about reward and volatility provide a mechanism through which beliefs about volatility (and reward) are updated.

For the reviewers, we have also assessed the correlation between the mean post-feedback beta activity (power) and the degree of motor variability across trials during the learning blocks, cvIKI, and we found no significant association (Spearman ρ < 0.08, P = 0.56). This suggests that post-feedback beta activity is not associated on a trial-by-trial basis with the overall degree of motor variability, but rather with the step of the updates in beliefs (e_1_, e_2_).

By contrast, during the initial exploration phase, there was a significant non-parametric correlation between the averaged beta activity after the STOP signal and the degree of motor variability across trials (Spearman ρ < -0.4397, P = 0.0001). This result links increased use of motor variability during exploration with a reduction in beta power following trial performance. See new Figure 8—figure supplement 6.

4) In general, the figures could benefit from more labeling and clarification. Some specific examples are mentioned below, but in general, it was not always clear which electrodes data were from, what time periods were shown, which groups, etc.

Agreed. We have made the labeling of analyses and figures more explicit.

In the large figures with subplots, e.g. Figure 8, and former Figure 8—figure supplements 1-5 we had used one topographic sketch to illustrate the electrodes of the effect across all measures, although the sketch was used in only one of the subplots, in the one with more empty space to allow for the inset. We have kept this system for the figure, but we now added a clarification in the figure caption.

5) Please include model fits with the results (i.e. how well do they estimate subjects' behavior on a trial-by-trial basis and are there any systemic differences in the model fits across groups?).

Agreed. In the revised manuscript we provide as Figure 5—figure supplement 3 the grand-average of the trial-by-trial residuals in each group. The residuals represent the trial-by-trial difference between the observed responses (y) and those predicted by the model (predResp): res = y – predResp.

In the winning response model (see below for new response models tested), the relevant response variable that was identified was cvIKI_trial_ (cv of IKI values across keystroke positions in a trial).

We also summarise here the results from Figure 5—figure supplement 3 by computing in each group the mean residual values across trials:

cont: 0.0001 (0.0002)

anx1: 0.0001 (0.0001)

anx2: 0.0002 (0.0001)

In the second control experiment we obtained the following mean residual values per group:

cont: 0.0008 (10^-6^)

anx3: 0.0001 (0.0008)

There were thus no systematic differences in the model fits across groups and the low mean residual values further indicate that the model captured the fluctuations in data well.

6) Please provide a summary figure showing what data is included in the model and perhaps a schematic that illustrates what the model variables are and example trajectories that the model generates.

Thanks for the suggestion. We have added a schematic in Figure 5 illustrating the model's hierarchical structure and the belief trajectories.

In addition, we have provided the detailed update equations for belief and precision estimates in the two-level HGF perceptual model (equations 3-10). This will improve the understanding of how relevant model output variables evolve in time. Moreover, in the revised manuscript we have used more complete response models, using as reference the work by Marshall et al., (2016), that allow us to address the next question raised by the reviewers (Q7, see below). How the response model parameters influence the input to the two-level perceptual model is also reflected in the equations and the schematic in Figure 5. Details on the new response models are provided in Q7.

In Figure 5, we indicate how model parameters ω_1_ and ω_2_ influence the estimates at each level. Parameter ω_1_ represents the strength of the coupling between the first and second level, whereasω_2_ modulates how precise participants consider their prediction on that level (larger π^2 or smaller ω_2_). Thus, ω_1_ and ω_2_ additionally characterise the individual learning style (Weber et al., 2019).

The new Figure 5—figure supplement 1 illustrates using simulated data how different values of ω_1_ or ω_2_ affect the changes in belief trajectories across trials, for an identical series of input scores. In Figure 5—figure supplement 1A we can observe how smaller values of ω_1_ attenuate the general level of volatility changes (less pronounced updates or reduction). By contrast, in panel Figure 5—figure supplement 1C, we note that ω_2_ regulates the scale of phasic changes on a trial-by-trial basis, with larger ω_2_ values inducing more sharp or phasic changes to prediction violations in the level below (changes in PE at level 1).

In terms of the analysis of the computational quantities, we have now added a between-group comparison in ω_1_ and ω_2_. The results highlight that “In addition to the above-mentioned group effects on relevant belief and uncertainty trajectories, we found significant differences between anx1 and control participants in the perceptual parameter ω1(mean and SEM values for ω1: -4.9 [0.45] in controls, -3.7 [0.57] in anx1, *P* = 0.031) but not in w_2_ : -2.8 [0.71] in controls, -2.4 [0.76] in anx1 (P > 0.05). The smaller values of ω_1_ in anx1 correspond with an attenuation of the updates in volatility (less pronounced updates or reduction). The perceptual model parameters in anx2 did not significantly differ from those in control participants either (P> 0.05; mean and SEM values for w_1_ and w_2_ in anx2 were -5.4 [0.81] and -1.8 [0.74]).”

In the second, control experiment, the group-average values of ω1 and ω2 were: -4.1 (SEM 0.53) and -3.3 (0.29) for controls; -4.4 (0.38) and -3.6 (0.32) in anx3. There were no significant differences between groups in these values, *P* > 0.05.

7) It would be helpful to provide examples to give some intuition about what types of behavior would drive a change in "volatility". For example, can more information be provided to help the reader understand if the results (presented in Figure 10 for instance) enable predictions about subjects' behavior? If beta is high on one trial during the feedback period, does that mean that the model makes a small change in the volatility estimate? How does this influence what the participants are likely to do on the next trial?

Thanks for this question, which has motivated us to make a substantial improvement in the response models we use in the HGF analysis. We provide a detailed explanation below, but the summary can be stated here:

Yes, a higher value of beta power or burst rate during feedback processing is associated with a smaller update in the volatility estimate (smaller pwPE on level 2, ε2) in that trial. But also, with a smaller update in the belief about reward (ε_1_).

Regarding e_2_ , if a participant had a biased estimate of volatility (underestimation or overestimation), a drop in beta activity during feedback processing would promote a larger update in volatility (through ε2) to improve this biased belief. Similarly, a reduction in beta activity would also increase updates in reward estimates (through ε_1_), which in the winning response model is linked to the performance measure, and thus increases cvIKI_trial_.

Following the anxiety manipulation in our study we find a combination of biased beliefs about volatility and reward and increased feedback-locked beta activity, which would be associated with reduced values of ε2andε_1._ Accordingly, biased beliefs are not updated appropriately in state anxiety.

In the revised manuscript, we provide a more complete description of the two-level HGF for the perceptual and response models. The perceptual model describes how a participant maps environmental causes to sensory inputs (the scores), whereas the response model maps those inferred environmental causes to the performance output the participant generates every trial.

In the following, we provide detailed explanations on these aspects: (A) how phasic volatility is estimated in the perceptual model, and (B) how changes in volatility may influence changes in behaviour. Ultimately, we address (C) how beta power and burst rate can drive the updates in volatility estimates.

A) Concerning the perceptual model, we have included the update equations for beliefs and precision (inverse variance) estimates at each level. This helps clarify what contributes to changes in the estimation of environmental volatility. An additional illustration is provided in the new HGF model schematic (Figure 5).

Estimates about volatility in trial k are updated proportionally to the environmental uncertainty, the precision of the prediction of the level below, π^1, and the prediction error in the level below, δ1; volatility estimates are also inversely proportional to the precision of the current level, π2:

μ2k=u^2k+121π2kw1kδ1k,

With

w1k=expμ2k-1+ω1π^1k

We have dropped parameter k and the time step t from these expressions (see Mathys et al., 2011, 2014), as they take value = 1.

The expression exp(μ2^k-1^+ ω1) is often termed environmental uncertainty, and is defined as the exponential of the volatility estimate in the previous trial (before seeing the feedback) and the coupling parameter ω1, also termed tonic volatility (Mathys et al., 2011, 2014).

The equations above illustrate the general property of the HGF perceptual model that belief updates depend on the prediction error (PE) of the level below, weighted by a ratio of precisions.

Thus, a larger PE about reward, δ1, will increase the step of the update in volatility – participants render the environment to be more unstable. However, the PE contribution is weighted with the precision ratio: when an agent places more confidence on the estimates of the current level (larger precision _2_), the update step for volatility will be reduced. On the other side, a larger precision of the prediction at the level below (π^1) will increase the update in volatility. If the prediction about reward is more precise, then the PE about reward will be used to a larger degree (through the product π^1δ1).

Therefore, in addition to constant contributions from the tonic volatility ω1 to the update, the main quantity that drives the updates in volatility is the ratio of precision between lower and current level, thereby affecting how much the PE about reward contributes to the belief updating in volatility.

B) The revised manuscript tested several new more complete response models using as reference the work by Marshall et al. (2016). In that work, the authors described in a different paradigm how the participant’s perceptual beliefs map onto their observed log(RT) responses on a trial-by-trial basis, with the responses log(RT) being a linear function of PEs, volatility, precision-weighted PEs, and other terms (multiple regression). For that purpose, they created the family of scripts tapas_logrt_linear_whatworld in the tapas software.

We have now implemented similar models, but adapted to our task (scripts tapas_IKI_linear_gaussian_obs uploaded to the Open Science Framework data repository). The response models we tested aimed to explain a relevant trialwise performance parameter as a linear function of HGF quantities (multiple regression). The alternative models used two different performance parameters:

– The coefficient of variation of inter-keystroke intervals, cvIKI_trial,_ as a measure of the extent of timing variability *within* the trial.

– The logarithm of the mean performance tempo in a trial, log(mIKI), with IKI in milliseconds.

Furthermore, for each performance measure, the response model was a function of a constant component of the performance measure (intercept) and other quantities, such as: the reward estimate (μ_1_), the volatility estimate (μ_2_), the precision-weighted PE about reward (ε_1_), or the precision-weighted PE about volatility (ε_2_). See details in the revised manuscript. In total we assessed six different response models. Using random effects Bayesian model selection (BMS), we obtained a winning model that explained the performance measure cvIKI_trial_ as a linear function of μ_1_ and ε_1_:

cvIKItrialk=β0+β1μ1k+β2ϵ1k+ζ

The β coefficients were positive and significantly different than zero in each participant group (P < P_FDR_, controlled for multiple comparisons arising from 3 group tests), as shown in the new Figure 5 —figure supplement 1.

Thus, in addition to the estimated positive constant (intercept) value of cvIKI_trial_, quantities μ_1_ and ε_1_ had a positive influence on cvIKI_trial_, such that higher reward estimates and higher pwPEs about reward increased the temporal variability on that trial (less isochronous performance).

The noise parameter z did not significantly differ between groups (P > 0.05), and therefore we found no differences in how the model was able to estimate predicted responses to fit observed responses in each group.

Overall, the BMS results indicate that response models that defined the response parameters as a function of volatility estimates and pwPE on level 2 trial-by-trial basis were less likely to explain the data. However, because μ_2_ drives the step of the Gaussian random walk for the estimation of the true state x_1_, an underestimation in the beliefs about volatility (smaller μ_2_ as found in anxiety groups) would drive smaller updates about x_1_, ultimately leading to smaller cvIKI_trial_ – as our winner response model establishes. This can also be observed in Equation 6, where smaller values of the volatility estimate in the previous trial, μ_2_^k-1^ increase the precision of the prediction about reward (π^1), leading to smaller updates for μ1 (Equation 3).

As reported in the Discussion section, “Volatility estimates impact directly the estimations of beliefs at the lower level, with reduced m_2_ leading to a smaller step of the update in reward estimates. Thus, this scenario would provide less opportunity to ameliorate the biases about beliefs in the lower level to improve them.”

The new HGF results are shown in Figure 6, precisely illustrating that anx1 underestimated μ_2_ relative to control participants – when using the improved winning response model – thus accounting for the smaller cvIKI_trial_ found in this group.

C) In a similar fashion to the way we constructed response models in the new HGF analysis, we used a multiple linear regression analysis to evaluate the measure of feedback-locked beta power, and separately, the rate of long bursts as a linear function of two quantities, e_1_ and e_2_. This analysis is similar to the one we did in the previous version of the manuscript, but it is an improvement in two respects: It assesses the simultaneous influence of e_1_ and e_2_ on the measures of beta activity, and it uses trial-wise data in each participant to obtain the individual beta coefficients.

8) Generally, the EEG analysis opens up a massive search space (all electrodes, several seconds of data, block-wise analyses, trial-wise analyses, sample-wise analyses, power quantifications, burst-quantifications, long bursts, short bursts, etc.), and the presentation of the findings often jump around frequently between power quantification, burst-quantifications, block-wise, and trial-wise analysis etc. It would be much easier to follow if a few measurements were focused on that were a priori justified. These could be clearly laid out in the introduction with some explanation as to why they were investigated and what each measure might tell the reader. Then, if additional analyses were conducted, these should be explained as post hoc with appropriate justifications and statistical corrections.

Thanks for this suggestion. We completely agree with the reviewers and have considerably simplified the EEG statistical analyses. In addition, we have more explicitly stated in the revised introduction all our main hypotheses. The detailed aims and measures of the EEG analyses have been included at the beginning of the Results section to provide a clear overview.

Introduction:

Now we explicitly mention that prefrontal electrode regions were one of the regions of interest, together with “sensorimotor” electrode regions. In addition, we cite more work that identifies prefrontal regions as central to the neural circuitry of anxiety.

“Crucially, in addition to assessing sensorimotor brain regions, we focused our analysis on prefrontal areas on the basis of prior work in clinical and subclinical anxiety linking the prefronal cortex (dmPFC, dlPFC) and the dACC to the maintenance of anxiety states, including worry and threat appraisal (Grube and Nitsche, 2012; Robinson et al. 2019). Thus, beta oscillations across sensorimotor and prefrontal brain regions were evaluated.”

“We accordingly assessed both power and burst distribution of beta oscillations to capture dynamic changes in neural activity induced by anxiety and their link to behavioral effects.”

“EEG signals aimed to assess anxiety-related changes in the power and burst distribution in sensorimotor and prefrontal beta oscillations in relation to changes in behavioral variability and reward-based learning.”

Subsection “Electrophysiological Analysis”:

“The analysis of the EEG signals focused on sensorimotor and anterior (prefrontal) beta oscillations and aimed to separately assess (i) tonic and (ii) phasic (or event-related) changes in spectral power and burst rate. Tonic changes in average beta activity would be an indication of the anxiety manipulation having a general effect on the modulation of underlying beta oscillatory properties. Complementing this analysis, assessing phasic changes in the measures of beta activity during trial performance and following feedback presentation would allow us to investigate the neural processes driving reward-based motor learning and their alteration by anxiety. These analyses focused on a subset of channels across contralateral sensorimotor cortices and anterior regions (See Materials and methods section).”

Below, in the Results section of the exploration phase, when we introduce the methodology to extract bursts, we now state that due to the complementary information provided by duration, rate and slope of the distribution of bursts, we exclusively focus on the analysis of the slope when assessing tonic burst properties. The slope is already a summary statistic of the properties of the distribution (e.g. smaller slope [absolute value] indicates a long-tailed distriution with more frequent long bursts).

This will hopefully make the Results section more concise, as general average burst properties can be characterised by the slope of their distribution of durations:

Subsection “Electrophysiological Analysis”: “Crucially, because the burst duration, rate and slope provide complementary information, we focused our statistical analysis of the tonic beta burst properties on the slope or life-time exponent, τ. A smaller slope corresponds to a burst distribution biased towards more frequent long bursts.”

The separate analysis of bursts into long and brief bursts was inspired by the previous burst studies in parkinson’s patients showing the presence of longn bursts (> 500 ms) in the basal ganglia and linking those to motor symptoms and poorer performance. However, this was indeed a post-hoc analysis in our study, additionally motivated by the clear dissociation between long and brief bursts shown in Figure 7, and determined by the difference in slope between anx1 and controls. This analysis has now been correctly identified as post-hoc analysis:

Subsection “Electrophysiological Analysis”: “As a post-hoc analysis, the time course of the burst rate was assessed separately in beta bursts of shorter (< 300 ms) and longer (> 500 ms) duration,.…”

This split analysis is important in our results, as the longer burst properties seem to align better with the power results. While brief bursts are more frequent in all participants (and physiologically relevant), they seem to be here less related to task performance.

Subsection “Electrophysiological Analysis”: “The rate of long oscillation bursts displayed a similar time course and topography to those of the power analysis, with an increased burst rate after movement termination and after the STOP signal “

9) The EEG results could be better connected to the other findings: for instance, by correlating beta results to model volatility estimates or cvIKI variability, as described above.

The measures of feedback-related beta oscillations have now been correlated across participants with the index of across-trials cvIKI, reflecting motor variabilityility (Q3b above). Another specific correlation we have computed is that between motor variability, across-trials cvIKI, and volatility (Q3a above).

As explained in question Q7, we consider that the Hierarchical Bayesian model – now assessed in combination with an improved family of response models – is able to explain how in individual participants behaviour and beliefs about volatility or reward relate on a trial-by-trial basis.

In addition, now we use a multiple linear regression in individual subjects to explain trialwise power measures as a function of pwPE about volatility and reward (the main measures that are expected to modulate the EEG signal, Friston and Kiebel, 2009). This new analysis thus already is an assessment of trialwise relations between power and relevant computational quantities.

We hope the reviewers agree in that these analyses are sufficient to clarify those relationships (which in the case of the multiple regression analysis is already a type of correlation analysis).

What our analyses do not clarify is the dissociation between beta activity being related to pwPE in level 1 and 2, respectively. It is likely that a combined analysis of beta and gamma oscillations in this context could help identify different neural mechanisms (potentially with a different spatial distribution) separately driving belief updating through e_1_ and e_2_. This an investigation that we are currently completing in the context of a different study.

10) The reviewers felt that an important contribution of this paper was the potential non-motor findings related to sensorimotor beta. However, because there were also motoric differences between conditions, it seems very important to verify whether the beta differences were driven by motoric differences or anxiety-related manipulations. We appreciate the analyses in Figure 8—figure supplement 1 to try to rule out the motoric contribution to the sensorimotor beta differences, but note that this only controlled for certain kinds of movement variability. We would like to see controls for other possible differences in movement between the conditions, for instance differences in movement length, or movement length variability.

This is a great suggestion. We have now made additional control analyses similar to the original Figure 8—figure supplement 1 to assess the differences in beta power and burst rate between a subset of control and anxious participants matched in these variables:

– Duration of the trial performance (movement length or total duration in ms) – Figure 8—figure supplement 2

– Variability of movement length (cv of movement length) – Figure 8—figure supplement 3

– Mean use of keystroke velocity in the trial – Figure 8—figure supplement 4

The results indicate that when controlling for changes in each of these motor parameters, anxiety alone could explain the findings of larger post-movement beta-band PSD and rate of longer bursts, while also explaining the reduced rate of brief bursts during performance.

In the original manuscript, we had reported that “General performance parameters, such as the average performance tempo or the mean keystroke velocity did not differ between groups, either during initial baseline exploration or learning”. This outcome also accounts for why the new control analyses support that motor parameters such as the mean performance duration or keystroke velocity are not confounding factors when explaining the beta activity effects in anxiety.

Finally, is there a way to verify if the participants moved at all after they performed the task?

The best way to address this question, in the absence of EMG recordings from e.g. neck or torso muscles, is to look at broadband high-frequency activity (gamma range above 50 Hz), which has been consistently associated in previous studies with muscle artifacts. For instance, in this review paper by Muthukumaraswamy (2013), the author identified 50-160 Hz gamma activity with postural activity of upper neck muscles, generated by participants using a joystick (Figure 2). Changes in beta activity in this task were identified as true brain activity related to neural processing of the task requirements.

The author also reported that EEG activity contaminated by muscle artifacts is typically maximal at the edges of the electrode montage (e.g. temporal electrodes) but can be also observed at central scalp positions.

In our experimental setting, we instruct participants to not move the torso or head during the total duration of the trial, from the warning signal through to the sequence performance until the end of the trial (2 seconds after the feedback presentation). And we always monitor EEG for muscle artifacts while participants familiarise with the apparatus and the sequences at the beginning of the experimental session.

We have performed a control analysis of higher gamma band activity, between 50-100Hz, and display the results in Figure 9—figure supplement 2. This figure excludes the power values at 50Hz and 100Hz related to power line noise (and harmonics).

We have evaluated these conditions in the learning blocks:

A) Gamma power within 0-1 s after feedback presentation, where participants should be at rest after completing the trial performance.

B) Gamma power within 0-1 s locked to a key press, when participants are moving their fingers.

C) Gamma power within 0-1 s locked to the initiation of the trial, when participants are cued to wait for the GO response, and can be expected to be mentally preparing but otherwise at rest.

We then performed the following statistical analyses to test for differences in gamma power:

– Condition A versus Condition C in bilateral temporal electrodes

– Condition A versus Condition C in bilateral and central sensorimotor electrode regions

– Condition B versus Condition C in bilateral temporal electrodes

– Condition B versus Condition C in bilateral and central sensorimotor electrode regions

In addition, focusing now only on the target period of the manuscript, the feedback-locked changes (A), we assessed differences between experimental and control groups:

– Condition A: anx1 versus controls in bilateral temporal electrodes

– Condition A: anx1 versus controls in bilateral and central sensorimotor electrode regions

– Condition A: anx2 versus controls in bilateral temporal electrodes

– Condition A: anx2 versus controls in bilateral and central sensorimotor electrode regions

Overall, we found no significant changes in high gamma activity in any of the assessed contrasts (P-values for panels A-F range 0.2-0.6; two-sample permutation test between two conditions/groups after averaging the power changes across the ROI electrodes and the frequency range 52-98Hz). This result rules out that the beta-band effects reported in the manuscript are confounded by simultaneous systematic differences in muscle artifacts contaminating the EEG signal (or by differences in non-task-related movement).

11) We would like to see what the between group differences for beta power and beta bursts look like during the rest period before the baseline? (For instance, if Figure 7 were generated for rest data?)

We have included this figure directly here as part of the reviewing process. The figure illustrates how during the resting state recording prior to the experimental task there are no apparent (nor significant) differences in the burst distribution between experimental and control groups (assessed in all electrodes and separately in contralateral sensorimotor electrodes).

[Editors' note: further revisions were suggested prior to acceptance, as described below.]

Essential revisions:Related to the updated model: We have summarized some aspects of the modeling that we believe would benefit from additional explanation (to make the manuscript more broadly accessible). We apologize that we did not bring some of these up in the first submission, but these questions arose either due to the use of the new model or because of clarifications in the revision that provided new insight to us about the model.1) Explanation/interpretation of the Bayesian modeling – Definitions:Thank you for Figure 5 – this added clarity to the modeling work but we still are having trouble understanding the general structure of the model. It would be helpful to clearly define the following quantities that are used in the text (in the Materials and methods section before any equations are listed), and ideally also in a figure of example data."input" – does this mean the score for a specific trial k? We found this a little misleading since an "input" would usually mean some sort of sensory or perceptual input (as in Mathys 2014), but in this case it actually means feedback score (if we understood correctly).

Agreed. The term “input” – as used in Mathys et al., (2014) – is now specified in the introduction to the HGF model in the Results section and the Materials and methods section. In subsection “Bayesian learning modeling reveals the effects of state anxiety on reward-based motor learning”, we also give two examples of “sensory input” being replaced by a series of outcomes:

“In some implementations of the HGF, the series of sensory inputs are replaced by a sequence of outcomes, such as reward value in a binary lottery (Mathys et al., 2014; Diaconescu et al., 2017) or electric shock delivery in a one-armed bandit task (De Berker et al., 2016). In these cases, similarly to the case of sensory input, an agent can learn the causes of the observed outcomes and thus the likelihood that a particular event will occur. In our study, the trial-by-trial input observed by the participants was the series of feedback scores (hereafter input refers to feedback scores).”

In the case of the binary lottery or a one-armed bandit task, participants select one of two images and observe the corresponding outcome, which can be reward (0,1), or some other type of outcome, such as pain shocks (binary 0-1; de Berker et al., 2016). Thus, although the perceptual HGF is described in terms of “sensory” input being observed by an agent, in practice several studies use the series of feedback values or outcomes associated with the responses as input. This is also what we did in our implementation of the HGF: the input observed by participants, labeled u^k^ in the equations, is the feedback score associated with the response in that trial. Here, the HGF models how an agent infers the estate of the environment, which is the reward for trial k, m_1_^k^ (true state: x_1_^k^), using the observed outcomes (observed feedback score each trial, u^k^). We have included De Berker et al., 2016, as a new reference in the manuscript.

The HGF was originally developed by C. Mathys as a perceptual model, to measure how an agent generates beliefs about environmental states. Based on those inferred beliefs, the HGF can be subsequently linked to participant’s responses using a response model. This is the procedure we followed in our study: the response model explains participants’ responses as a function of the inferred beliefs or related computational quantities (e.g. PEs). See below please for the implementation of new – more interesting – response models suggested by the reviewers.

Also, please define u^k^ before Equation 3 and Equation 4, and ideally somewhere in Figure 5;

We have included in Figure 5 the definition of input u^k^, which is the observed feedback score for the trial (normalized to range 0-1). The definition is also presented at the beginning of the subsection “Computational Model”:

“In many implementations of the HGF, the sensory input is replaced with a series of outcomes (e.g. feedback, reward) associated with participants’ responses (De Berker et al., 2016; Diaconescu et al.,2017).”

“The HGF corresponds to the perceptual model, representing a hierarchical belief updating process, i.e., a process that infers hierarchically related environmental states that give rise to sensory inputs (Stefanics, 2011; Mathys et al., 2014). In the version for continuous inputs we implemented (see Mathys et al. 2014; function tapas hgf.m), we used the series of feedback scores as input: u^k^: = score; normalized to range 0-1. From the series of inputs, the HGF then generates belief trajectories about external states, such as the reward value of an action or a stimulus.”

In Figure 5 we have additionally indicated which performance measure we used as response y^k^, based on the winning model.

Please clarify how the precision of the input is measured? (used in Equation 3).

Here we followed Mathys et al., (2014) and the HGF toolbox that recommend to use as prior on the precision of the input (p_u_^0:^ estimated in the logarithmic space) the negative log-variance of the first 20 inputs (observed outcomes). More specifically:

log(p_u_^0^) is the negative log-variance of the first 20 feedback scores.

This prior is now included in Table 1.

That is, for a participant with very stable initial 20 outcomes, the variance would be small (<1), and the log-precision on the input would be large: the participant is initially less uncertain about the input.

By contrast, a participant with larger variability in feedback scores across the first 20 trials would have a small prior value on the precision of the feedback sores: the participant attributes more uncertainty to the input.

When mentioning the precision of the input in the manuscript (subsection “Computational Model”) we refer the readers to Table 1.

"predicted reward" – from the Mathys, 2014 paper we gathered that this is the mean of x_1_ obtained on the previous trial? Is this correct? If so, please clarify/emphasize. To increase broad accessibility of the manuscript, it would be helpful to summarize in words somewhere what the model is doing to make predictions. For example, in subsection “Computational Model” we weren't sure what the difference is between prediction of x_1_ and expectation of x_1_. Typically this terminology would correspond to: prediction = E(x_1_ on trial k | information up to trial k-1), and expectation = E(x_1_ on trial k | information up to trial k) but it wasn't clear to us.

Agreed. Thanks for pointing this out. Yes, the reviewers assumed correctly:

The difference between the prediction of an estimate (denoted by the diacritical mark “hat” or “^”), μ^ik and its expectation μik , is that the prediction is the value of the estimate before seeing the input in the current trial k, therefore μ^ik=μik-1. We have made this more explicit in the equations and in the text in subsection “Computational Model”:

The first term in the above expression is the change in the *expectation* or current belief μik for state x_i_, relative to the previous expectation in trial k-1, μik-1. The expectation in trial k-1 is also termed *prediction*, μik-1=μ^ik, denoted by the “hat” or diacritical mark “^”. The term prediction refers to the expectation before seeing the feedback score on the current trial and therefore corresponds with the posterior estimates up to trial k-1. By contrast, the term expectation will generally refer to the posterior estimates up to trial k. In addition, we note that the term belief will normally concern the current belief and therefore the posterior estimates up trial k. “

In addition, when referring to Variational Bayes and the derivation of update equations (Mathys et al., 2014, appendices), we add in subsection “Computational Model”:

“coupling between levels indicated above has the advantage of allowing simple variational inversion of the model and the derivation of one-step update equations under a mean-field approximation. This is achieved by iteratively integrating out all previous states up to the current trial k (see appendices in Mathys et al., 2014).”

"variance vs. precision vs. uncertainty" – These are well defined words but it would also help immensely to only use "variance" or "precision" or "uncertainty" in the explanation/equations. Mentally jumping back and forth gets confusing.

Agreed. In the Results section we have now more consistently used uncertainty, as this is the quantity that is directly obtained in the HGF toolbox and may also be understood in a more intuitive way by the readers. In the methods and materials section, however, we have maintained the term precision in the equations, as they have a simplified form this way.

When introducing precision-weighted PEs, we have of course kept that term, as this is what al authors use. But when analyzing the HGF belief trajectories and related uncertainty we have tried to avoid using “precision”.

The connection between both terms is now additionally made in subsection “Computational Model”:

uncertaintyσioritsinverse,precisionπi=1σi

"belief vs. expectation" – Are these the same? What is the mathematical definition (is it Equation 3 and Equation 7)?

See above.

“In addition, the term belief will generally refer to the current belief and therefore to the posterior estimates up trial k.”

"pwPE" – please list the equation for this somewhere in the methods, ideally before use of the epsilons in the response variable models.

We have clarified this in subsection “Computational Model”:

“Thus, the product of the precision weights and the prediction error constitute the precision-weighed prediction error (pwPE), which therefore regulates the update of the belief on trial k”

Δμik=ϵi

And have included Equation (14) and Equation (15) for e_1_ and e_2_, respectively. These equations are simply a regrouping of terms in Equation (6) and Equation (10) in subsection “Computational Model”.

2) Inputs/outputs of the model:Inputs – we gather that the input to the model is the score that the participant receives. Then x_1_ gets updated according to Equation 3. So x_1_ is tracking the expected reward on this trial assuming that the reward on the previous trial must be updated by a prediction error from the current trial? Is this reasonable assumption for this task? What if the participants are exploring new strategies trial to trial? Why would they assume that the reward on the next trial is the same as the current trial (i.e. why is the predicted reward = uik1?) Or is this the point (i.e. if trial to trial the subjects change their strategy a lot that this will end up being reflected as a higher "volatility"?) It would be helpful to outline how the model reflects different regimes of behavior (i.e. what does more exploratory behavior look like vs. what is learning expected to look like).

Using the HGF and the new response models (see below, we have followed the reviewers suggestion to link the change in responses cvIKI_trial_ from trial k-1 to k to computational quantities in the previous trial k-1), we can better address the relation between a behavioral change (i.e. a change in strategy) and the belief estimates. We have also created Figure 5—figure supplement 1 for simulated responses. This figure allows us to observe how different behavioral strategies impact belief and uncertainty estimates. We considered agents whose performance is characterized by (a) small and consistent task-related behavioral changes from trial to trial, (b) larger and slightly noisier (or more exploratory) task-related behavioral changes from trial to trial, (c) very large and very noisy (high exploration) task-related behavioral changes from trial to trial.

We explain below in our answer to point 3, the details of how these types of behavior influence belief and uncertainty estimates but the summary is:

If “the participants are exploring” more “new strategies trial to trial” then they will observe more different types of scores, and the distribution of feedback scores will be broader. This leads to a broader distribution of the expectation of reward, m_1_, and therefore higher uncertainty about reward. Simultaneously this is associated with increased volatility estimates and smaller uncertainty about volatility. The higher volatility estimates obtained in agents that exhibit a more exploratory behavior do not necessarily reflect pronounced increases across time in volatility but rather a lack of reduction in volatility. This effect results from smaller update steps in volatility estimates, due to both high s_1_ in the denominator of the update equations for volatility and low s_2_ in the numerator, see Equation (5).

So the main link is between a more exploratory behavior leading to more variable reward estimates (which feedback into the update equations as prediction errors at the lower level and as an enhanced uncertainty in volatility, s_1_). These effects ultimately maintain volatility estimates to a high level, or may even increase them.

Please, see below question 3 as we provide a more detailed explanation of Figure 5—figure supplement 1 and also of the new response model – which was suggested by the reviewers and it is actually a much better model (in terms of log-model evidence and also in terms of allowing to understand better the between-group differences).

Outputs – response models; please clarify why cvIKI_trial_ and log(mIKI) are the chosen responses since these are not variables that are directly responsible for the reward? We thought that the objective of this response modeling was to determine how a large prediction error on the previous trial would influence action on the next trial? Perhaps an output metric could be [similarity between trial k, trial k-1] = B0 + B1(uik-1) + B2(pwPE_k-1_)? So, depending on the reward and previous prediction error, you get a prediction of how similar the next trials' response is to the current trials' response? Right now, we don't understand what is learned from seeing that cvIKI_trial_ is higher with higher reward expectation (this is almost by necessity right? because the rewarded pattern needs high cvIKI) or higher prediction error.

Yes, we completely agree that this type of response model is more interesting. In the last manuscript we followed Marshall et al., which explain responses log(RT) in trial k as a function of HGF quantities in trial k. However, in our paradigm it is more interesting to link the HGF perceptual beliefs and their precision-weighted prediction errors to the “change” in behavior. We have now replaced as suggested the original response variables (cvIKI_trial_ and log(mIKI_trial_) at trial k) with their trial-wise difference: ΔcvIKI_trial_ or Δlog(mIKI_trial_) reflecting the difference between current trial k and previous trial k-1.

First, a clarification on why we had chosen as performance variables cvIKI_trial_ and log(mIKI_trial_), see subsection “Bayesian learning modeling reveals the effects of state anxiety on reward-based motor learning”:

“Variable cvIKI_trial_ was chosen as it is tightly linked to the variable associated with reward: higher differences in IKI values between neighboring positions lead to a higher vector norm of IKI patterns but also to a higher coefficient of variation of IKI values in that trial (and indeed cvIKI_trial_ was positively correlated with the feedback score across participants, nonparametric Spearman ρ = 0.69, P < 10e − 5). Alternatively, we considered the scenario in which participants would speed or slow down their performance without altering the relationship between successive intervals. Therefore, we used a performance measure related to the mean tempo, mIKI. “

Now we use those performance variables as well however the new response models include the difference between trial k and trial k-1 in those performance variables and link them to the belief estimates and pwPE in the trial before, k-1. The code is provided at the Open Science Framework, under the accession number sg3u7.

We have done family-level Bayesian model comparison (one family of models for ΔcvIKI_trial_ and a separate family of models for Δlog(mIKI)), followed by additional BMC within the winning family. The response model that had more evidence is based on the pwPEs (model HGF14, Equation 2):

cvIKItrialk=β0+β1μ1k+β2ϵ1k+ζ

This model explains the change in cvIKI_trial_ from trial k to k-1 as a function of pwPE on reward and volatility in the preceding trial. Moreover, we obtained an interesting between-group difference in the β_2_ coefficients of the response model, supporting that large pwPE on volatility promote larger behavioral changes in the following trial in control participants, yet they inhibit or constrain behavioral changes in anx1 and anx2 participants (see Figure 5 —figure supplement 3). In addition, in all groups, beta1 is negative, indicating that smaller pwPE on reward on the last trial (reduced update step in reward estimates) promotes an increase in the changes in the relevant performance variable, thus an increase in exploration. By contrast, in increase in m_1_ updates through large pwPE on reward, is followed by a reduction in cvIKI_trial_ (more exploitation).

Additional examples illustrating the implications of the winning response model are included as Figure 5—figure supplement 4 and Figure 5—figure supplement 5.

The former response models that assessed whether cvIKI_trial_ and log(mIKI_trial_) in trial k can explained by pwPE or belief estimates in the same current trial k have not been included in the new manuscript. However, for the reviewer team we provide the results of the BMS applied to the total of four families of models (two old families F_1_ and F_2_ for cvIKI_trial_ and log(mIKI_trial_) in trial k, HGF quantities in trial k; and two new families F_3_ and F_4_ for the change k-1 to k in cvIKI_trial_ and log(mIKI_trial_), HGF quantities in trial k-1). BMS using the log-family evidence in each family provided more evidence for the new families, F3 and F4, as indicated by an expected frequency of:

0.0160 0.0165 0.9335 0.0340

And an exceedance probability of

0 0 1 0

This demonstrates that the third family of models (related to ΔcvIKI_trial_) outperforms the other families.

3) Interpretation of the model:Is the message from Figure 6A that the expected reward is lower for anx1 than anx2 and control? Since the model is trying to predict scores from actual score data, isn't this result expected given Figure 4A. Can the authors please clarify this?

Correct. The HGF as a generative model of the observed data (feedback scores) provides a mapping from hidden states of the world (i.e. true reward x_1_) to the observed feedback scores (μ). Anx2 and control participants achieved higher scores (Figure 4) and therefore the HGF perceptual model naturally provides trajectories of beliefs about reward with higher expectation values, μ1, than in anx1. We acknowledge that this result is a kind of “sanity check” and is not the emphasis of the interpretation and discussion in the new manuscript. A mention of this expected result is included in the new manuscript, subsection “Bayesian learning modeling reveals the effects of state anxiety on reward-based motor learning”:

“Participants in the anx1 relative to the control group had a lower estimate of the tendency for x_1_.… This indicates a lower expectation of reward on the current trial. Note that this outcome could be anticipated from the behavioral results shown in Figure 4A.”

Using the new winning response model and associated results, the manuscript now places more emphasis on the obtained between-group differences in the response model parameters (β coefficients, Figure 5—figure supplement 3; see also Figure 10, Figure 10—figure supplement 1), as well as on the parameters of the perceptual HGF model (w_1_ and w_2_, with w_2_ being different between anx1 and control participants, and thus reflecting a different learning style or adaptation of volatility estimates in anx1).

We noticed that log(μ2) is lower for anx1 and anx2 than control. Should this correspond to a shallower slope (or a plateau in score that is reached more quickly) in Figure 4A over learning for anx1? If so, why don't we see that for anx2? If this is true, and given that cvIKI is no different for anx1, anx2, and control, wouldn't that mean that the reward rate is plateauing faster for anx1 and anx2 while they are still producing actions that are equally variable to control? So, are participants somehow producing actions that are variable yet getting the same reward – so they're getting "stuck" earlier on in the learning process? Can the authors provide some insight into what type of behavior trends to expect given the finding of Figure 6B-C? Right now all the reader gets as far as interpretation goes is that the anx1 group underestimates "environmental volatility" and that the mean behavior and cvIKI is the same across all groups.

To answer this question we have created Figure 5—figure supplement 1 for simulated responses (see legend for details).

The simulated responses have been generated by changing the pattern of inter-keystroke intervals on a trial by trial basis to a different degree, e.g. leading to a steeper (green lines) or shallower (pink lines) slope of change in cvIKI_trial_ (Figure 5—figure supplement 1B) and associated feedback score (Figure 5—figure supplement 1A). The feedback scores are illustrated in Figure 5—figure supplement 1A to align it to Figure 5—figure supplement 1C below displaying reward estimates, m_1_.

The figure demonstrates that a shallower slope in the feedback score function is associated with a shallower slope in the trajectory of reward estimates, m_1_, and smaller estimation uncertainty on that level, s_1_ (Figure 5—figure supplement 1E). More importantly, this scenario is also associated with smaller log(m_2_) estimates (Figure 5—figure supplement 1D) and greater estimation uncertainty s_2_ (Figure 5—figure supplement 1F). This case of shallower slope could represent anx1 participants (Figure 6).

These results also confirm the relationship between higher estimation uncertainty on one level, s_i_, and larger updates in the beliefs on that level, m_i_, that characterize the HGF. See Equation (5).

In addition to simulating responses that lead to different slopes of the feedback score trajectory, we have also simulated responses with different levels of noise or variation from trial to trial (while keeping the slope constant as underlying trend: green and pink trajectories). We considered these three scenarios:

i) Smooth trial-by-trial change in cvIKI_trial_ and corresponding feedback scores (linear trends in panels A and B)

ii) Slightly noisy or variable transition from trial to trial in cvIKI_trial_ and corresponding feedback scores – moderate noise level (slightly jerky trajectories, shown as darker green or pink lines)

This scenario represents an agent changing slightly more randomly their responses from trial to trial.

iii) Highly noisy or variable transition from trial to trial in cvIKI_trial_ and corresponding feedback scores – high noise level (pronounced jerky trajectories, shown as the darkest green or pink lines).

This scenario represents an agent changing significantly more randomly their responses from trial to trial.

Green lines, constant steep slope: Increasing level of noise in the behavioral responses associated with higher variation in trial-by-trial changes leads to higher log(m_2_) and reduced uncertainty about volatility, s_2._ In addition, the more variable changes in reward estimates have higher uncertainty, s_1_.

Pink lines, constant shallow slope: Similar results for increasing level of noise as described for the steep slope trajectories.

Thus, based on these simulation results, higher expectation on volatility in the HGF for continuous inputs can result from:

1) A steeper slope in feedback scores and therefore a steeper slope in the trajectory of perceptual beliefs for reward, m_1_.

2) More variable trial-to-trial changes in the observed feedback scores (corresponding with a more exploratory or noisier performance). This would also lead to more variable trial-to-trial changes in the perceptual beliefs for reward, m_1_.

These two cases come down to one single general case:

A broader range of values in the distribution of observed inputs (μ) that lead to a broader distribution of reward estimates, m_1_.

With regard to the HGF belief trajectories for volatility, μ_2_, in our experimental and control groups, we have noted in subsection “Bayesian learning modeling reveals the effects of state anxiety on reward-based motor learning” that:

“As indicated above, volatility estimates are related to the rate of change in reward estimates, and accordingly we predicted a higher expectation of volatility μ_2_ in participants exhibiting more variation to μ_1_ values.”

This is interesting but also simply implies that in participants achieving more different feedback score values (i.e. because they encounter all values from low to high scores), the volatility estimate will be higher (control group). By contrast, participants getting stuck at low score values (anx1) will have a reduced volatility estimate (due to a smaller rate of change of the estimate on the level below). This is what our findings in Figure 5 confirm, in line with the results for simulated responses in Figure 5—figure supplement 1. We anticipate this behavior of the HGF model in subsection “Bayesian learning modeling reveals the effects of state anxiety on reward-based motor learning”:

“Additionally, the HGF estimation of volatility (as change in reward tendency) was expected to be higher in participants modulating more their performance across trials and thereby observing a broader range of feedback scores (see different examples for simulated performances in Figure 5 —figure supplement 1).”

The case of anx2 is interesting as these participants had a similarly steep slope in feedback scores and in the trajectory for μ_1_ as the control group, however their log-volatility estimates μ_2_ and their uncertainty s_2_ resemble more the trajectories observed in anx1.

Accordingly, from the two cases contributing to higher volatility estimates indicated above, the likely explanation for the results in anx2 is that these participants must have a narrower distribution of encountered scores than control participants, and/or a smaller trial to trial change in the performance measure cvIKI_trial_.

We tested this prediction and found:

- The mean difference between trial k-1 and k in cvIKI_trial_ (our performance measure ΔcvIKI^k^_trial_) was significantly smaller in anx2 than control participants: mean 0.005 (SEM 0.0011) in controls, 0.0032 (0.0007) in anx2, P_FDR_ < 0.05. In anx1 participants this parameter was also smaller than in control participants: 0.0013 (0.0009), P_FDR_ < 0.05.

- The variance of the observed feedback scores was significantly smaller in anx2 than in control participants: mean 0.064 (SEM 0.004) in controls; 0.052 (SEM 0.003) in anx2, P_FDR_ < 0.05. A non-parametric Spearman correlation between these two parameters (rho = 0.4563, P = 0.0282) further confirmed that higher volatility estimates were associated with a larger variance of the distribution of feedback scores.

This is now presented as a post-hoc analysis in subsection “Bayesian learning modeling reveals the effects of state anxiety on reward-based motor learning”:

“…Thus, anx2 participants achieved high scores, as did control participants, yet they observed a reduced set of scores. In addition, their task-related behavioral changes from trial to trial were more constrained but also goal-directed as they indicated a tendency to exploit their inferred optimal performance, leading to consistently high scores. This different strategy of successful performance ultimately accounted for the reduced estimation of environmental volatility in this group, unlike the higher μ_2_ values obtained in control participants.”

Anx2 participants therefore showed a tendency to exploit more their inferred best response and thus observed fewer outcomes: they moved quickly from low to high feedback.

Interestingly, however, volatility estimates log(μ_2_) and ΔcvIKI^k^_trial_ were not correlated in the N = 60 population. We only found a correlation between log(μ_2_) and the variance of the feedback scores distribution, r = 0.30, p = 0.019. This also explains why there were no significant effects between groups in the degree of across-trials variability (cvIKI, Figure 4). So it seems that, although behavioral changes directly fed to the score modulation across trials, the most robust association was between the variance of the distribution of scores and volatility estimates.

In the adapted manuscript, following other papers using the HGF (see e.g. Marshall et al., 2016, Weber et al., 2019), the emphasis is placed now on the between-group differences in perceptual or response model parameters. Additionally, we maintain our emphasis on the analysis of pwPEs and how they relate to beta oscillatory activity and behavioral responses.

Does underestimating volatility mean that subjects just keep repeating the same sequence over and over? If so, can that be shown? Or does it mean that they keep trying new sequences but fail to properly figure out what drives a higher reward? Since the model is fit on the behavior of the participants, it should be possible to explain more clearly what drives the different model fits.

See above, please.

Related EEG Analysis: We greatly appreciated the clarified EEG analysis. Re-reading this section, we were able to understand what was done much better, but had two queries related to the analysis.1) We noted that the beta envelope in Figure 7A looks unusual. It looks almost like the absolute value of the beta – filtered signal rather than the envelope, which is typically smoother and does not follow peaks and troughs of the oscillation. Can the authors please clarify how this was calculated?

Thanks for spotting this. Yes, the figure was not correct. We have amended it and also uploaded to the OSF (https://osf.io/nv4m3/) the original code we used to compute the amplitude envelope from the band-pass filtered and Hilbert-transformed data. As in our earlier work (e.g. Herrojo Ruiz et al., 2014), the amplitude envelope A(t) of the instantaneous analytic signal was computed after applying the Hilbert transform to the bandpass-filtered raw data (12–35 Hz; two-way least-squares FIR filter applied with the eegfilt.m routine from the EEGLAB toolbox, Delorme and Makeig, 2004) spanning the full continuous recording of the task performance. Next, from the total beta-band amplitude envelope we extracted data segments corresponding with the epochs locked to the feedback presentation from -9 to 2 s.

We highlight here the main MATLAB steps:

% EEGdata: dimensions 64 channels x Nsampl, continuous data

% srate: sampling rate, 512Hz

f1=12; f2=35;% bounds for band-pass filter

betatot = eegfilt(EEGdata,srate,f1,f2);

amplitudebetatot=transpose(abs(hilbert(betatot')));

% after this step we extracted the epochs that were used to detect oscillation bursts

2) In subsection “Analysis of power spectral density”, the authors write: "The time-varying spectral power was computed as the squared norm of the complex wavelet transform, after averaging across trials within the beta range." This sounds like the authors may have calculated power after averaging across trials? Is this correct (i.e. was the signal averaged before the wavelet transform, such that trial to trial phase differences may cancel out power changes)? Or do the authors mean that they averaged across trials after extracting beta power for each trial? If the former the author should emphasize that this is what they did, since it is unconventional.

We have clarified this in the new version of the manuscript. In brief, the time-frequency transformation is first performed for each trial separately, followed by averaging. This is the standard practice to obtain the total oscillatory activity (induced + evoked). This thus converges with the reviewers’ expectations.

The analysis was done using Morlet wavelets based on convolution in the time domain. After the time-frequency transformation of each epoch, we obtained for each trial the wavelet energy, which was computed as the squared norm of the complex wavelet transform of signal x (for each trial):

Ext,f=Wxt,η2πf²

In this expression equation, h is the wavelet family function or number of cycles. The expression is taken from our earlier work e.g. Herrojo Ruiz et al. (2009).

Next, we assessed the spectral content of the oscillatory activity using the trial-average of the wavelet energy.

αWe have modified the text in the manuscript to clarify the analysis steps (and corrected a typo: the windows were set every 50ms). Subsection “Analysis of power spectral density”:

“Artefact-free EEG epochs were decomposed into their time-frequency representations using a 7-cycle Morlet wavelet in successive overlapping windows of 5 0ms within the total 12s-epoch. The frequency domain was sampled within the beta range from 13 to 30 Hz at 1 Hz intervals. For each trial, we thus obtained the complex wavelet transform, and computed its squared norm to extract the wavelet energy (Ruiz et al., 2009). The time-varying spectral power was then simply estimated by averaging the wavelet energy across trials within the beta range. “

In our earlier work we had used our own code to obtain the wavelet transformation with Morlet wavelets. Accordingly, we manually coded the trial-based time-frequency analysis followed by the calculation of the squared norm and then trial-averaging.

For this study, however, we used the built-in functions in the fieldtrip toolbox, which also follow this approach. The link to the uploaded code is provided in the next question. Here we only highlight the details of the fieldtrip analysis configuration:

cfg = [];

cfg.output = 'pow';

cfg.channel = 'all';

cfg.precision = 'single'

cfg.method = 'tfr';% implements wavelet time frequency transformation

% (using Morlet wavelets) based on convolution in the

% time domain.

cfg.foi = [13:1:30];

cfg.toi = -9:0.05:3;

cfg.width = 7;% default

cfg.trials = 1: length(EEG.trial);

cfg.keeptrials = 'yes'

TFRwav7 = ft_freqanalysis(cfg, EEG);

3) To try to understand point 2 above, we checked if the authors had shared their code, and found that, although data was shared, code was not, as far as we could tell. eLife does require code sharing as part of their policies (https://reviewer.elifesciences.org/author-guide/journal-policies) so please include that.

We have now included the code in the folder “Code for analysis of bursts and time-varying spectral power” of the Open Science Framework website for this study:

https://osf.io/nv4m3/

The script get_timecourse_wavelet.m (and Wiki) illustrates how to compute the time-varying spectral power in the beta-band (13-30Hz) after implementing the wavelet time frequency transformation (using Morlet wavelets) based on convolution in the time domain. It calls fieldtrip function ft_freqanalysis.m